# Unifying soil organic matter formation and persistence frameworks: the MEMS model

Andy D. Robertson*[1,2], Keith Paustian[1,2], Stephen Ogle[2,3], Matthew D. Wallenstein[1,2], Emanuele Lugato[4], M. Francesca Cotrufo[1,2]

[1] Department of Soil and Crop Sciences Colorado State University, Fort Collins, CO 80523, USA
[2] Natural Resources Ecology Laboratory, Colorado State University, Fort Collins, CO 80523, USA
[3] Department of Ecosystem Science and Sustainability, Colorado State University, Fort Collins, CO 80523, USA
[4] European Commission, Joint Research Centre (JRC), Ispra (VA), Italy

*Correspondence to*: Andy D. Robertson (Andy.Robertson@colostate.edu)

**Abstract.** Soil organic matter (SOM) dynamics in ecosystem-scale biogeochemical models have traditionally been simulated as immeasurable fluxes between conceptually-defined pools. This greatly limits how empirical data can be used to improve model performance and reduce the uncertainty associated with their predictions of carbon (C) cycling. Recent advances in our understanding of the biogeochemical processes that govern SOM formation and persistence demand a new mathematical model with a structure built around key mechanisms and biogeochemically-relevant pools. Here, we present one approach that aims to address this need. Our new model (MEMS v1.0) is developed upon the Microbial Efficiency-Matrix Stabilization framework which emphasises the importance of linking the chemistry of organic matter inputs with efficiency of microbial processing, and ultimately with the soil mineral matrix, when studying SOM formation and stabilization. Building on this framework, MEMS v1.0 is also capable of simulating the concept of C-saturation and represents decomposition processes and mechanisms of physico-chemical stabilization to define SOM formation into four primary fractions. After describing the model in detail, we optimise four key parameters identified through a variance-based sensitivity analysis. Optimisation employed soil fractionation data from 154 sites with diverse environmental conditions, directly equating mineral-associated organic matter and particulate organic matter fractions with corresponding model pools. Finally, model performance was evaluated using total topsoil (0-20 cm) C data from 8192 forest and grassland sites across Europe. Despite the relative simplicity of the model, it was able to accurately capture general trends in soil C stocks across extensive gradients of temperature, precipitation, annual C inputs and soil texture. The novel approach that MEMS v1.0 takes to simulate SOM dynamics has the potential to improve our forecasts of how soils respond to management and environmental perturbation. Ensuring these forecasts are accurate is key to effectively informing policy that can address the sustainability of ecosystem services and help mitigate climate change.

## 1 Introduction

The biogeochemical processes that govern soil organic matter (SOM) formation and persistence impact more than half of the terrestrial carbon (C) cycle, and thus play a key role in climate–C feedbacks (Jones and Falloon, 2009; Arora *et al.*, 2013). In order to predict changes to the C cycle, it is imperative that mathematical models describe these processes accurately. However, most ecosystem-scale biogeochemical models represent SOM dynamics with first-order transfers between conceptual pools defined by turnover time, limiting their capacity to incorporate

recent advances in scientific understanding of SOM dynamics (Campbell and Paustian, 2015). Due to the use of conceptual pools, empirical data from SOM fractionation cannot be used directly to constrain parameter values that govern fluxes between pools because diverse SOM compounds can have similar turnover times but are differentially influenced by environmental variables (Schmidt *et al.*, 2011; Lehmann and Kleber, 2015). As a result, empirical data is commonly abstracted and transformed before being used to parameterize or evaluate the processes of SOM formation and persistence that the model is intended to simulate (Elliott *et al.*, 1996; Zimmermann *et al.*, 2007). This has resulted in many conventional SOM models (e.g., RothC, [Jenkinson and Rayner, 1977], DNDC [Li *et al.*, 1992], EPIC [Williams *et al.*, 1984] and CENTURY [Parton *et al.*, 1987]) being structurally similar (i.e., partitioning total SOM into discrete pools based on turnover times determined from radiocarbon experiments; see Stout and O'Brien [1973] and Jenkinson [1977]) but each taking different approaches to simplify the complex mechanisms that govern SOM dynamics. Consequently, simulations of SOM can vary greatly between models, often predicting contrasting responses to the same driving inputs and environmental change (e.g., Smith *et al.*, 1997).

Structuring SOM models around functionally-defined and measurable pools that result from known biogeochemical processes is one way to help minimise these discrepancies. Two recent insights into SOM dynamics present a path towards addressing this issue. There is now strong evidence that: 1) low molecular weight, chemically labile molecules, primarily of microbial origin (Liang *et al.*, 2017), persist longer than chemically recalcitrant C structures when protected by organo-mineral complexation (Mikutta *et al.*, 2006; Kögel-Knabner *et al.*, 2008; Kleber *et al.*, 2011); and 2) each soil type has a finite limit to which it can accrue C in mineral-associated fractions (i.e., the C-saturation hypothesis) (Six *et al.*, 2002; Stewart *et al.*, 2007; Gulde *et al.*, 2008; Ahrens *et al.*, 2015). Structuring a SOM model around these known and quantifiable biogeochemical pools and processes has the potential to drastically reduce uncertainty by enhancing opportunities for parameterization and validation of models with empirical data. Furthermore, mechanistic models can have value in process explanation as well their value in predictive capabilities; such models can pinpoint the processes that have the greatest influence on a system even when they are not traditionally determined empirically.

Conventional SOM models readily acknowledge the importance of microbes in plant litter decomposition and SOM dynamics but model improvement was initially constrained by the concept that stable SOM included 'humified' compounds (Paul and van Veen, 1978). This quantified stable SOM using an operational proxy (high pH alkaline extraction) rather than relating stabilization to the mechanisms that are now widely recognised, such as organo-mineral interactions and aggregate formation (Lehmann and Kleber, 2015). As our contemporary understanding of stable SOM moves away from humification theory, so too must the way we represent SOM stabilization pathways in biogeochemical models. Similarly, many SOM models partition plant residues into labile and recalcitrant pools with turnover times that reflect the assumption of 'selective preservation' (i.e., chemically recalcitrant litter-C is only used by microorganisms when labile compounds are scarce). While many existing models do include a flux from labile residues into stable SOM, this is typically a much smaller absolute amount than the flux from recalcitrant residues. Evidence indicates that biochemically recalcitrant structural litter C compounds may not be as important in the formation of long-term persistent SOM as originally thought (Marschner *et al.*, 2008; Dungait *et al.*, 2012; Kallenbach *et al.*, 2016). Instead, they form light particulate organic

matter (POM) (Haddix *et al*., 2015), a relatively vulnerable fraction of SOM with a turnover time of years to
decades (von Lützow *et al*., 2006; 2007). Consequently, there have been several calls to represent this new
understanding and re-examine how microbial activity is simulated in SOM models (Schmidt *et al*., 2011;
Moorhead *et al*., 2014; Campbell and Paustian, 2015; Wieder *et al*., 2015).
Current conceptual frameworks more clearly link the role of microbes to SOM dynamics (e.g., Cotrufo *et al*.,
2013 and Liang *et al*., 2017), and generally isolate two discrete litter decomposition pathways for SOM formation
(Cotrufo *et al*., 2015): a 'physical' path through perturbation and cryomixing to move fragmented litter particles
into the mineral soil forming coarse POM, *vs* a 'dissolved' path where soluble and suspended C compounds are
transported vertically through water flow and, when mineral surfaces are available, form mineral associated
organic matter (MAOM). Microbial products and very small litter particles can be transported by both pathways,
forming a heavy POM fraction with 'biofilms' and aggregated litter fragments around larger mineral particles
(i.e., sand; Heckman *et al*., 2013; Ludwig *et al*., 2015; Buks and Kaupenjohann, 2016). Attempts to formulate
these empirical observations of litter decomposition into mathematical frameworks recently culminated with
development of the LIDEL model (Campbell *et al*., 2016), which in turn built upon the relationships of litter
decomposition described by Moorhead *et al*. (2013) and Sinsabaugh *et al*. (2013). While the LIDEL model was
evaluated against a detailed lab experiment of litter decomposition (Soong *et al*., 2015), it does not simulate SOM
pools and dynamics. In nature, litter decomposition processes and SOM formation processes are necessarily
coupled but are often studied and modelled separately. However, models that link litter decomposition to SOM
formation are required to represent SOM dynamics in ecosystem models.
Beside the processes of leaching and fragmentation that control the two pathways mentioned above, litter
decomposition processes that form SOM are governed by the balance between microbial anabolism and
catabolism (Swift *et al*., 1979; Liang *et al*., 2017). A recent paradigm has emerged that emphasizes the role of
microbial life strategies (e.g., K *vs* r, referring to copiotrophic and oligotrophic microbial functional groups) and
carbon use efficiency (CUE) in the formation of SOM from plant inputs (Dorodnikov *et al*., 2009; Cotrufo *et al*.,
2013; Lehmann and Kleber, 2015; Kallenbach *et al*., 2016). As a result, scientists have explored several
approaches to represent microbes in SOM models. Research has indicated that explicitly representing microbes
in a SOM model can provide very different predictions of SOM dynamics and include important feedbacks such
as acclimation, priming and pulse responses to wet-dry cycles (Bradford *et al*., 2010; Kuzyakov *et al*., 2010;
Lawrence *et al*., 2009; Schmidt *et al*., 2011). This research has shown that, compared to conventional models,
microbially-explicit SOM models have drastically different simulated responses to environmental change (Allison
*et al*., 2010; Wieder *et al*., 2015; Manzoni *et al*., 2016). However, these responses are generally validated against
data at microsite spatial scales and are not necessarily generalizable over larger spatial scales (Luo *et al*., 2016).
Microbes have been explicitly represented in SOM models in many ways and for many years, from relatively
simple approaches using a single microbial biomass pool or fungal:bacterial ratios (e.g., McGill *et al*., 1981,
Wieder *et al*., 2013 and Waring *et al*., 2013), to more complex associations with microbial guilds or community
dynamics based on dominant traits derived through genetic profiling (Miki *et al*., 2010; Allison *et al*., 2012;
Wallenstein and Hall, 2012). The MIcrobial-MIneral Carbon Stabilization (MIMICS) model (Wieder *et al*., 2014)

consolidated existing research at the time and uses the size of a microbial biomass pool together with Michaelis–Menten kinetics to feedback on C decay rates of SOM pools. While the MIMICS model and others (for an example see Manzoni *et al*., 2016), provide a potentially viable framework for explicitly representing microbes in a SOM model, it remains unclear whether this is practical given the lack of input data required to drive and validate these relationships (Treseder *et al*., 2012; Sierra *et al*., 2015). Furthermore, parsimony and analytical tractability are both key concerns for ecosystem models designed to operate over large spatial and temporal scales. While microbially-explicit models may be essential for addressing research questions at small spatial scales, they may introduce unnecessary, additional uncertainty to global simulations (Stockmann *et al*., 2013).

While microbial efficiency largely controls SOM formation rates, and microbial products are major components of the MAOM and the coarse, heavy POM fractions of SOM (Christensen 1992; Heckman *et al*., 2013) the long-term persistence of SOM is determined by mineral associations that are subject to saturation. Saturation limits for SOM were proposed more than a decade ago (Six *et al*., 2002) and have been supported by several empirical studies (e.g., Gulde *et al*., 2008; Stewart *et al*., 2008; Feng *et al*., 2012; Beare *et al*., 2014). Briefly, the concept of C-saturation suggests that each soil has an upper limit to the capacity to store C in mineral-associated (i.e., silt + clay, < 53μm) fractions, due to biochemical and physical stabilization mechanisms (e.g., cation bridging, surface complexation and aggregation) that are limited by a finite area of reactive mineral surfaces. While saturation kinetics are easy to define conceptually (Stewart *et al*., 2007), C-saturation as a concept has been adopted by only a few SOM models (Struc-C, Malamoud *et al*, 2009; COMISSION, Ahrens *et al*., 2015; MILLENNIAL, Abramoff *et al*., 2017). This is partly because its use in a SOM model requires a robust estimate of the specific site's saturation capacity. SOM saturation has been modelled using i) empirical regressions between silt + clay content and C concentration of that fraction (Six *et al*. 2002, as applied in COMISSION), and ii) empirical relationships between clay content and the derived '$Q_{max}$' parameter of Langmuir isotherm functions (Mayes *et al*., 2012, as applied in MILLENNIAL). As noted by Ahrens *et al*. (2015), the use of C-saturation kinetics in an ecosystem model would require a map of mineral-associated C saturation capacity, and since soil C stocks in silt + clay fractions can make up the majority of total soil C stocks, a lot of weight would be put on that single driving variable for each site. However, it is worth noting that when applying C-saturation concepts, only the mineral-associated organic matter (MAOM) fraction saturates. Other SOM fractions (e.g., particulate organic matter, POM) theoretically have no saturation limit (Stewart *et al*., 2008; Castellano *et al*., 2015).

Attempts to consolidate the concepts of microbial control on litter decomposition and mineral control on SOM stabilization resulted in the MEMS framework (Cotrufo *et al*. 2013). To date, we are aware of only one attempt to represent MEMS within a mathematical model, the MILLENNIAL model (Abramoff *et al*., 2017). However, this model does not simulate litter decomposition explicitly and as a result does not include the impact of litter input chemistry, which is a fundamental component of the MEMS framework and needed to improve ecosystem modelling, as discussed previously.

In this study we describe and demonstrate the application of a new mathematical model (MEMS v1.0) that applies three major concepts of SOM dynamics: 1) litter input chemistry-dependent microbial CUE informing SOM formation (Cotrufo *et al*., 2013), 2) separate dissolved *vs* physical pathways to SOM formation (Cotrufo *et al*.,

2015); and 3) soil C-saturation related to litter input chemistry (Castellano *et al*., 2015). The scope of this inaugural
model description is limited to representing these three concepts and is not intended to include every mechanism
relevant to SOM cycling. Our objective is to demonstrate the benefits of structuring a SOM model around key
biogeochemical processes, rather than turnover times. Using measured SOM physical fractions from 154 forest
and grassland sites across Europe, key parameters were optimised to improve model performance when simulating
POM-C (consisting of both light and heavy POM) and MAOM-C, under equilibrium conditions. The resulting
model was then used to test whether the behaviour of simulated SOM dynamics concur with the expected
theoretical relationships. Finally, the model performance in predicting soil C stocks at equilibrium was evaluated
by simulating 8192 forest and grassland sites across Europe, representing a diverse set of driving variables (i.e.,
climate, soil type and vegetation type).

## 2 Materials and Methods

### 2.1 Model description

The MEMS model (herein MEMS v1.0) is designed to be as parsimonious as possible while simulating the spatial
and temporal scales relevant to management and policy decision making. The model is structured (Figure 1) to
simulate plant litter decomposition explicitly with decomposition products defining C inputs to discrete soil pools
that can be isolated with common SOM fractionation techniques (Table 1). Each state variable in MEMS v1.0 can
be quantified directly using common measurement protocols and therefore calibration/evaluation data can be
generated with a single fractionation scheme (Table S1). Detailed information about the model structure, the
mathematical representation (i.e., differential equations) and how each mechanism is described mathematically
can be found in the supplementary material. All model parameters can be found in Table 2.

MEMS v1.0 is a SOM model that operates at the ecosystem-scale on a daily timestep. Carbon inputs to the model
are resolved for each source (in the case of multiple input streams, e.g., manure, crop residue, compost) discretely,
partitioning daily C inputs between solid-phase (C1, C2, C3) and dissolved (C6) litter pools as a function of litter
chemistry (nitrogen [N] content and the acid-insoluble [i.e., 'lignin'] fraction) that influences microbial
decomposition processes. This structure is similar to the LIDEL model (Campbell *et al*., 2016) and follows the
hypotheses that both N availability and lignin content influence decomposition by affecting microbial activity
(Aber *et al*., 1990; Manzoni *et al*., 2008; Sinsabaugh *et al*., 2013; Moorhead *et al*., 2013). Similar approaches have
also been used in many of the updated traditional SOM models (e.g., lignin:N ratios in CENTURY; Kirschbaum
and Paul, 2002). These input partitioning coefficients can be determined experimentally for each C input source
(Table 1 & S1). Upon reaching the soil, C compounds are then subject to biotic and abiotic processes that
transform and transport organic matter through an organic horizon and subsequent mineral soil layers. As
described here, MEMS v1.0 currently only simulates a surface organic horizon and a single mineral soil layer,
and does not yet differentiate between above- and below-ground litter input chemistry to avoid requiring
additional input parameters on root litter chemistry. However, the model architecture is sufficiently generalizable
to apply to multiple soil layers and/or multiple discrete sources of C input. Where possible we use the parameter
names and abbreviations from the LIDEL model (Campbell *et al*., 2016).

### 2.1.1 Microbe mediated transformations and dissolved organic matter (DOM) production

Many of the biogeochemical processes represented by MEMS v1.0 are assumed to be microbially-mediated (and therefore result in exo-enzyme breakdown and $CO_2$ production), but only two lead to C assimilation into a distinct microbial biomass pool – from the water-soluble and acid-soluble litter pools (C1 and C2, respectively). In the mineral soil (i.e., pools C5, C8, C9 and C10), microbial anabolism and catabolism are implicit and considered part of the turnover of each pool. This ensures parsimony and allows model parameters to represent the differences in microbial community for each pool, as opposed to the alternative of explicit microbial pools. The C transferred from the C1 and C2 litter pools into microbial biomass is defined by a dynamic CUE parameter controlled by the N content of the input material and the lignocellulose index (LCI; defined as the ratio between acid-insoluble to the sum of acid-soluble + acid-insoluble) of the litter layer (i.e., lower CUE results when a higher proportion of the litter is acid-insoluble). Including microbially-explicit processes in the litter layer helps to determine the proportion of C inputs that result in MAOM *vs* POM formation (see Liang *et al*., 2017) and allows for future model versions to account for distinctions between different points of entry for inputs (Sokol *et al*., 2018). The lack of C transferred from other pools (e.g., C3) into microbial biomass implies their decay from co-metabolism with the more labile C sources (i.e., Klotzbucher *et al*., 2011; Moorhead *et al*., 2013). Once assimilated within microbial biomass, the anabolism of microbial activity results in generation of microbial products (i.e., necromass) that form tightly bound aggregates of biofilms and small litter fragments around sand-sized soil particles (Huang *et al*., 2006; Buks and Kaupenjohann, 2016), and dissolved organic matter (DOM). These contribute to the heavy POM (C5) and litter DOM (C6) pools, respectively. While these specific processes are well supported by relevant literature, to retain parsimony and the generalizable structure required by an ecosystem scale model MEMS v1.0 represents microbial metabolism processes more generally (i.e., by linking them to a dynamic microbial CUE rather than specific community traits).

Even though not all pools explicitly produce microbial biomass, all pools do produce DOM. Recent studies have shown that DOM and small suspended particulates result from the decomposition and fragmentation of all forms of inputs including those characterized as 'inert', such as pyrolized material (Soong *et al*., 2015). Consequently, the model assumes that all microbially-mediated decomposition produces some C in DOM with rates specific to the pool from which the C originates. Since DOM generation is strongly influenced by the elemental composition of the input material (Soong *et al*., 2015), it is intrinsically linked to microbial CUE, employing the same formulation as LIDEL, which accounts for input N content and LCI of the litter layer (Campbell *et al*., 2016). At present, root exudation is not explicitly represented but the presence of a soil DOM pool (C8) will allow for incorporation of root exudation processes in later versions. More detail regarding the microbially transformed organic matter inputs *vs* those directly incorporated into the soil can be found in the supplementary materials.

### 2.1.2 Perturbation and physical transport

While microbial activity directly influences DOM production and therefore its transport with water flow (pool C8), the physical pathway to SOM formation (i.e., forming pools C5 and C10; POM) results from perturbation and fragmentation processes (Cotrufo *et al*., 2015). The exact mechanisms of perturbation are hard to generalize over the globally diverse conditions that an ecosystem scale model such as MEMS v1.0 is designed to operate. Consequently, the litter fragmentation and perturbation rate ($LIT_{frg}$) in MEMS v1.0 is represented as a first-order

process where the default value of $LIT_{frg}$ was informed by empirical estimates (e.g., Scheu and Wolters, 1991;
Paton *et al.*, 1995; Yoo *et al.*, 2011); but uncertainty can be reduced by relating this rate to specific site conditions
that reflect, in particular, soil macro- and mesofauna activity. The division of litter fragmentation between the C5
and C10 pool is derived from fractionation results that separate the light and heavy POM. The split between these
two fractions appears to vary with land use (Poeplau and Don, 2013), although the exact relationship is unclear.
Consequently, MEMS v1.0 applies an average over all land uses. Particulate organic matter is divided between a
heavy and a light pool because recent evidence suggests the two fractions are differentially influenced by
temperature and management linked to aggregation and land-use change (deGryze *et al.*, 2004; Tan *et al.*, 2007;
Poeplau *et al.*, 2017). Furthermore, the heavy, coarse POM pool can play an important role in soil nutrient cycling
(Wander, 2004) and it has a different turnover time to either the MAOM or light POM fraction (Crow *et al.*, 2007;
Poeplau *et al.*, 2018).

### 2.1.3 Liquid phase transport

Vertical transport of DOM can be simulated as a function of water flow in a process-based soil hydrology model.
However, in this first, standalone version, MEMS v1.0 assumes that DOM is transported rapidly downward
through percolation and advection according to a constant water flux. As with the $LIT_{frg}$ parameter, the rate of
vertical C transport (controlled by parameter $DOC_{frg}$) would ideally be site-specific, but is currently fixed at a
general, default value informed by relevant literature (Trumbore *et al.*, 1992; Kindler *et al.*, 2011). More
information can be found in the supplementary material and in Table 2.

### 2.1.4 Sorption and desorption with mineral surfaces

The organo-mineral complexes that define a large portion of MAOM-C in MEMS v1.0 operate under the
principles of Langmuir isotherms, which have also been used in the COMISSION and MILLENNIAL models
(Ahrens *et al.* (2015) and Abramoff *et al.* (2017), respectively). These isotherms represent a net C transfer between
soil DOM (pool C8) and MAOM (pool C9) that encapsulates all sorption mechanisms (e.g., cation bridging,
surface complexation, etc.). While MEMS v1.0 uses the same general Langmuir saturation function as the
MILLENNIAL model, it estimates maximum sorption capacity (parameter $Q_{max}$) differently. Here, we use sand
content to derive the maximum C concentration of the silt + clay fraction according to a regression calculated by
pooling all soils data reported by Six *et al.* (2002). This is then converted to C density using the site-specific soil
bulk density provided as a driving variable to the model.
In addition to the $Q_{max}$ parameter, the isotherm saturation function also relies on an estimate of a specific soil's
'binding affinity' (parameter $K_{lm}$). Typically, this is a product of a soil's specific mineralogy, influencing the type
of organo-mineral bonds that are formed and the strength of those bonds (Kothawala *et al.*, 2009). Furthermore,
the type of C compounds being sorbed are also key to defining an isotherm's binding affinity (Kothawala *et al.*,
2008; Kothawala *et al.*, 2012). This parameter can be very difficult to generalise without requiring exhaustive
information on soil physiochemical conditions (e.g., clay type, Fe/Al concentration, etc.), but the work of Mayes
*et al.* (2012) presented an empirical relationship between $K_{lm}$ and native soil pH, with pH acting as a proxy for
mineralogical conditions. As a result, sorption rates to mineral surfaces are dependent on pH (see Equation 35 in
supplementary). This relationship (derived from isotherms calculated for 138 soils of varying taxonomies)
provides a good starting point for estimating $K_{lm}$ and is also used by the MILLENNIAL model (Abramoff *et al.*,
2017). It is worth noting that desorption is implicit in the Langmuir saturation function used by MEMS v1.0
(unlike the explicit representation in COMISSION, Ahrens *et al.*, 2015), meaning that when the MAOM pool
reaches saturation the net transfer from soil DOM to MAOM may be negative and C is transferred from MAOM
to DOM. The simulated sorption–desorption processes in MEMS v1.0 are directly derived from empirical data
and are similar to other SOM models (Wang *et al.*, 2013; Ahrens *et al.*, 2015; Dwivedi *et al.*, 2017).
**2.1.5 Heterotrophic respiration and controls on microbial activity**
Aside from the litter layer DOM (pool C6), each of the state variables in MEMS v1.0 decay with unique specific
maximum rates, with the resultant C flux being partitioned into $CO_2$ (aggregated into the C7 sink term) and an
accompanying decomposition product flux into other pools, mainly DOM. Thus, the decay rate constants represent
total mass loss potential, embodying DOM-C generation as well as $CO_2$ emissions, as per a recent decomposition
conceptualization (Soong *et al.*, 2015). The total amount of heterotrophic respiration is the sum of $CO_2$ produced
from the biotic decay of all model pools after other fluxes (e.g., DOM generation) are calculated (more detail can
be seen in the supplementary). While the maximum specific decay rates for most pools are fixed parameters
informed by empirical data (**Error! Reference source not found.**), several studies suggest linking decay rates of
recalcitrant compounds to those of more microbially-accessible compounds (Moorhead *et al.*, 2013; Campbell *et*
*al.*, 2016). This follows similar hypotheses to the priming effect, that chemically recalcitrant compounds (e.g.,
lignin, cutin and suberin) are processed co-metabolically when microbes act preferentially on more energetically
favourable compounds nearby (Carrington *et al.*, 2012; Větrovský *et al.*, 2014). Consequently, MEMS v1.0
applies this through use of the same functions as those used by the LIDEL model (Campbell *et al.*, 2016),
estimating the maximum specific decay rate of pool C3 with a relationship to parameter $k_2$ (i.e., the maximum
specific decay rate of the acid-soluble litter fraction, pool C2). At present, $CO_2$ emitted from soil mineralization
of DOM is associated with the values presented in Kalbitz *et al.* (2005).
**2.1.6 Decay rate modifiers**
Temperature is used as the main environmental control on maximum specific decay rates of each pool. The rate
modifying function used by MEMS v1.0 is adapted from that of the StandCarb model (Harmon and Domingo,
2001). This function is consistent with empirical data and enzyme kinetics, implying that microbial decomposition
rates peak at an optimum temperature with reduced rates above and below. Coefficients that define the function
also include the $Q_{10}$ and reference temperature for that specific pool. Therefore, the function can utilise empirical
data if available for a site. This is a relatively simple function that only accounts for temperature. Simulating the
influence of other important controls on decomposition, such as water, oxygen, pH and nutrients, are beyond the
scope of this inaugural version of the MEMS model but are central to future development efforts.
**2.1.7 Model implementation and driving variables**
MEMS v1.0 is a series of ordinary differential equations solved for discrete time steps by numerical integration
using finite differencing techniques from the Runge-Kutta family of solvers. Implementation is performed through
the deSolve package (Soetart *et al.*, 2010) written for R (all equations and associated detail can be found in
Supplementary Information). Parameters used to solve MEMS v1.0 are described along with their default values
and associated references in Table 2.

Initializing MEMS v1.0 requires external inputs of basic site characteristics (climatic and edaphic conditions as
well as land management information) and ideally measurements of daily C input. However, C inputs are rarely
available at daily time scales. Consequently, for this inaugural version of the MEMS model we employ a simple
function to interpolate daily C inputs from annual Net Primary Productivity (NPP), partitioning
aboveground/belowground and to the simulated soil layer using land-use specific root:shoot ratios and a simple
root distribution function (Poeplau, 2016). These driving variables are external inputs of the initial model version
but may be obtained from coupled climate and plant growth submodels in future versions, when incorporated into
a full ecosystem model. Details of these approaches are given in the supplementary materials and all required
driving variables are shown in Table 3. Since the major C pools can each be quantified using common analytical
methods (**Error! Reference source not found.**), the best way of initializing the size of these pools in MEMS v1.0
is to use measured data. However, when measured data are not available, a typical site simulation employs a
spinup that runs the model to steady-state conditions based on average climatic and edaphic conditions, as well as
average C inputs.

## 2.2 Global sensitivity analysis

The default parameter values (i.e., those governing C turnover and fluxes between pools) used by MEMS v1.0 are
informed by data from relevant literature (**Error! Reference source not found.**Table 2). However, different
studies may suggest different values based on discrete site conditions, meaning *a priori* estimates may not
necessarily be generalizable across all sites that the model could simulate. A variance-based global sensitivity
analysis was performed to determine each parameter's relative contribution to the change in each state variable
(i.e., determining which parameters have the largest influence on the size of each model pool). The sensitivity
analysis was repeated for different simulation lengths (1 – 1000 years) as different fluxes operate at different
temporal scales, thereby meaning that the relative importance of each parameter changes through time. Initial pool
sizes were set to 0 and the model was initialized to simulate a steady-state scenario based on average site
conditions (derived from ~8000 forest and grassland sites in the Land-Use/Land Cover Area Frame Survey
(LUCAS) dataset ([Toth *et al.*, 2013] – see Table 3). Specifically, this meant starting a model run with no C in the
system and gradually building up the litter and soil pools until they reached equilibrium based on driving variables
(soil type, C inputs, climate) that remain fixed over time. To evaluate how much each model parameter (e.g.,
decay rates, DOM generation rates, etc.; see Table 2) effects the amount of C in each pool (i.e., C1-C11; Figure
1) parameter values were changed to be higher or lower from their baseline and pool sizes are tracked over
simulation time. Note that all temperature modifier parameters ($T_{ref}$, $T_{opt}$, $T_{Q10}$, $T_{lag}$ and $T_{shp}$; Table 2) were
excluded in this sensitivity analysis as the resulting $T_{mod}$ has the same effect on all decay rates. Maximum and
minimum values of all other parameters (n = 24**Error! Reference source not found.**) were defined as 50 % above
and below the literature-derived (baseline) value (Table 2). Using Latin Hypercube techniques to sample within
the full parameter space, a global sensitivity varying all parameters was used to determine total variance for
changes to each model pool (i.e., how much each pool changes in size when all parameters vary up to 50 %).
Then, in turn, each individual parameter was fixed at its baseline value while all others varied. This defines each
parameter's contribution to a pool's variance, averaged over variations in all other parameters (Sobol, 2001;
Saltelli *et al.*, 2008) (i.e., how much each pool changes in size when all parameters, except one, vary up to 50%).
When normalized over the global sensitivity variance, a contribution index provides the proportion of variance
explained by each parameter. The analysis was run 10,000 times to define the total parameter space and the whole
procedure was repeated annually for simulation lengths between 1 to 1000 years. Put simply, 10,000 different
combinations of parameter values between the minimums and maximums were used to repeatedly run the model
for 1000 years given average site conditions. The results showing changes in pool size correspond to the changes
in parameter values (e.g., when maximum decay rate of MAOM is increased, pool C9 may decrease in size but
other pools may increase). The impact that a single parameter has on pool size, compared to that of all parameters,
is described by the contribution index, where the total effect of all the parameters is equal to the maximum change
in pool size. Note that the results of a global sensitivity analysis of this kind are non-directional and do not indicate
whether a parameter increases or decreases a pool size, but rather that it simply changes from the baseline.
**2.3 Model response to changes in driving variables**
To determine the model's steady-state response to changes in each individual driving variable, a local one-at-a-
time (OAT) sensitivity analysis was performed by sequentially simulating different equilibrium conditions for
1000 years. The baseline estimates for edaphic inputs, temperature and C input quantity were informed by the
LUCAS dataset ([Toth *et al.*, 2013] – see Table 3 and below for more details), with mean values defining the mid-
points and ranges defined as the minima and maxima. Litter chemistry driving variables were adapted from the
ranges described by Campbell *et al.* (2016). Note that while typically described as a sensitivity analysis, an OAT
approach is not as robust as variance-based techniques because it cannot determine interactions between input
variables. However, OAT results are easier to interpret as there are no confounding impacts and relationships
observed are solely a result of changing one variable. Additionally, we assess the model's qualitative relationships
between driving variables by comparison to a study by Castellano *et al.* (2015); combinations of high/low sand
content and high/low soil pH were used to examine whether model projections agree with the hypothesized
relationships between input litter chemistry and MAOM-C stocks at steady-state. In these scenarios, alfalfa
(*Medicago sativa*) and ponderosa pine (*Pinus ponderosa*) were used as examples of a high- and low-quality litter
input, respectively, with litter chemistry driving variables adopted from Campbell *et al.* (2016).
**2.4 Parameter optimization**
**2.4.1 LUCAS dataset and soil fractionation data**
Parameter optimization for MEMS v1.0 used data from the LUCAS dataset (Toth *et al.*, 2013). This dataset
contains basic soil properties including C data for almost 20,000 sites across Europe, sampled in 2009,
representing a wide spatial range over 25 countries with diverse gradients of soil types, climates and land uses
(Figure S1). Complimented with geo-referenced estimates of annual NPP from MODIS satellite data (ORNL
DAAC, 2009), and daily temperature data from the Climate Prediction Center's Global Temperature (CPC-GT)
database (NOAA, 2018), this provided all driving variables required to run MEMS v1.0. The use of
modelled/interpolated NPP and climate data is not recommended over measurement data directly collected from
the site(s) being simulated, but for the analysis herein these measured data were unavailable.

A representative subsample (Figure S2) of forest and grassland sites from LUCAS were selected for fractionation
to generate data for POM and MAOM pools (see dataset online available at the European Soil Data Centre).
Specifically, topsoil (0-20 cm) samples from 78 grassland sites and 76 forested sites were fractionated by size (53
μm) after full soil dispersion in dilute (0.5 %) sodium hexametaphosphate with glass beads on a shaker. The
fraction passing through (< 53 μm) was collected as the MAOM, while the fraction remaining on the sieve was
collected as the POM. It is worth noting that this fractionation did not separate the POM into a light and a heavy
POM, as represented in MEMS v1.0 (i.e., C5 and C10), thus these model fractions were combined for data-model
comparisons (see below). After drying to constant weight in a 60 °C oven, each fraction was analysed for C and
N concentration in an elemental analyser (LECO TruSpec CN). Samples from sites with a soil inorganic C content
greater than 0.2 % (as reported in the LUCAS database) were acidified before elemental analyses to remove
carbonates, so that the %C of each fraction represented the organic C only. Carbon concentrations of each fraction
and the total soil organic carbon (SOC) were converted to stocks for the top 20 cm soil layer using bulk density
estimates reported with the LUCAS database. A georeferenced summary of these 154 sites can be seen in Figure
S2 and summary information of the fractionation data and comparisons between land use classes is shown in
Figures S3 and S4.

**2.4.2 Optimization procedure**

Informed by the global sensitivity analysis, four parameters accounted for ~60 % of the variation in steady-state
bulk (and MAOM/total POM) soil C stocks. These were *Nmid*, *k5*, *k9* and *k10* (see Table 2 **Error! Reference**
**source not found.**for details) and were used for optimization to improve model performance. Maximum and
minimum values representing realistic ranges of each parameter were informed by relevant literature and rounded
to appropriate boundaries (Table 2; Table S2): *Nmid* (0.875, 2.625), *k5* ($6.0^{-5}$, $1.0^{-3}$), *k9* ($1.0^{-5}$, $4.0^{-5}$), *k10* ($1.0^{-4}$,
$1.0^{-3}$). These values set the limits for Latin Hypercube sampling to define 1024 unique parameter sets that,
together, span the full range of each parameter. The fractionated LUCAS site data was used to train and test the
model, applying a repeated *k*-fold cross-validation approach (Kuhn and Johnson, 2013) to identify best parameter
values for the full variation of conditions at all 154 sites. Comparisons were made between measured soil C stocks
and those resulting from steady-state simulations for each site. Of these sites, 120 (78 %) were used for training
and the remaining 34 (22 %) were used for testing. Root mean squared error (RMSE) was applied as the objective
function. Using the training results, the set of parameters that reported the lowest RMSE for each fraction was
used to ensure this 'best' parameter set also performed well (i.e., RMSE was within 10 % of that reported for the
training sites) against the 34 sites of measured data withheld for testing. This process was repeated 10 times using
different subsets of the 154 sites for training and testing (i.e., 10 'folds' in the cross-validation approach).
To determine the optimized parameter values, a single fold was chosen at random from those that reported the
lowest RMSE for each subset of training sites (i.e., each fold). Optimized values differ depending on which
measured fraction is compared to model predictions (whether comparing pool C9 to measured MAOM-C, the sum
of pools C5 and C10 to measured total POM-C, or the sum of pools C5, C8, C9 and C10 to measured bulk SOC).
The new, optimized parameter values (Table S2) were derived from a randomly chosen fold that minimized RMSE
when compared to the MAOM fraction. This was chosen (instead of those optimized for POM or bulk SOC) since
the MAOM fraction is typically the largest single soil C pool and using this approach led to the biggest overall
decrease in RMSE when compared to all available data (Table S2). In future analyses, a more rigorous approach
may be to apply a cost function regarding all available measured pool data (e.g., including litter pool data when it
is also measured) but for our initial model evaluation we deemed this random choice sufficient.

## 2.5 Model evaluation for forests and grasslands in Europe

Having optimized key parameter values, the new global parameter set for MEMS v1.0 was used to simulate the
remaining forest and grassland sites of the LUCAS dataset for independent evaluation. Driving variables of
edaphic conditions and land-use type were extracted for each site from LUCAS and combined with daily estimates
of C inputs and temperature (derived from simple interpolations assuming a normal distribution of MODIS annual
NPP data [see Supplementary for details] and CPC-GT daily maximum and minimum air temperature data,
respectively). Where these data were unavailable, the site was removed from further evaluation. Three forest land-
use classes (as described in LUCAS) were included, along with the pure grassland land-use class. This resulted in
a final dataset of 8192 sites (3487 grasslands, 1713 coniferous forests, 1590 broadleaved forests and 1402 'mixed'
forests). Mixed forests are defined to contain coniferous and broadleaved species that each contribute > 25% to
total tree canopy. Summary information for these sites can be found in Figure S1. To differentiate between input
litter chemistry, root:shoot ratios and root distribution of the four land-uses, generic driving variables for each
were derived from relevant literature. Details of these inputs are shown in Table 3.
Each of the 8192 sites was initialized with zero pool sizes and simulated for 1000 years to achieve steady-state
conditions. This assumed the same intra-annual distribution of daily temperature and C input for each year.
Organic carbon content reported in LUCAS was converted to SOC stock using the estimated bulk density reported
with the database and reduced according to the measured rock/gravel content (Equation 1), i.e.,
$$SOC = C_{conc} * {}^{L}\rho * (1 - {}^{L}rock) \hspace{3cm} (1)$$
where $SOC$ is soil organic carbon stock in Mg C ha$^{-1}$, $C_{conc}$ is the measured C content in percent, ${}^{L}\rho$ is the bulk
density of soil layer $L$ in g cm$^{-3}$ and ${}^{L}rock$ is the rock content of soil layer $L$ expressed as a fraction. This total
SOC stock, was compared to MEMS v1.0 model output. In addition to comparing measured values with those
predicted at steady-state (which may not be an accurate assumption for many sites), a more general comparison
was performed to examine groups of sites under similar site conditions. Model performance was evaluated for
several classes of environmental conditions, with sites divided into above and below median values of mean
annual temperature (MAT, 8.3 ºC), mean annual precipitation (MAP, 687 mm), annual NPP (647 gC m$^{-2}$ yr$^{-1}$) and
sand content (50 %), for each land-use type. Several standard metrics for error and bias were used to evaluate
model performance following the flowchart presented in Smith *et al*. (1997), including Mean Absolute Error
(MAE), Mean Bias Error (MBE), Root Mean Square Error (RMSE), modelling efficiency (EF), and Coefficient
of Determination (CofD). Additionally, we used 16 environmental classes to derive an estimate of measurement
uncertainty based around sites of similar conditions (e.g., hot, wet, low input, sandy soil) for each land use. To
include both measurement and simulation error in the same evaluation metric, we applied a modified *F*-test
statistic that uses lack-of-fit sum of squares to account for both experimental and prediction uncertainty (see Sima
*et al*., 2018 for more information). The variance required to calculate these was derived by using the full number
of environmental classes as described above ($n = 16$). Due to the lower number of fractionated sites in each group,
only temperature and sand content were used as environmental classes (i.e., $n = 4$) to evaluate performance at
these 154 sites. One-way ANOVAs were performed to show where average model results were significantly
different from average measured C stocks. An α level of 0.05 was used to determine the significance of the
ANOVA and $F$-tests. Finally, we also use the standard errors for bulk topsoil C stocks of each environmental class
to determine the significance of RMSE assuming a two-tailed Student's t distribution and 95% confidence interval,
as described by Smith *et al.* (1997). All data processing and statistical analysis was performed in R (v3.4; R Core
Modelling Team, 2018).
**3 Results**
**3.1 Sensitivity and behaviour of MEMS v1.0**
**3.1.1 Parameter sensitivity at different timescales**
Bulk SOC stocks were sensitive to different sets of parameters depending on the duration of the simulation (Figure
2; Figure S5). Parameters that define litter fragmentation and perturbation rates (*LITfrg*) or microbial CUE (mainly
*LCmax*, *Nmax* and *Nmid*) are responsible for rapid (< 2 years) changes in C stocks, particularly those in the litter
layer and light POM. As simulation time increases, the influence of these parameters declines relative to the litter
and POM decay rate parameters, particularly *k5* and *k10*. Fifty years after simulations are initialized, more than
75 % of the sensitivity in total soil C stock was due to the maximum specific decay rate of light POM (i.e.,
parameter *k10*). After this point, its relative contribution to total C stock sensitivity diminishes (to approximately
45 %) as the parameters that define MAOM-C sorption become more important (i.e., coefficients that determine
the regression to calculate MAOM-C saturation capacity [*scIcept* and *scSlope*]). Overall, our sensitivity analysis
showed that the expected dynamics with different processes (e.g., litter fragmentation, microbial processing and
sorption) are operating at the appropriate timescales to structure SOM dynamics, and their associated parameters
are more, or less, important depending on the initial pool sizes and model run/experiment duration. Figure 2 can
be interpreted as a depiction of how the C pools of MEMS v1.0 are impacted by different parameters as each pool
accumulates over time.
**3.1.2 Soil carbon response to changing environmental conditions**
Alone, each driving variable (edaphic conditions, temperature, and input litter quantity/quality) in MEMS v1.0
has a discrete and non-linear relationship to the proportion of soil C stored in the MAOM and POM pools under
steady-state conditions (Figure 3). This analysis alters only one driving variable at time while holding others
constant at an average value. Bulk C stocks are predicted to be mostly MAOM in all cases except when C inputs
(*annNPP*) are very high (i.e., > 1.5 kg C m$^{-2}$ yr$^{-1}$; Figure 3). This results from the fact that the MAOM pool will
saturate at high input rates whereas the POM pools do not (Castellano *et al.*, 2015). Sand content and soil pH
influence a site's MAOM saturation capacity, and therefore a low capacity (i.e., high sand content) with
mineralogy associated with weaker organo-mineral bonding (i.e., high soil pH) has proportionally more total
POM. Litter input chemistry variables also have different, and sizable, impacts on whether SOM forms and
persists primarily in MAOM or in POM (as denoted by the MAOM:POM ratio). Note that POM in the

MAOM:POM ratio refers to total POM (i.e., pools C5 and C10 combined). The fraction of litter input that is hot-water extractable (*fSOL*) is a key determinant of MAOM formation rates and when *fSOL* is high, MAOM-C stocks at steady-state are predicted to be more than four times higher than POM-C stocks (Figure 3). Conversely, when input material has a high acid-insoluble (*fLIG*) content and a low N content (*LitN*) the size of the organic horizon increases and, over time, POM-C stocks approach a 1:1 ratio with MAOM-C stocks. Figure 3 shows the impact of changing one driving variable while all others remain constant. When many of these inputs vary at the same time, the relationships to MAOM:POM can be very different (for example, the model predicts twice as much POM-C as MAOM-C when simulating a sandy soil with coniferous vegetation and high *annNPP*).

MAOM-C saturation in the model is largely dependent on an interaction between the quantity of C inputs, the soil texture (i.e., sand content) and mineralogy (i.e., for which soil pH is used as a proxy). Figure 4 shows that our mathematical formulation of sorption to mineral surfaces generated a very similar relationship to that proposed by Castellano *et al*. (2015). When C inputs are low, litter input chemistry has the greatest influence on the MAOM-C stock under steady-state conditions. This is particularly true in soils with the strongest mineral bonding (i.e., low pH) and high sorption capacity (i.e., low sand %; Figure 4 top right panel).

**3.2 Improved simulation due to parameter optimization**

Initial parameter values derived from relevant literature provided good estimates judging from model performance with measured fractionation data (Table S2). Prior to optimisation, the difference between measured and modelled bulk soil C stocks of fractionated LUCAS sites was insignificant for all four land-uses (one-way ANOVA, $p > 0.05$). However, accounting for experimental and simulation uncertainty (variance calculated by four groups: divisions of high/low mean annual temperature and sand content) MEMS v1.0 only accurately described bulk SOC stocks for the grassland land-use class (*F*-statistic < 0.05). After optimisation, overall model fit with all soil C fractions (MAOM, total POM and bulk) was improved by increasing the maximum decay rate of MAOM (parameter *k9*) and decreasing the maximum decay rate of light POM (parameter *k10*), the maximum decay rate of coarse, heavy POM (parameter *k5*), and the inflection point for the logistic curve that defines the N effect on microbial CUE (parameter *Nmid*). This resulted in a lower RMSE against all measured data compared to baseline values (Table S2). Despite the improved model fit, the error in simulated values for broadleaved forest sites was still more than the error inherent to the measured data (at a 95% threshold and as defined by the modified *F*-test from Sima *et al*., 2018). This was primarily caused by two sites where measured total POM-C stocks were reported to be > 95 Mg C ha$^{-1}$ in the top 20 cm (Figure 5). When these sites were removed from statistical comparisons there were no significant differences between modelled and measured bulk SOC stocks for any land use class.

Measured fractionation data from the four major land-use classes showed a wide range of soil C stocks and a significantly different MAOM:POM ratio between grassland and forests (Figure 5; Figure S4). This was predominantly due to grassland topsoil (0-20 cm) having more MAOM and less total POM, compared to coniferous soils (Figure S3). On average, simulations of the fractionated sites agreed well with measured data, demonstrating no significant differences (p > 0.05) between measured and modelled C stocks of total POM or bulk soil for all land uses, and for MAOM at broadleaved, mixed and coniferous forest sites (Figure 5). The only statistically significant difference was between measured and modelled MAOM-C stocks for grassland sites (p <

0.01). However, measurements have a considerably larger range between minimum and maximum values than
did model simulations, particularly for total POM, which largely explained the high overall RMSE when
comparing all 154 sites (Table S2).

### 3.3 Model evaluation for forests and grasslands in Europe

Despite only including a few of the many factors that influence SOM dynamics, MEMS v1.0 was able to capture
the expected relationships between site conditions and total mineral soil C stocks based on an evaluation of the
optimized model with independent data (Figure 6). Mean absolute error over all sites (n = 8192) was low (MBE
= 1.1 MgC ha$^{-1}$) and CofD was above 1, indicating that the simulated C stocks capture the trend of the measured
data better than the mean of the measurements (Table 4). The main lack of fit was observed as the model
consistently underestimated bulk soil C stocks in forest systems with low mean annual temperature (MAT < 8.3
ºC) and sandy soil textures (sand content > 50 %) (Figure S6). When divided by land-use classes, grassland sites
had the lowest residuals and mixed forest sites had the highest (Figure 6; Figure S6). Using low and high divisions
of MAT, MAP, sand content and C input quantity, to account for variance between each of these groups ($n$=16),
RMSE indicated that the model predictions of C stocks fell within the 95 % confidence interval of the
measurements for coniferous and mixed forest sites. Using the same groups but also accounting for simulated
variance indicated that the accuracy of MEMS v1.0 predictions were statistically significant for all land uses
besides broadleaf forest sites ($F$-statistic > 0.05; Table 4). A geographic analysis of model performance indicated
that the model performed best across France and Northeastern Europe but poorly across the UK, Ireland and
Southern Sweden (Figure 7). Furthermore, topsoil C stocks of broadleaved sites in Southeastern Europe,
particularly Romania, were consistently overestimated by the model, especially when sites had low MAP (Figure
6; Figure 7).

In general, discrepancies between measured and modelled values were largest for the broadleaved forest land use
class (Figure S6). Results from analysis of the fractionated sites suggest that the model cannot achieve the very
high POM-C stocks measured at some sites. Optimized parameter values aim to produce a good overall model fit
but are unlikely to be able to capture the full range of measured values (for example, the lowest bulk topsoil C
stock for a broadleaved site was 7 Mg C ha$^{-1}$ whereas the highest was 218 Mg C ha$^{-1}$). A summary of model
performance against these 8192 evaluation sites is shown in Table 4. While the model's performance comparing
absolute C stocks appears good, this is done with the assumption that these topsoil C stocks at forest and grassland
sites in our analysis are at steady-state. This is unlikely to be true and therefore it is encouraging when general
trends are as expected (as is the case for many of the land uses and for many of the different environmental
divisions; Figure 6).

## 4 Discussion

MEMS v1.0 was designed to consolidate recent advances in our understanding of SOM formation and persistence into a parsimonious mathematical model that uses a generalizable structure which, after further development, can be implemented in Ecosystem and Earth System model applications. In this study we aimed to provide proof-of-concept that a model structure built around known biogeochemical mechanisms (Figure 1) and measurable pools could be advantageous for application over varied site conditions. Another advantage of using this novel structure is that each aspect is empirically quantifiable, allowing for straightforward model evaluation of both total and fractionated SOM, addressing a common concern among conventional SOM models (Campbell and Paustian, 2015).

### 4.1 Sensitivity and behaviour of MEMS v1.0

The relationships between model driving variables and soil C stocks at steady-state highlight the importance of litter chemistry on relative proportions of MAOM and total POM in MEMS v1.0 (Figure 3). This is generally because both POM pools accumulate C when input litter has a high acid-insoluble fraction and a low N content, resulting from reduced microbial accessibility and reduced DOM production (Scheibe and Gleixner, 2014). This trend is also common in empirical studies and often associated with land-use change from herbaceous to woody vegetation (Filley *et al*., 2008). Many of the parameters that influence the processes of POM formation and persistence (e.g., *LITfrg*, *Nmid*, *LCImax*, etc.) have relatively high importance (i.e., sensitivity) to changes in total SOM within relatively short time frames (i.e., < 10 years; Figure 2). This may potentially capture the important real-world trend that POM is typically more vulnerable to decomposition with disturbance compared to MAOM (Cambardella and Elliott, 1992). However, disturbance impacts were not evaluated in the inaugural study.

One main objective of structuring MEMS v1.0 around empirically-defined biogeochemical processes is so that it can accurately represent the timescales on which different processes operate, rather than being solely dependent on turnover times of conceptual pools. This is particularly relevant given our new understanding that the MAOM fraction has short-term dynamics (Jilling *et al*., 2018). Consequently, it is reassuring to see that this knowledge, which is incorporated into the MEMS v1.0 design, can be seen in Figure 2 (and Figure S5), where the parameters that operate on short time-scales also have an immediate impact on the MAOM pool given the complexity of controls in the model structure. The model's agreement with the hypothesized relationship from Castellano *et al*. (2015) is also reassuring, and represents an important proof of concept that associates litter chemistry and C saturation capacity with MAOM-C stocks at steady-state (Figure 4).

### 4.2 Model evaluation of MEMS v1.0

While average agreement between measured and modelled soil C stocks was very good for MEMS v1.0, the model failed to capture the wide range in total POM-C stocks that were observed at the fractionated LUCAS sites (Figure 5). This may be because this first version of the model does not include several of the key controls on POM dynamics, such as water/oxygen limitations (Keiluweit *et al*., 2016), aggregation (Gentile *et al*., 2011), activity of soil fauna (Frouz, 2018) and nutrient availability (Bu *et al*., 2015; Averill and Waring, 2018). There are also limitations of our approach given that very few of the sites will likely be under true steady-state conditions, leading to further discrepancies between model predictions and measured values. Furthermore, the variability in driving

variables of litter chemistry, N content and root:shoot ratios are underestimated when using our approach of
grouping many different land uses into broad classes.

When examining the comparison between measured and modelled bulk soil C stocks for the 8192 forest and
grassland sites, residuals were particularly large for high latitude forestry sites in southern Sweden and the UK
(Figure 7). We hypothesize that this is primarily due to the fact that MEMS v1.0 does not simulate soil moisture
controls on decomposition, and temperature effects are applied through a simple function. In reality, these sorts
of forest soils are known to have very high total POM-C stocks, resulting from decades of consistent inputs and
cold, wet climates resulting in low decomposition rates (Berg, 2000). Differences between measured and modelled
soil C stocks are also likely due to uncertainties with driving variables and specifically the MODIS estimates of
NPP. The 2009 NPP data from MODIS were used to estimate the C inputs to soils in our simulations, and these
data may not be representative of the average historical C inputs for those sites, which would impact the observed
amounts of soil C.
**4.3 Improving the parameters of MEMS v1.0**
The current iteration of the MEMS model is not intended to be able to simulate all scenarios and environmental
conditions, but this study indicates it can be reasonably accurate in simulating forest and grassland sites in Europe
under steady-state conditions (Figure 6; Table 4). That said, several of the parameters in MEMS v1.0 are either
poorly constrained or loosely defined in the current model. The *LITfrg* parameter, for example, defines a fixed
litter fragmentation and perturbation rate that transfers C from the structural litter pools (C2 and C3) belowground
(to C5 and C10). The global sensitivity analysis of MEMS v1.0 indicates that *LITfrg* is particularly important for
several model pools and total SOC early in a simulation (Figure 2; Figure S5). There are several areas of research
that may help make this process more mechanistic in MEMS and allow for feedbacks with site conditions (e.g.,
Scheu and Wolters, 1991; Yoo *et al*., 2011). One option to generalise the vertical transport of structural litter into
the soil may be to apply a diffusion approach that can be valid at the ecosystem scale, as described in the
SOMPROF model (Braakhekke *et al*., 2011). More empirical data to link site conditions to perturbation processes
(e.g., cryoturbation, bioturbation, churning clays) would help with this area of MEMS model development.

As with vertical distribution of physical SOM, the transport of DOM vertically between layers lacks a mechanistic
foundation in MEMS v1.0. A noteworthy approach that attempts to simulate this transport while also representing
bioturbation through diffusion and sorption-desorption processes is presented in the COMISSION model (Ahrens
*et al*., 2015). While these models apply more mechanistic functions to represent these key processes, one can
debate whether the increased complexity and computational demands are necessary. This, of course depends on
the model objectives and in MEMS v1.0 we have prioritised parsimony and deliberately minimised the number
of algorithms and parameters. While the model cannot yet address hypotheses about litter fragmentation or DOM
leaching, the generic structure of MEMS v1.0 can incorporate these processes in a more explicit manner in future
versions.

Additional parameters of MEMS v1.0 that are poorly constrained include those associated with the LIDEL model.
These parameters (specifically those related to DOM generation and microbial assimilation, see Table 2) were

estimated using Bayesian analysis that employed empirical data (Soong *et al*., 2015), but resulted in large posterior distributions with high uncertainty as noted by Campbell *et al*. (2016). Consequently, more data is required from different litter types to help constrain these parameter values. In particular, the amount of DOM leached from decaying microbial biomass (parameter $la_2$) is particularly important for MAOM formation when the pool is relatively small (< 25 years in Figure 2). MEMS v1.0 currently uses the estimated value from Campbell *et al*. (2016) for this parameter (0.19 g DOM g decayed microbial biomass$^{-1}$) but it is worth noting the reported posterior interval width was more than double this value (0.398 g DOM g decayed microbial biomass$^{-1}$). Similarly, the rate of microbial product generation from microbial biomass (parameter *B3*) was seen to be even more variable (Campbell *et al*., 2016). Empirically, the rate that microbial products are generated from microbial turnover is highly variable depending on the microbial community and the site conditions (Xu *et al*., 2014). While improving these parameters was outside the scope of this study, the path towards improved model performance can be addressed with new empirical data that better inform the model parameters.

**4.4 Opportunities for further development in MEMS v1.0**

In its current capacity, MEMS v1.0 is far from being able to simulate full ecosystems and is limited in scope regarding the land use scenarios it can simulate accurately. Specifically, the initial model does not simulate the hydrological or nitrogen cycles, and currently operates on a single soil layer. However, MEMS v1.0 has been built to have a modular architecture, with careful consideration given to how additional processes can be addressed through future model development.

The relationship between C and N in soils is fundamental to SOM dynamics (McGill and Cole, 1981), and therefore simulating the N cycle is at the forefront of plans to develop in the MEMS model. Since the MEMS model structure is based on soil fractions that can be physically isolated, each current soil C pool in MEMS v1.0 (i.e. pools C5, C8, C9 and C10) can also have a direct equivalent for N, and be consistent with the fractionation scheme for the C dynamics (Table S1). However, additional pools of nitrate and ammonium (and associated mechanisms to describe N- fixation, nitrification and denitrification) are needed to accurately describe plant-soil nutrient feedbacks. This highlights a major objective of future MEMS model development, i.e., to ensure the model can be easily coupled with existing modules that describe other aspects of the ecosystem (e.g., plant growth routines).

Another key feature of MEMS v1.0 is its ability to test specific hypotheses directly against empirical data, such as effects of soil priming on soil C stocks, effects of microbial feedbacks on OM sorption to mineral surfaces, or the effects of soil fauna on SOM formation. Because each of the existing model pools can be isolated physically and quantified, the rates of flux between these pools can also be quantified with isotopic tracer studies. Not only does this mean parameterization and evaluation data can be generated easily, but also that experiments can be designed with this mathematical framework in mind, specifically generating the data required to develop, evaluate and improve the model. While the current scope of MEMS v1.0 does not address all climate-C feedbacks, it does provide the basis for a more mechanistic model that can simulate SOM dynamics at the ecosystem scale.

## 5 Conclusions

As a carbon model designed around the processes that govern SOM formation, MEMS v1.0 provides an analytically tractable framework that can be used to test specific hypotheses by pairing empirical experiments with model simulations. While the inaugural version of this new model has limitations for direct evaluation with real-world measurements, on average, its performance with simulating steady-state conditions equates well with topsoil C stocks measured for ~8000 forest and grassland sites across Europe. Using a structure that aligns with our contemporary understanding of soil C dynamics, we also show that MEMS v1.0 is capable of accurately proportioning SOM between particulate and mineral-associated fractions by accounting for litter chemistry of the input material. By using litter chemistry to inform SOM formation pathways and edaphic conditions to inform the C-saturation capacity of a soil, MEMS v1.0 also shows consistent trends with experimental findings.

Next steps for MEMS model development will require detailed routines of N and hydrological cycling, as well as additional external drivers of SOM dynamics (e.g., land management practices). To reliably incorporate these aspects in the MEMS model will require effective collaboration between modellers and experimentalists to design studies that can both i) elucidate the underlying mechanisms that MEMS is built upon and ii) generate the parameterization and validation data required to reduce model uncertainty. Successful execution of this strategy will help to develop an ecosystem scale model that can improve assessments of management and policy action on sustainability of soils and associated ecosystem services.

**Code and data availability**

The LUCAS dataset can be found at https://esdac.jrc.ec.europa.eu/content/lucas-2009-topsoil-data with details of the larger European Soil Data Centre project at http://doi.org/10.17616/R34069. The additional MAOM and POM fractionation data for the 154 sites used in this analysis can also be found at European Soil Data Centre (ESDAC) of the European Commission Joint Research Centre (http://esdac.jrc.ec.europa.eu/). Access to model code is currently restricted to those directly collaborating with the MEMS development team. This is to ensure all bugs are caught and treated before release to the public. Detailed information and code relevant to specific questions can be provided upon request.

**Supplementary materials**

See separate attachments

**Author Contribution**

All authors contributed to the conceptualization of the MEMS model framework with MFC, KP and MDW formalizing the original foundational science. The *in-practice* model structure was then formalized by ADR, MFC, KP, SO and MWD. All model building, coding, statistical analyses and data analysis on the measured fractionation data and all model-measure comparisons was performed by ADR. Guidance on the optimisation procedures was provided by SO. The LUCAS database was provided by EL and all initial analysis and preparation of the data (e.g., refining bulk density estimates and NPP values for each site) was performed by EL. The project was overseen by all authors but primarily led by MFC. Funding was initially provided by MDW and later through grants awarded to MFC and KP. Developing, testing and evaluating the model was performed solely by ADR, as was all data presentation apart from the final conceptual diagram (Figure 1) which was outsourced (see acknowledgments). The manuscript was written and edited by ADR with comments and feedback from all co-authors.

**Competing Interests**

The authors declare that they have no conflict of interest.

**Disclaimer**

**Acknowledgments**

This research was supported by a National Science Foundation CAREER grant (number 255228) awarded to MDW, the US DOE Advanced Research Projects Agency-Energy program (ROOTS project; DE-FOA-00001565), the NSF-DEB Award #1743237 and the JRC (purchase order D.B720517). The authors like to thank Michelle Haddix for the soil organic matter fractionation work and Dr. Yao Zhang for help with regards to various parts of data generation

(e.g., climate inputs) and model development. The conceptual figure diagram was redrawn and stylized by Katie
Burnet.

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

**Figure legends**

**Figure 1 - Conceptual model diagram of MEMS v1.0 (see Table 1 for detailed information regarding each pool). Litter pools of MEMS v1.0 are defined as > 2mm particles and comprise of hot-water extractable (C1), acid-soluble (C2) and acid-insoluble (C3) fractions. A microbial pool (C4) and dissolved carbon pool (C6) are also part of the organic horizon and litter decomposition processes (see LIDEL for more information, Campbell et al., 2016). Soil organic matter (< 2mm particles belowground) comprises of a light particulate organic matter pool (light POM, C10) formed from the input through fragmentation and physical transfer of the structural litter residues (C2 and C3), a coarse heavy POM pool (C5) formed from both litter fragmentation and microbial residues coating sand-sized particles, a dissolved organic matter (DOM) pool (C8) formed from the decomposition of all other pools and receiving DOM from the organic soil layer, and a mineral-associated organic matter pool (MAOM C9), which exchanges C through sorption and desorption with the DOM. Arrows indicate the fluxes of carbon between the different pools. Carbon dioxide is produced from a number of these fluxes but for simplicity of graphical representation, these arrows are not linked to the carbon dioxide pool (C7). Deeper soil layers can be represented by the same structure, with or without root inputs depending on depth, but are not implemented in this inaugural version of MEMS v1.0.**

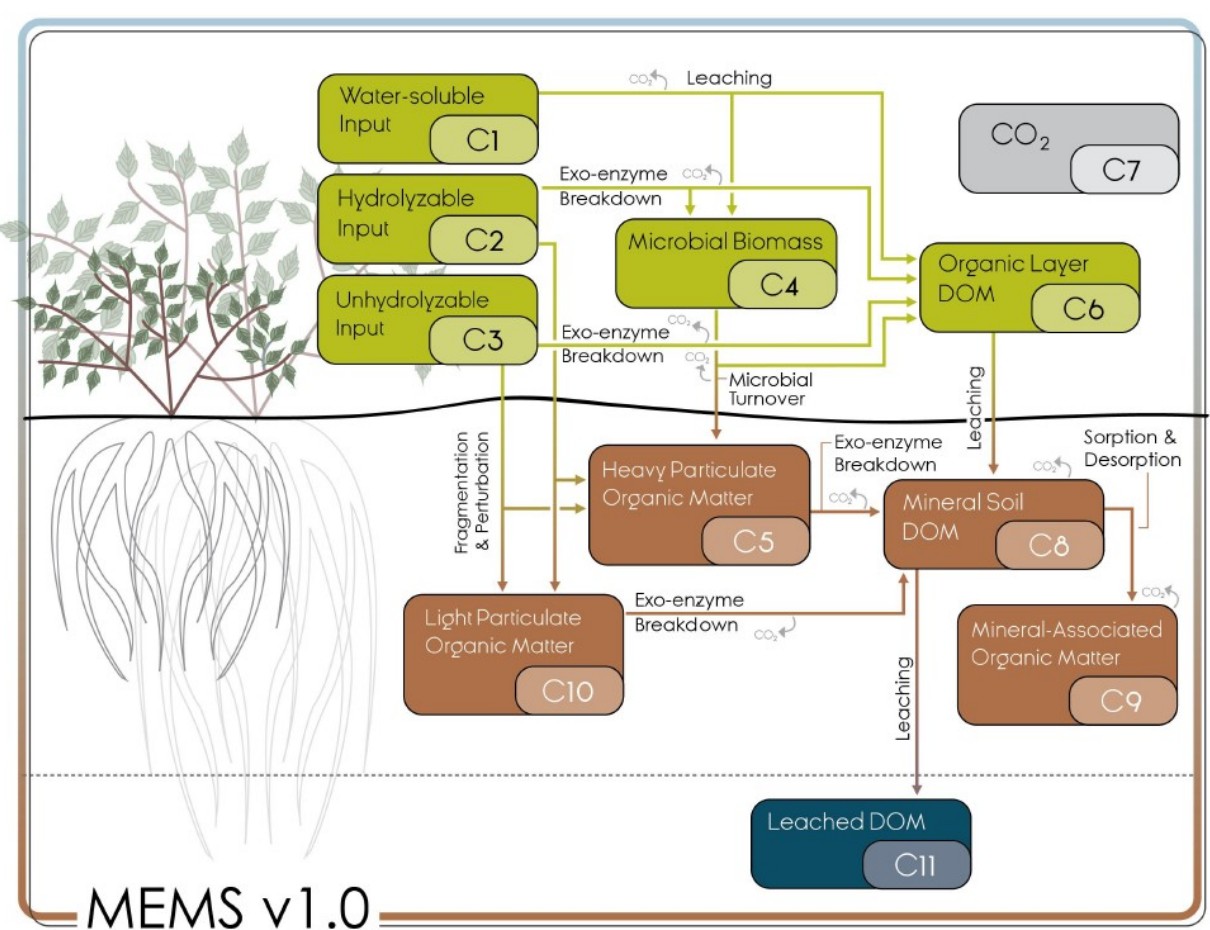

**Figure 2 - Global sensitivity analysis results showing the relative contribution of each parameter to a change in carbon stock of each pool in MEMS v1.0 (leached carbon to deeper soil layers [pool C11] is omitted for clarity) after simulation to steady-state. The two top left panels represent the sum of soil pools (C5, C8, C9 and C10) and organic layer pools (C1, C2, C3, C4 and C6), respectively. Details of each parameter and the abbreviations used can be found in Table 2. The sensitivity analysis was repeated annually for simulation times between 1 and 100 years, every 10 years after that to 400-year simulations and every 100 years after that up to a 1000-year simulation. Results are presented on a log scale in years. The four parameters that were optimized in our analysis (Table S2) are coloured to highlight their importance in the different pools (mid-point of logistic curve where nitrogen content of input influences microbial carbon use efficiency, *Nmid*, red; maximum decay rate of heavy particulate organic matter, *k5*, orange; maximum decay rate of mineral-associated organic matter, *k9*, blue; maximum decay rate of light particulate organic matter, *k10*, green). A fully colourised version of these results can be in Figure S5.**

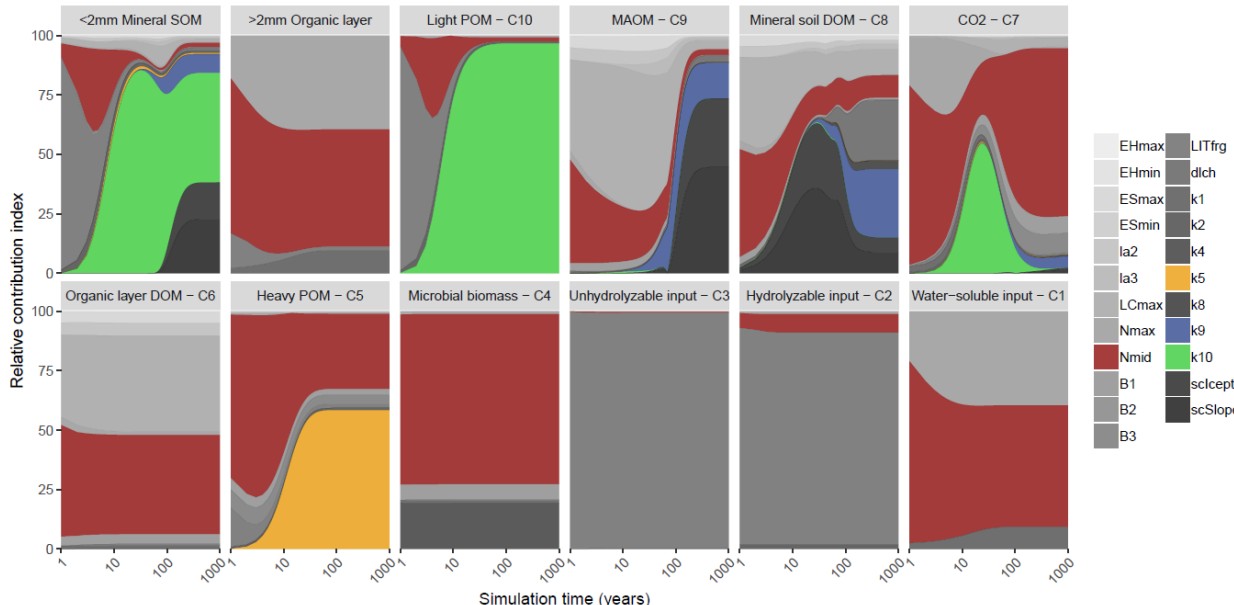

**Figure 3 - The ratio between mineral-associated organic matter and total particulate organic matter (MAOM:POM) under steady-state input conditions in MEMS v1.0 as a response to the full, realistic range of driving variables. Note, total POM refers to the sum of pools C5 and C10. Each input was varied individually while all others remained fixed at baseline values (indicated by dashed lines) – mean, maximum and minimum values for litter chemistry driving variables (*LitN*, *fDOC*, *fLIG* and *fSOL*) were derived from Campbell *et al*. (2016) and edaphic, climatic and C input driving variables (soil bulk density, sand content, soil pH, mean annual temperature and annual net primary productivity) were derived from the LUCAS dataset (Toth *et al*., 2013).**

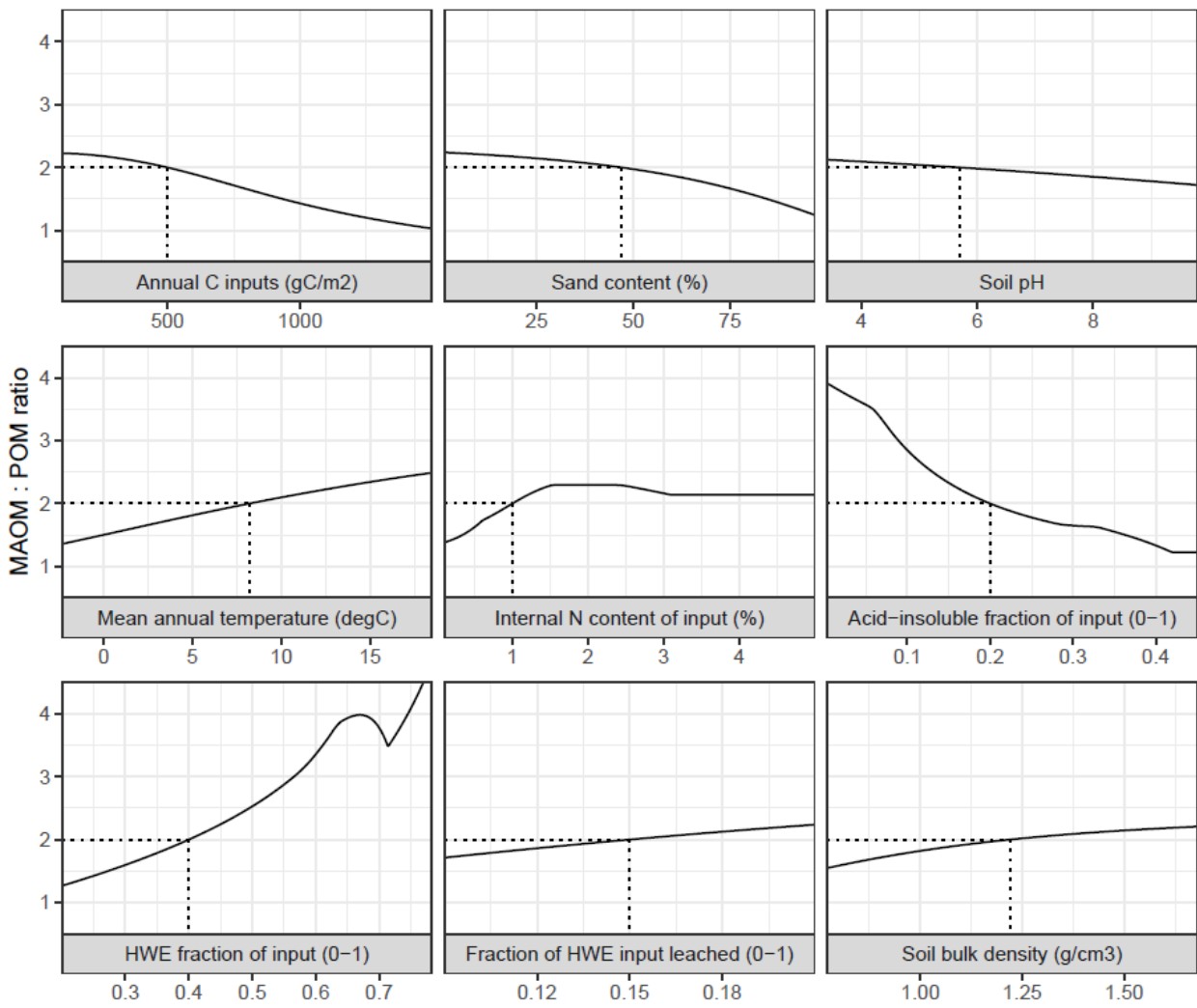

**Figure 4 - Mineral-associated organic matter (MAOM) stock response to different levels of input litter quality and quantity, compared for edaphic conditions which equate to different MAOM sorption relationships in MEMS v1.0. Formatting adopted from Castellano *et al.* (2015) to aid comparison between the hypothetical relationship postulated and the actual response simulated by MEMS v1.0 here.**

**Figure 5 - Measured and modelled soil C stocks (split into mineral-associated organic matter, MAOM, total particulate**
**organic matter, POM, and total soil organic carbon, SOC) for the forest and grassland land-use classes of the fractionated**
**sites from the LUCAS dataset ($n = 154$). Note that the MAOM:POM ratio facet is unitless, not as shown by the y-axis label.**
**Also note the free y-axis scales and that total POM is a sum of both light and heavy fractions.**

**Figure 6 - Comparisons between average (± 1 standard error) measured (red) and modelled (blue) bulk SOC stocks for 8192 forestry and grassland sites over a climatic and edaphic gradient across Europe. Each comparison is partitioned into high and low groups of mean annual precipitation, MAP (top *vs* bottom panels), mean annual temperature, MAT (left *vs* right panels) and soil texture (alternating panels left to right). ANOVA comparisons of means is performed to show significant differences (\*\*\* p < 0.001, \*\* p < 0.01, \* p < 0.05). Number of samples for each land use and division is shown at the base of each bar.**

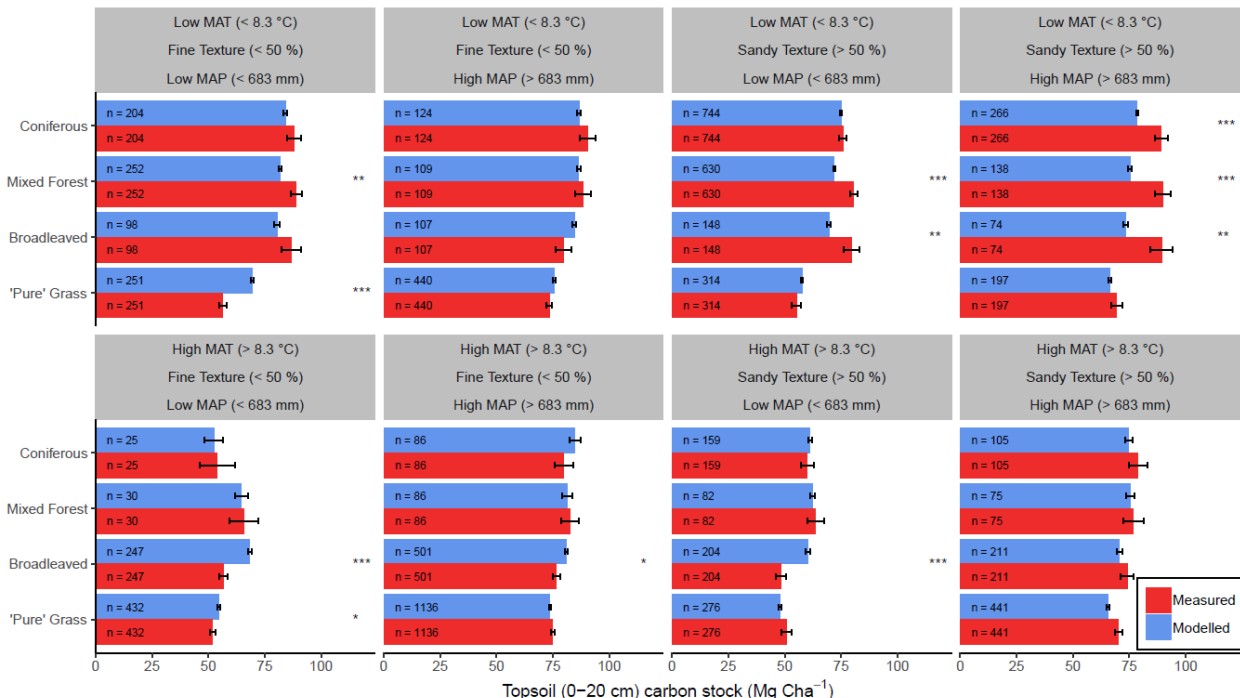

**Figure 7 - Model residuals of topsoil (0-20 cm) C stocks (Mg C ha[-1]) for 8192 sites (3487 grasslands, 1713 coniferous forests,**
**1590 broadleaved forests and 1402 'mixed' forests) across Europe, comparing measured values from the LUCAS database**
**(Toth *et al.*, 2013) to simulated steady-state estimates from the MEMS v1.0 model. All land uses are grouped for averages.**
**Residuals are averaged across all sites within each NUTS2 region (populations between 800,000 and 3 million) and coloured**
**accordingly. Measured site C stocks were subtracted from modelled values, meaning the model underestimates SOC stocks**
**in positive (blue) regions and overestimates SOC stocks in negative (red) regions. Residuals average to within 10 Mg C ha-**
**[1] in areas with the lightest yellow colour. The size of circles within each region represents the number of sites simulated.**
**Grey regions included no sites.**

## All Land-uses

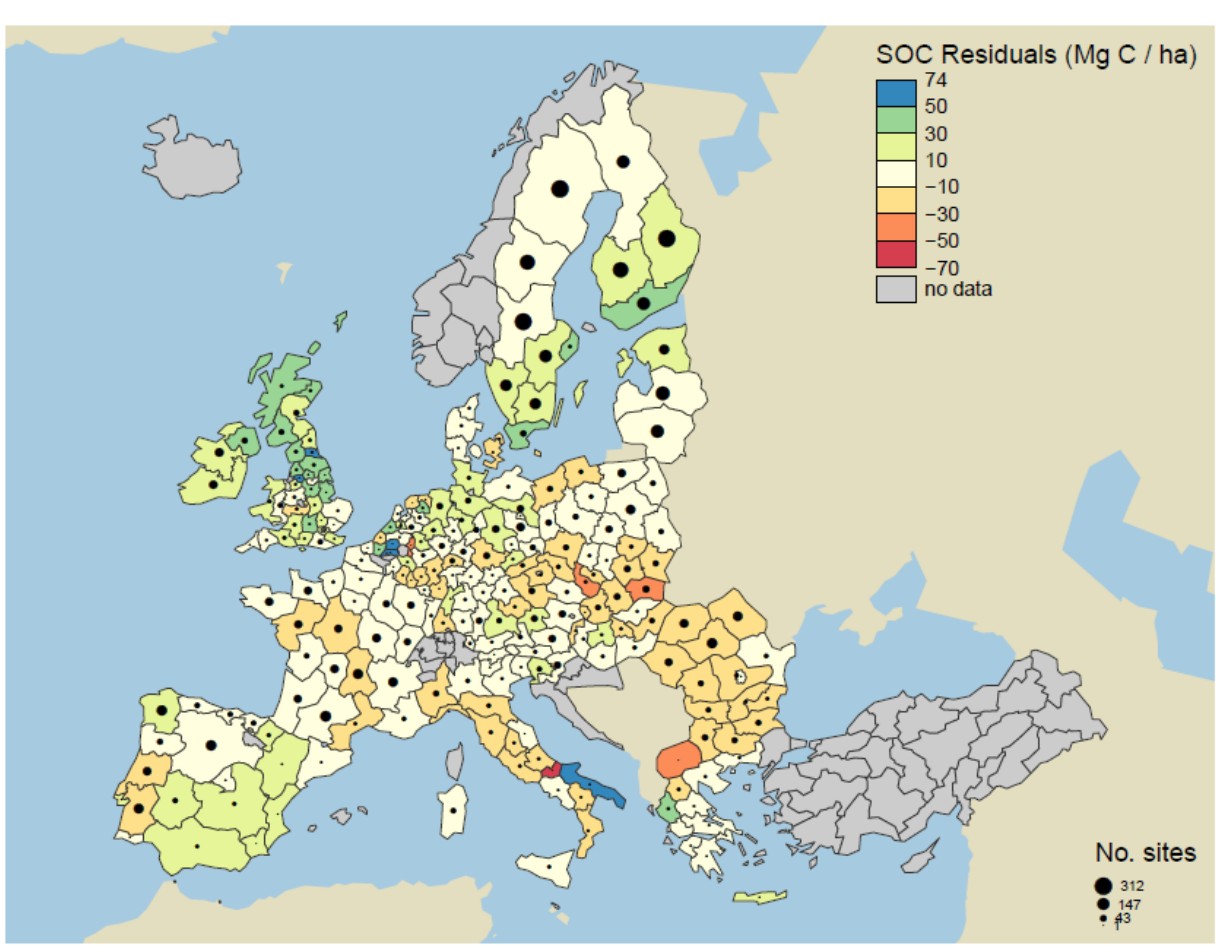


 **Tables**

**Table 1 - State variables of MEMS v1.0 and fractionation definitions (measurement proxy and protocol) for isolating each**
**pool. C1 to C4, and C6, refer to the litter layer, while C5 and C8 to C10 refer to the mineral soil. POM, Particulate organic**
**matter; DOM, Dissolved organic matter; OM, Organic Matter. All SOM fractions are primary fractions obtained after**
**dispersion to break up aggregates. For detail on a fractionation scheme to quantify each pool of the MEMS model please**
**refer Table S1.**

| State variable | Pool description | Measurement proxy | Method reference |
|---|---|---|---|
| C1 | Water soluble litter | Hot-water extractable C | Tappi (1981) |
| C2 | Acid-soluble litter | Hydrolyzable fraction | Van Soest and Wine (1968); Van |
| C3 | Acid-insoluble litter | Unhydrolyzable fraction | Soest *et al.* (1991) |
| C4 | Microbial biomass | Direct extraction | Various (e.g., Setia *et al.*, 2012) |
| C5 | Coarse, heavy POM | $> 1.8$ g cm$^{-3}$ and $> 53$ μm C | Christensen, 1992 |
| C6 | Litter layer DOM | $< 0.45$ μm extractable C | Kolka *et al.*, 2008 |
| C7 | Emitted $CO_2$ | Heterotrophic soil respiration | See Subke *et al.*, 2006 |
| C8 | Soil layer DOM | $< 0.45$ μm extractable C | Kolka *et al.*, 2008 |
| C9 | Mineral-associated OM | $> 1.8$ g cm$^{-3}$ and $< 53$ μm C | Christensen, 1992 |
| C10 | Light POM | $< 1.8$ g cm$^{-3}$ | Christensen, 1992 |
| C11 | Leached DOM | Suction cups / pans etc. | See Kindler *et al.*, 2011 |


**Table 2 - Description and default values of all parameters used with MEMS v1.0. Where possible, notation has been used to remain consistent with further details in the supplementary information. Driving variables are reported in Table 3. Ranges are indicative of those observed in literature. Refer to Materials and Methods and Table S2 for details of the optimized parameter ranges.**

| Parameter | Parameter definition | Default value (range) | Units | Reference(s) |
|---|---|---|---|---|
| $B1$ | Maximum growth efficiency of microbial use of water-soluble litter carbon (C1) | 0.6 (0.4 – 0.7) | g microbial biomass C/g decayed | Sinsabaugh *et al.*, 2013 |
| $B2$ | Maximum growth efficiency of microbial use of acid-soluble structural litter carbon (C2) | 0.5 (0.3 – 0.6) | g microbial biomass C/g decayed | Sinsabaugh *et al.*, 2013 |
| $B3$ | Heavy, coarse particulate organic matter (C5) generation from microbial biomass carbon (C4) decay | 0.33 (0.028 – 0.79) | g microbial products C/g decayed C | Campbell *et al.*, 2016 |
| $LIT_{frg}$ | Carbon in structural litter inputs (C2 and C3) transported to soil particulate organic matter (C5 and C10) each time step | 0.006 ($1 \cdot 10^{-5} – 2 \cdot 10^{-3}$) | g C/g C decayed | - |
| $POM_{split}$ | Fraction of fragmented litter inputs that form heavy particulate organic matter (C5) | 0.30 (0.07 – 0.83) | 0-1 scaling | Poeplau and Don, 2013; Soong *et al.*, 2016 |
| $DOC_{frg}$ | Carbon in litter layer DOM (C6) transported to soil DOM (C8) each time step | 0.8 (0.2 – 0.99) | g DOM-C/g DOM-C | - |
| $DOC_{lch}$ | Maximum specific rate of leaching to represent vertical transport of carbon in DOM through the soil profile | 0.00438 ($1 \cdot 10^{-5} – 0.02$) | g C day$^{-1}$ | Trumbore *et al.* 1992 |
| $EH_{max}$ | Maximum amount of carbon leached from decayed acid-soluble litter carbon (C2) to litter layer DOM (C6) | 0.15 | g DOM-C/g decayed C | Campbell *et al.*, 2016 |

| | | | | |
|---|---|---|---|---|
| $EH_{min}$ | Minimum amount of carbon leached from decayed acid-soluble litter carbon (C2) to litter layer DOM (C6) | 0.005 | g DOM-C/g decayed C | Campbell *et al*., 2016 |
| $ES_{max}$ | Maximum amount of carbon leached from decayed water-soluble litter carbon (C1) to litter layer DOM (C6) | 0.15 | g DOM-C g decayed C$^{-1}$ | Campbell *et al*., 2016 |
| $ES_{min}$ | Minimum amount of carbon leached from decayed water-soluble litter carbon (C1) to litter layer DOM (C6) | 0.005 | g DOM-C g decayed C$^{-1}$ | Campbell *et al*., 2016 |
| $k_1$ | Maximum decay rate of water-soluble litter carbon (C1) | 0.37 (0.16 – 0.70) | day$^{-1}$ | Campbell *et al*., 2016 |
| $k_2$ | Maximum decay rate of acid-soluble litter carbon (C2) | 0.009 (0.0011–0.0200) | day$^{-1}$ | Campbell *et al*., 2016 |
| $k_3$ * | Maximum decay rate of acid-insoluble litter carbon (C3) | 0.0002 ($2 \cdot 10^{-5}$– $1 \cdot 10^{-3}$) | day$^{-1}$ | Moorhead *et al*., 2013 |
| $k_4$ | Maximum decay rate of microbial biomass carbon (C4) | 0.57 (0.11-0.97) | day$^{-1}$ | Campbell *et al*., 2016 |
| $k_5$ | Maximum decay rate of heavy, coarse particulate soil organic matter (C5) | 0.0005 ($6 \cdot 10^{-5}$– $1 \cdot 10^{-3}$) | day$^{-1}$ | Campbell *et al*., 2016; Del Galdo *et al*., 2003 |
| $k_8$ | Maximum decay rate of soil DOM (C8) | 0.00144 | day$^{-1}$ | Kalbitz *et al*., 2005 |
| $k_9$ | Maximum decay rate of mineral-associated soil organic matter (C9) | $2.2 \cdot 10^{-5}$ ($1 \cdot 10^{-5}$– $4 \cdot 10^{-5}$) | day$^{-1}$ | Del Galdo *et al*., 2003 |
| $k_{10}$ | Maximum decay rate of light particulate soil organic matter (C10) | $2.96 \cdot 10^{-4}$ ($4 \cdot 10^{-3}$–$1 \cdot 10^{-4}$) | day$^{-1}$ | Del Galdo *et al*., 2003 |
| $la_2$ | Carbon leached from decayed microbial biomass carbon (C4) | 0.19 (0.022 – 0.42) | g DOM-C g decayed C$^{-1}$ | Campbell *et al*., 2016 |

| | | | | |
|---|---|---|---|---|
| $la_3$ | Carbon leached from acid-insoluble litter carbon and heavy, coarse particulate organic matter carbon (C3 and C5) | 0.038 (0.014 – 0.050) | g DOM-C g decayed C$^{-1}$ | Campbell *et al.*, 2016; Soong *et al.*, 2015 |
| $LCI_{max}$ | Maximum lignocellulosic index that influences DOM generation from litter decay | 0.51 | - | Campbell *et al.*, 2016; Soong *et al.*, 2015 |
| $N_{max}$ | Maximum N content that influences rates (above this, there is no limit) of DOM generation and microbial carbon assimilation | 3 | % | Sinsabaugh *et al.*, 2013 |
| $N_{mid}$ | Mid-point of logistic function that describes N limitation | 1.75 | % | Campbell *et al.*, 2016; Soong *et al.*, 2015 |
| $T_{opt}$ | Optimum temperature at which decay rates are highest | 45 | °C | Harmon and Domingo, 2001 |
| $T_{Q10}$ | Rate at which the decomposition rate increases with a 10 °C increase in soil temperature | 2 | - | Harmon and Domingo, 2001 |
| $T_{ref}$ | The reference temperature of estimated maximum decay rates (i.e., parameters $k_x$) | 13.5 | °C | Del Galdo *et al.*, 2003 |
| $T_{shp}$ | Shape of the excessive temperature limitation for temperature modifier on decay rates beyond optimum temperature | 15 | - | Harmon and Domingo, 2001 |
| $T_{lag}$ | Difference from optimum temperature to the decline above that threshold applying to the temperature modifier on decay rates | 4 | °C | Harmon and Domingo, 2001 |
| $T_{range}$ | Difference between the maximum and minimum soil temperature values over a given year (*unused when temperature inputs are available*) | 24 | °C | Toth *et al.*, 2013 |
| $SC_{icept}$ | Intercept coefficient used for the linear regression that estimates the maximum sorption capacity (parameter $Q_{max}$) of a soil | 11.08 | g C in < 53 μm fraction kg soil$^{-1}$ | Six *et al.*, 2002 |

| | | | | |
|---|---|---|---|---|
| $SC_{slope}$ | Slope coefficient used for the linear regression that estimates the maximum sorption capacity (parameter $Q_{max}$) of a soil | 0.2613 | - | Six *et al.*, 2002 |
| $^{L}k_{lm}$ * | Binding affinity for carbon in soil DOM (C8) sorption to mineral surfaces (C9) of the soil layer $L$ | 0.25 | gC day$^{-1}$ | Mayes *et al.*, 2012; Abramoff *et al.*, 2017 |
| $^{L}Q_{max}$ * | Maximum sorption capacity of mineral-associated soil organic matter carbon (C9) of soil layer $L$ | - | gC m$^{-2}$ depth$^{-1}$ | Six *et al.*, 2002 |

* These parameters are calculated as functions of others. For example, $Q_{max}$ is a function of sand content, soil bulk density, rock fraction, $SC_{icept}$ and $SC_{slope}$. More details can be found in the supplementary materials.

**Table 3 - List of required driving variables for the MEMS v1.0 model. Baseline values represent mean values as reported in the LUCAS database (Toth *et al.*, 2013) of 8192 forest and grassland sites across Europe and were used for all qualitative testing and sensitivity analyses.**

| Driving variable | Symbol | Units | Baseline value | Land-use specific values | | | | Reference |
|---|---|---|---|---|---|---|---|---|
| | | | | Grass land | Broadleaf forest | Mixed forest | Coniferous forest | |
| *Site condition variables* | | | | | | | | |
| Annual net primary productivity | *annNPP* | g C m$^{-2}$ yr$^{-1}$ | 681 | Site-specific values required | | | | ORNL DAAC, 2009 |
| Sand content of soil layer | *Sand* | % | 47.8 | | | | | |
| Bulk density of soil layer | *BD* | g cm$^{-3}$ | 1.21 | | | | | Toth *et al.*, 2013 |
| Rock fraction of soil layer | *Rock* | % | 7.62 | | | | | |
| Soil pH of layer | *pH* | - | 5.58 | | | | | |
| * Daily total carbon input | *CT* | g C m$^{-2}$ day$^{-1}$ | 1.30 | | | | | - |
| * Mean daily soil temperature | *soilT* | ºC | 8.28 | | | | | NOAA, 2018 |
| *Litter chemistry variables* | | | | | | | | |
| Hot-water extractable fraction | *fSOL* | 0-1 | 0.45 | 0.35 | 0.40 | 0.38 | 0.35 | |
| Acid-insoluble fraction | *fLIG* | 0-1 | 0.20 | 0.15 | 0.27 | 0.30 | 0.32 | Campbell *et al.*, 2016 |
| Internal nitrogen content | *LitN* | % | 1.00 | 1.10 | 1.32 | 0.87 | 0.41 | |
| *Root distribution variables* | | | | | | | | |
| Maximum rooting depth | *Rdepmx* | cm | 300 | 260 | 290 | 340 | 390 | Canadell *et al.*, 1996 |
| Depth to which 50% of root mass is distributed | *Rdep50* | cm | 20 | 15 | 25 | 27.5 | 30 | Jackson *et al.*, 1996 |
| Root to shoot ratio | *RtoS* | - | 1.00 | 3.70 | 0.23 | 0.21 | 0.18 | Jackson *et al.*, 1996 |

5 * - When daily measurements are not available annual values can be used to interpolate daily estimates. For more information please refer to the supplementary materials.

**Table 4 - Evaluation results of comparisons between measured and modelled topsoil (0-20 cm) C stock for 8192 grassland and forest sites across Europe (see Figure 7 for geographic distribution of residuals). Mean absolute error (MAE) and mean bias error (MBE) describe the overall difference and directional difference between measured and modelled values, respectively. The model is deemed to describe the trend of the measured data better than the mean of the measurements when the modelling efficiency (EF) is positive, or when the Coefficient of Determination (CofD) is above 1. Each is a discrete evaluation metric. Divisions of high/low site conditions (mean annual temperature, mean annual precipitation, annual C inputs, sand content) were used to derive statistical significance (root mean square error, RMSE, and *F*-statistic) of differences between measured and modelled values while accounting for measurement variance within these divisions. An RMSE value below RMSE$_{95}$ indicates that simulated C stocks fall within the 95 % confidence interval of the measurements. An F-statistic below 0.05 also shows that simulated values are not significantly different to measurements at a 95 % confidence level.**

| Land use | *n* | Mean ± 1 S.E. (Mg C ha$^{-1}$) | | MAE (Mg C ha$^{-1}$) | MBE (Mg C ha$^{-1}$) | EF | CofD | RMSE (Mg C ha$^{-1}$) | RMSE$_{95}$ (Mg C ha$^{-1}$) | *F*-statistic |
|---|---|---|---|---|---|---|---|---|---|---|
| | | Observed | Predicted | | | | | | | |
| Pure grass | 3487 | 65.9 ± 0.5 | 66.3 ± 0.3 | 24.7 | -0.4 | -0.047 | 4.52 | 13.0 | 10.3 | 0.009 |
| Broadleaved | 1590 | 71.2 ± 1.0 | 73.8 ± 0.4 | 31.0 | -2.5 | -0.062 | 5.54 | 19.0 | 14.7 | 0.052 |
| Mixed Forest | 1402 | 82.3 ± 1.1 | 75.2 ± 0.3 | 35.4 | 7 | -0.173 | 8.36 | 12.9 | 19.2 | 0.042 |
| Coniferous | 1713 | 79.0 ± 1.1 | 76.3 ± 0.3 | 36.1 | 2.7 | -0.057 | 10.35 | 13.5 | 18.7 | 0.006 |
| * All | 8192 | 72.5 ± 0.4 | 71.4 ± 0.2 | 30.2 | 1.1 | -0.048 | 6.32 | 14.9 | 15.7 | 0.020 |

Evaluation metrics for individual site performance / Evaluation metrics using site condition *divisions* to include variance

* All sites use 64 divisions (high/low site conditions and land use type)

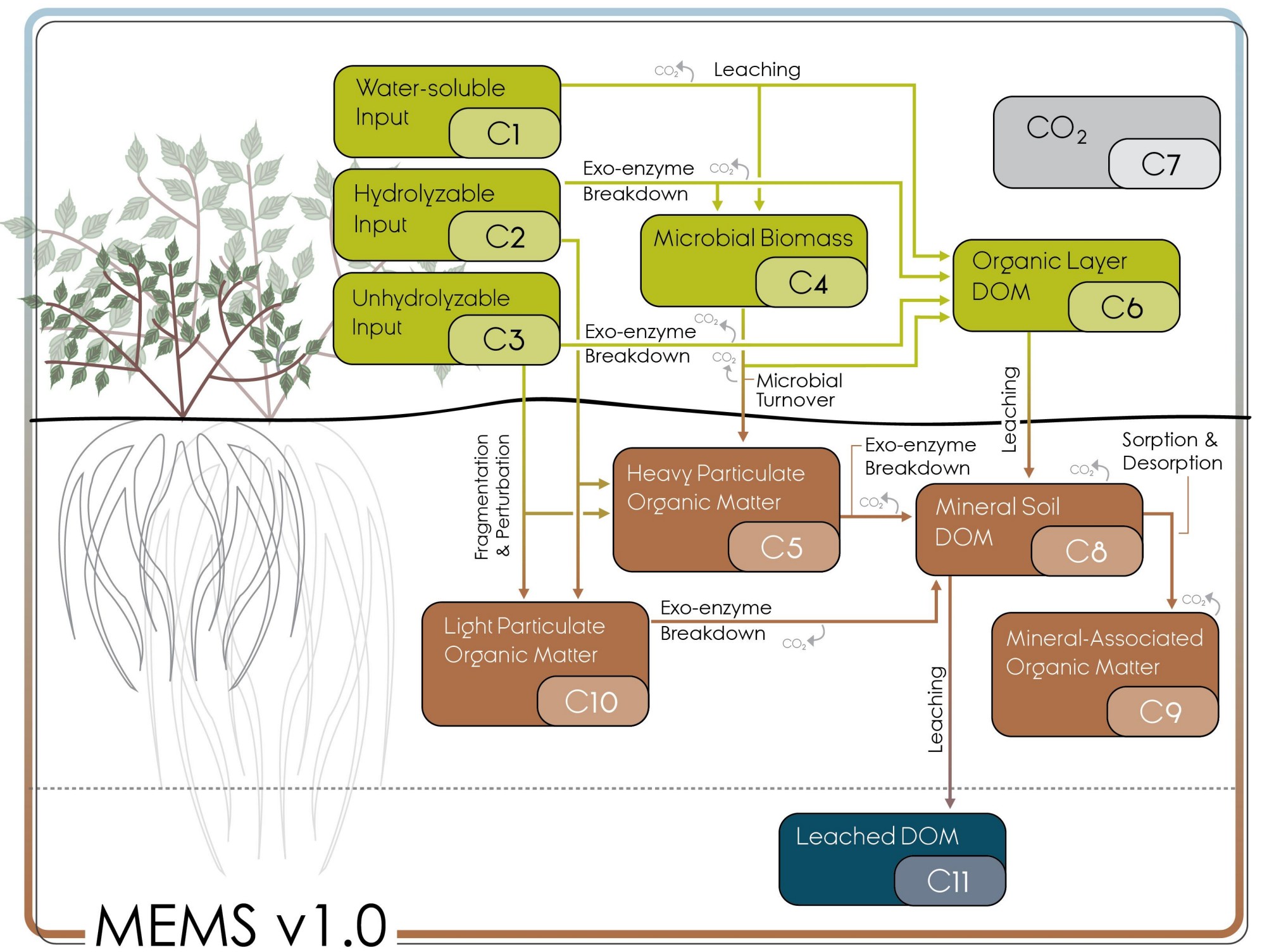

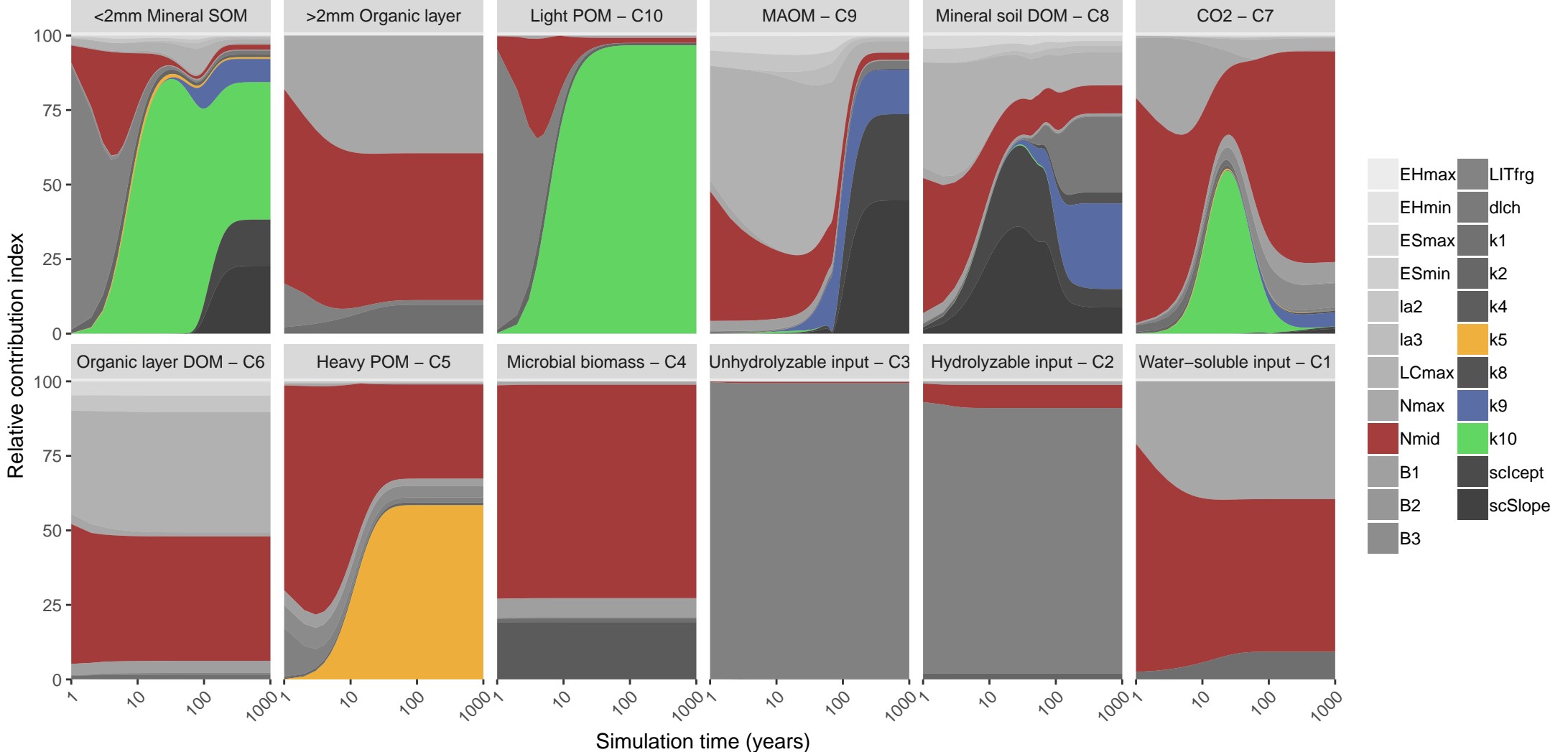

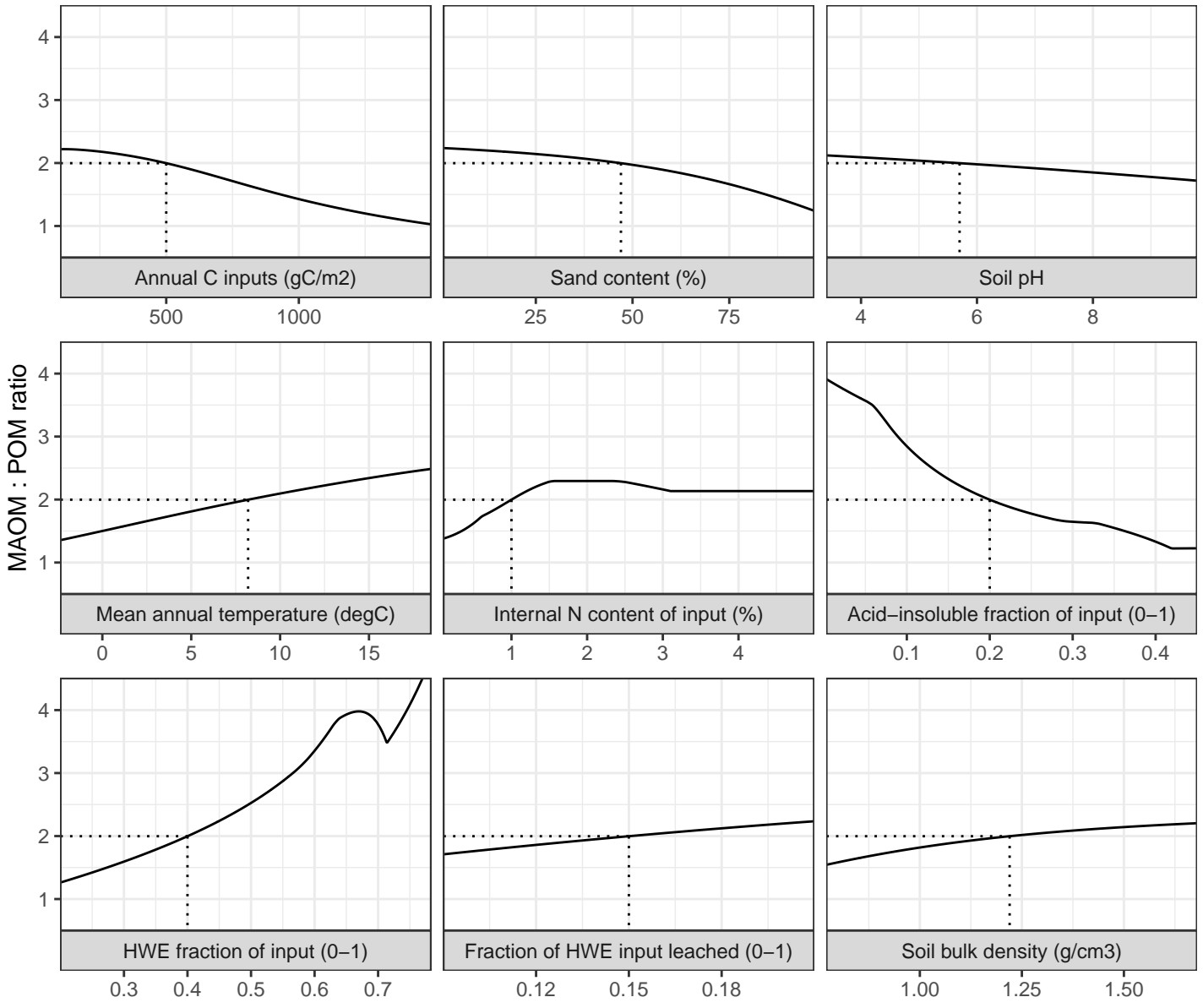

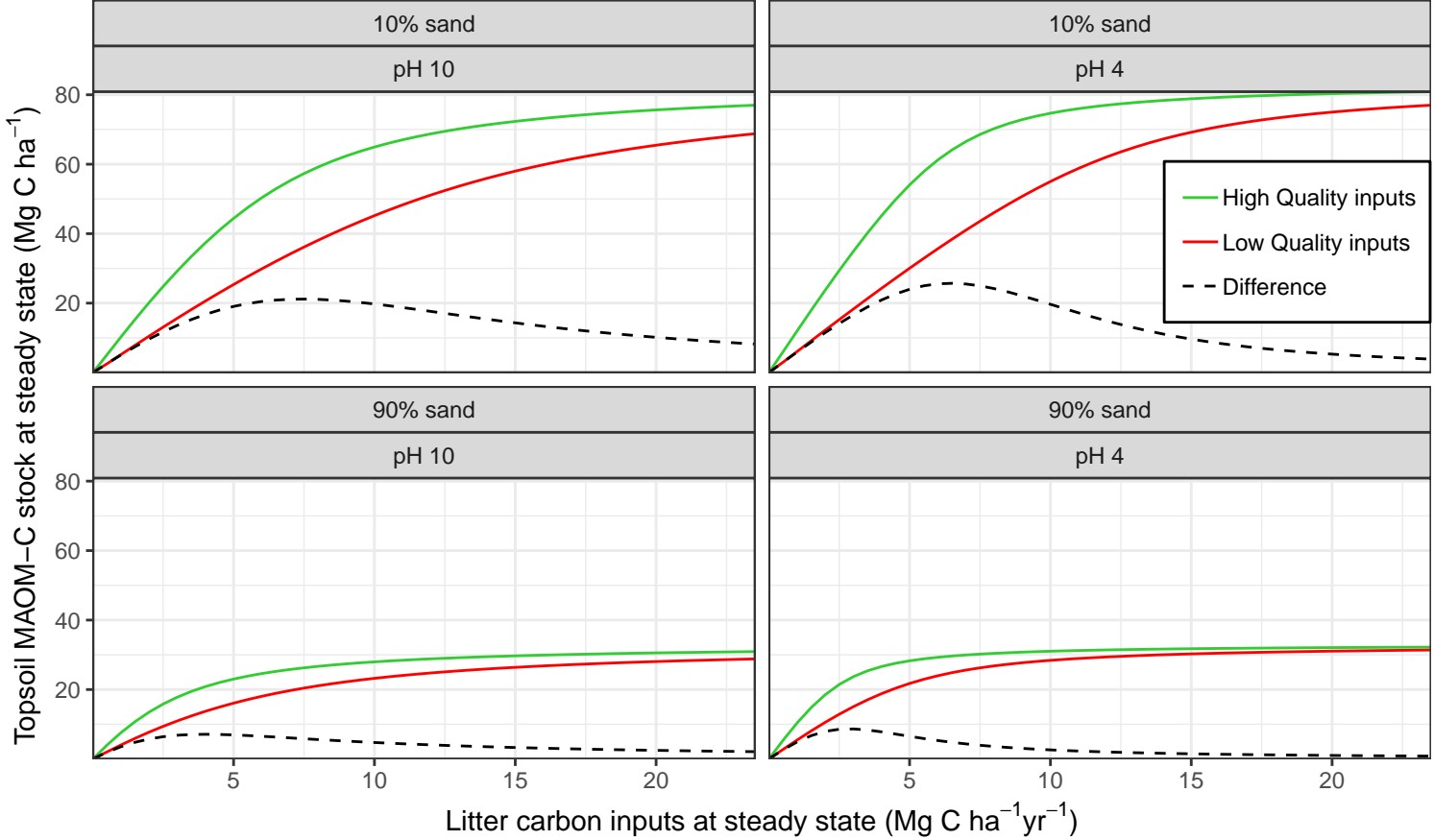

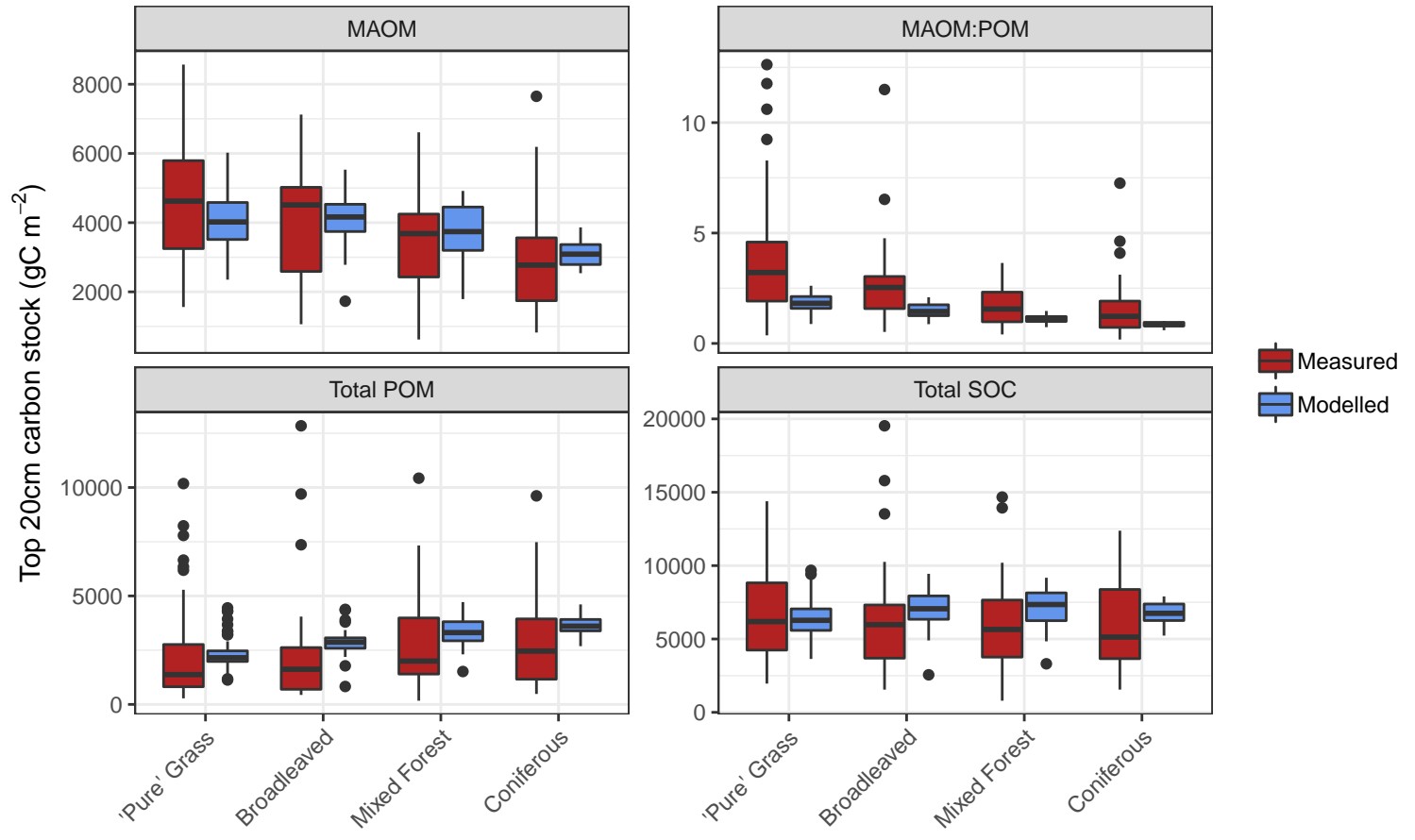

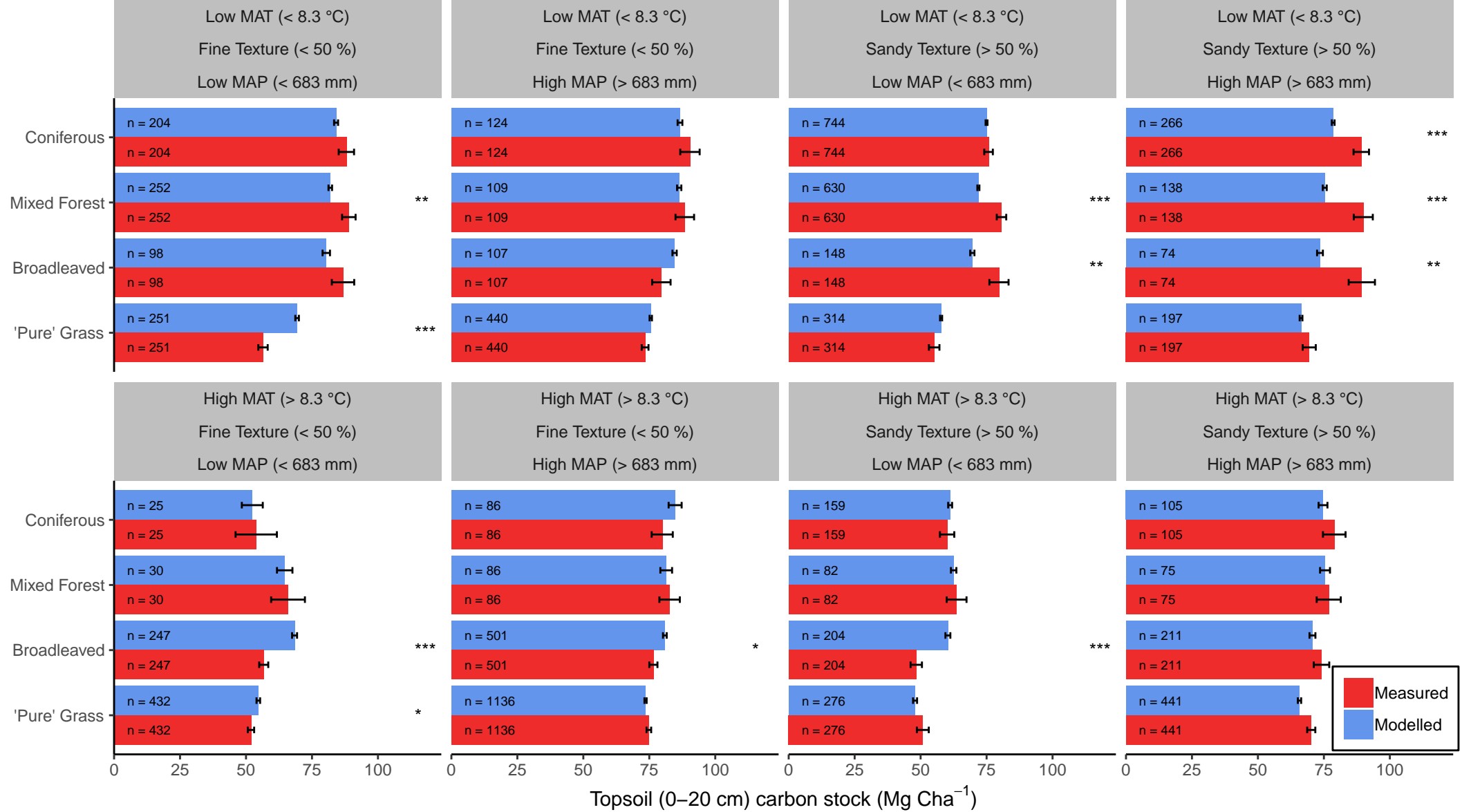

# All Land−uses

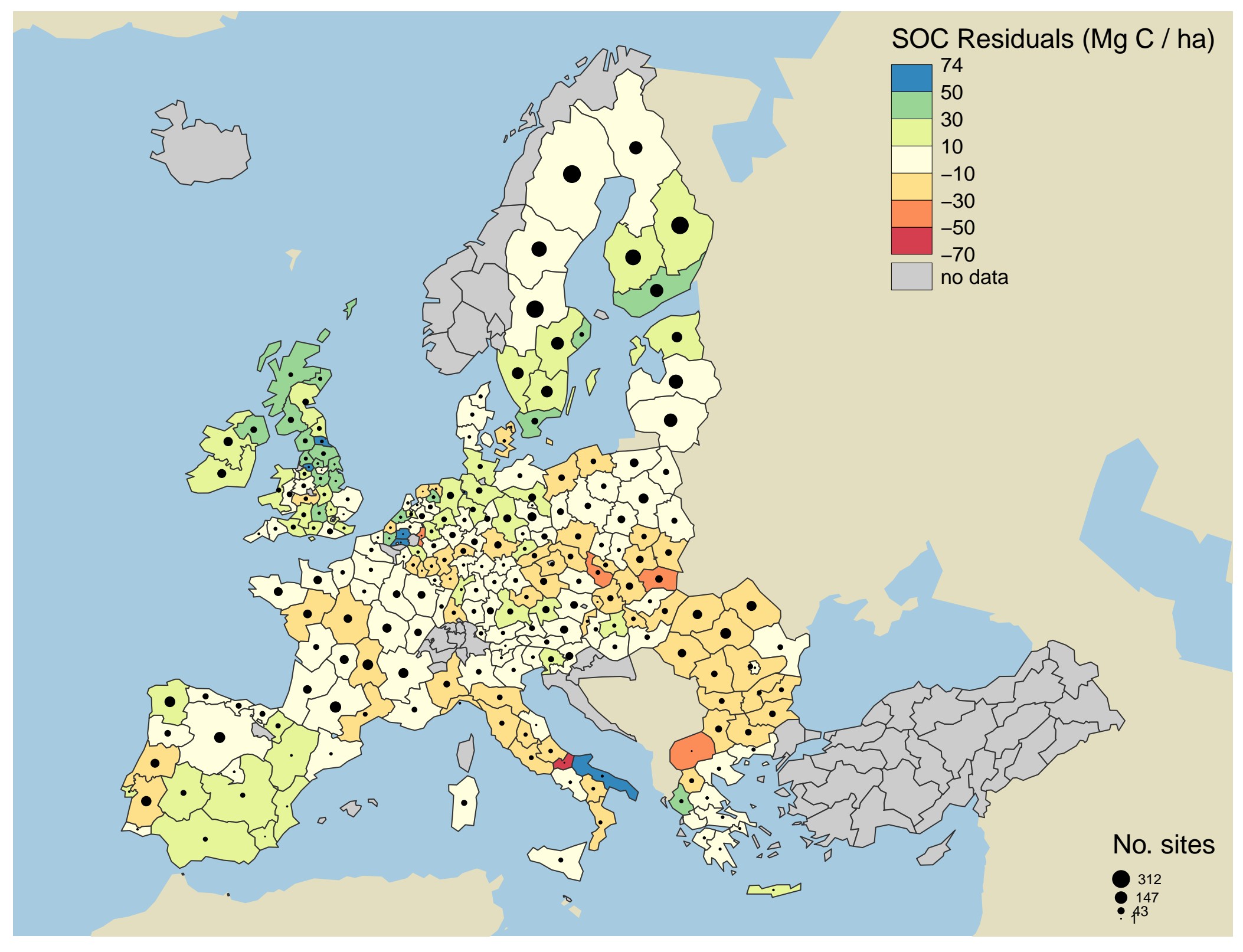