# Peer review of "Unifying soil organic matter formation and persistence"

_Biogeosciences, 2018_

## Referee Comment (RC1) · T. Wutzler (Referee) · 6 Nov 2018

The study of Robertson et al. present a first version of the MEMS model, a parsimonious dynamical model of soil organic carbon (SOC) development at ecosystem scale, together with a validation across a many sites. I enjoyed reading the manuscript, which is well structured and succeeds in getting the fundamental ideas across in a concise way and provides the details in the appendix.

The proposed model is of similar complexity as classical pool-based models but better incorporates recent mechanistic understanding and is better comparable to measurable pools. Hence, it is of great interest to soil science, ecosystem research, and potentially also global change communities. It adds a complementary alternative in the

suite of simple to much more detailed SOC models and the study should be published after revisions.

I liked the approach of directly modeling relevant quantities at the scale model purpose, the management scale. I liked the simulation time dependent sensitivity analysis, although Fig. 2 is hard to read.

The supplementary is complicated by already anticipating several mineral soil layers and sometimes is inconsistent with the main text. For example, there is explicit microbial assimilation in mineral layers in the supplementary, but the main text states that microbes are implicit there. Please, provide a version that matches the main text and the presented model structure.

The wording of "litter layer" and "mineral soil" are used in a fuzzy way. Also it did not became clear to me, how model-data comparison dealt with organic layers, which are neither part of the litter (in the model lacking particulate organic matter (POM) pools) nor the mineral soil (in the model simulating sorption to minerals). Maybe this partly causes the large model-data discrepancies for broadleaved sites with large POM stocks.

I have several more detailed comments or questions that are intended to help with clarification and setting the model into context.

Detailed comments for model structure:

Could you, please, elaborate a bit more why you (as well as the LIDEL model) choose microbes to not consume DOM?

In the LIDEL model there is a C5 microbial products pool also in the litter layer, why do you assume in MEMS that all microbial turnover is transferred to the mineral soil?

You choose decomposition to be independent of the size of biomass pool to avoid some problematic feedback. Then I suggest to simplify the model even more by replacing microbial biomass turnover by the sum of inputs to the biomass pool. Then you do not

need to simulate this pool, save one state variable and several model parameters. If microbial biomass is required for data comparison, you can still compute it assuming near steady state with inputs (e.g. Wutzler 2013).

Detailed comments for model-data integration:

It should be clarified better also in the discussion, that the model performance was judged by comparing steady state MAOM pools to observations. I am still looking forward to a comparison where a model successfully simulated dynamics compared to observed changes decadal stock changes across many sites.

L376: averaging parameters is dangerous, because of nonlinearities. I suggest to use only one non-averaged parameter combination. You may pick the fold randomly.

L376: You can avoid the choice of one criterion among three data streams of MAOM-C, POM-C and bulk SOC by using a cost function based on the sum of squared residuals of all the data streams.

Fig 5: The classification to land use not particularly helpful, because variables are very similar with a high range across these classes, including the mentioned significant different of MAOM:POM (L 485). Furthermore, plotting the distribution of observations and distribution of predictions separately does not help to judge model performance (L488).

I suggest instead inspecting and plotting the distribution of model-data residuals of several variables and relating these differences to classes and other environmental conditions. This would indicate which variables and processes are most urgent to extend MEMS v1, as done with the discussing Fig 7.

Other detailed comments main text

Text in Figs 2 and 6 are hard to read. Can you provide a vector graphics of this figure? There are too many classes to distinguish by color, but I have no suggestion how to improve.

Fig 4: Suddenly, pH is popping up, but was never introduced as a driving variable. I suggest to shortly state that sorption rate is pH dependent, and refer to the eq. 35 in the Appendix.

L 529: These are interesting effect of N in a C-only based model. While the microbially detailed models of Perveen 2014 and Wutzler 2017 attribute low litter N effects to N mining in older usually N-rich pool and accumulation of less processed material, MEMS attributes this to reduced microbial accessibility and reduced DOM production. Do you think that chemical and stoichiometric effects are two sides of the same coin, or are these competing hypotheses? I am looking forward to the version that explicitly simulates N fluxes.

Other detailed comments appendix

To me its difficult to always keep a list of meaning of pool 1 to 10 in my head. Could you come up with more expressive pool names?

L 70: I do no find more information on ub and uk in Table 2 in the main text. I suggest referring to eqs. 19-22.

L 110-112: The long sentence did not became clear to me. Is L_j_C5_C4_gen really a combined flux of bioturbation, ..., and DOC leaching? I thought the latter one is covered by eq. 33.

eq 47: k8 does not match the text before that states k5.

Thanks for this work. I suspect MEMS to be included in further model comparisons as a complementary model.

References

Perveen N, Barot S, Alvarez G, Klumpp K, Martin R, Rapaport A, Herfurth D, Louault F & Fontaine S (2014) Priming effect and microbial diversity in ecosystem functioning and response to global change: a modeling approach using the SYMPHONY model.

Global Change Biology, Wiley-Blackwell, 20, 1174 - 1190 10.1111/gcb.12493

Wutzler T & Reichstein M (2013) Priming and substrate quality interactions in soil organic matter models. Biogeosciences, 10, 2089-2103 10.5194/bg-10-2089-2013

Wutzler T, Sönke Zaehle, Schrumpf M, Ahrens B & Reichstein M (2017) Adaptation of microbial resource allocation affects modelled long term soil organic matter and nutrient cycling. Soil Biology and Biochemistry, Elsevier BV, 115, 322-336 10.1016/j.soilbio.2017.08.031

---

## Referee Comment (RC2) · Anonymous Referee #2 · 12 Dec 2018

Overall Review

The authors present a new soil biogeochemistry model, MEMS v1.0, that explicitly represents biochemical complexity of litter pools, microbial biomass, mineral associated organic matter and particulate organic matter. The model has the capability of including variable CUE in litter decomposition and mechanisms leading to SOM stabilization and saturation of mineral associated carbon fraction. Four key model parameters are calibrated to reproduce soil fractionation observations of mineral associated and particulate organic matter fractions and the model is evaluated in reproducing topsoil SOC in more than 8000 sites across different land-uses in Europe with satisfactorily results.

Constructing models that are based on measurable carbon pools rather than on the old framework assigning turnover rates to a given number of unmeasurable carbon pools is

a very important endeavor and the authors are definitely moving beyond conventional SOC modeling. It is especially important to have models that link litter decomposition processes and SOM formation processes, which is rarely the case, as stated by the authors (L 89-91). I am very much in favor of such a type of approach and supportive of the author's effort. The manuscript is very well written and clearly presented and the introduction frames very well the problem.

I would be happy to have a few clarifications on some technical aspects and about one important assumption related to the role of the microbial pool. These are written in a number of minor comments that hopefully can be addressed.

I would also invite the authors to tone down the role of MEMS v1.0 as "ecosystem model", since the current version is still far from being there. As a matter of fact, in several instances (e.g., Line 606) the authors state that the model is incomplete (e.g., lack of hydrological and nutrient cycle) and that these deficiencies will be addressed in future model developments. The model represents SOM dynamics at the "ecosystem-scale". However, for various reasons but especially because the temporal dynamics are not evaluated in this article, I would invite to use cautious statements in the link with ecosystem models. Only the steady-state conditions are tested. A correct representation of temporal dynamics is key for coupling with other models. At this stage, this is a quite significant limitation for application in ecosystem models. Furthermore, feedbacks between soil and vegetation cannot be considered.

Other simplifications are that NPP is prescribed from MODIS, the model does not account for temporal dynamics of soil moisture or for nutrient cycles, the root:shoot ratio is prescribed for various biomes. However, these are overall clearly described. I would also appreciate some additional discussion about the issue in comparing pools, which are spun up at the equilibrium with observed pools (Line 366-367). The authors are aware of the issue and they briefly discussed it. However, most of the description of the results and the calibration effort convey somehow the intention to match C-pools as closely as possible. Given the expected difference between actual SOC and "steadystate” SOC, I would have allowed more freedom to the model and focus on comparing patterns as in Fig. 5 and 6 rather than absolute quantities.

Despite these limitations, the manuscript is undoubtedly a novel contribution to the field and surely a step in the right direction.

Minor Comments

Line 75. It is cited later on, however, Wieder et al 2015 would fit well also here.

Line 96. Maybe one sentence with additional explanations for K vs r strategies (e.g., copiotrophic and oligotrophic microbial functional groups) is necessary, not all the “modelers” may be aware of these concepts.

Line 113-114. The issue related to the lack of inputs or information to derive model parameters and validate model responses, of course, is a very important one and may compromise practicality as written by the authors. However, modeling efforts in the direction of more mechanistic representations of the soil system can shed light on the importance of processes and interactions that were not accounted or quantified before, they may provide guesses for the magnitude of certain pools/fluxes and may motivate the collection of those data that are necessary to testmechanistic predictions. In other words, they can have a value in process explanation rather than a predictive value.

Line 174-178. In a certain way, also the CENTURY model, especially in more updated versions (e.g., Kirschbaum and Paul, 2002) accounts for nitrogen and lignin content of the litter, which are affecting the turnover rates of the various litter pools. Additionally, their subdivision in metabolic and structural litter pools is not far from the subdivision in the pools C1, C2, C3. This may be acknowledged in the manuscript or if major differences, which I cannot recognize, do exist, they need to be remarked.

Line 189-190. The assumption of considering a microbial pool (C4) for the litter component is probably the decision in terms of model construction, which leaves me more bewildered. This pool, presumably, is mostly located aboveground, even though is not

stated explicitly, and does not have an explicit role in the turnover of soil organic matter. Now, if anything, I would have make the reverse choice. Because of accessibility constraints and relatively paucity of microbial biomass in the soil, the decomposition of SOM is likely controlled explicitly by microbial biomass, while the decomposition of litter, which is mostly located aboveground (especially for land covers different from grassland) and air exposed is unlikely limited by microbial biomass. Maybe, my understanding of the system is wrong, but it would be useful to have a clarification of the rationale of such an assumption and eventually of the potential consequences.

Line 200. Please explain better what do you mean “represents microbial metabolism processes implicitly”

Line 268-269. It could also be, simply, that microbial growth is stimulated and there are more microbes that can also degrade faster the chemically recalcitrant substrates. If I understood correctly, this is not an effect that can be captured by the model without an explicitly microbial pool acting on POM (C5, C10) and MAOM (C8) decomposition.

Line 270-273. Generally speaking, microbial respiration will be related to microbial activity and CUE. Being not considered microbial activity in the soil, it is not very clear without looking in detail at the Supp. Material how respiration is computed and which fraction of the decomposition is assumed to be. While you refer to CO2 efflux, “respiration” is never mentioned in the Supplementary Material, which is quite surprising.

Line 281. I would also add that pH controls are quite important. The authors are already well aware of this but neglecting soil moisture controls is a quite significant simplification.

Line 293. At this stage is not clear how NPP values are derived. Maybe, it is worth to state that this must be an external input to the model. This is actually what mostly separate a “soil organic matter model” from an “ecosystem model”.

Table 3. The text-box with “site-specific values required” applies to all the site condition

variables (e.g., from NPP to soil temperature). This is not clear from the current Table where site-specific values seem to refer to “rock fraction of soil layer” only. I would suggest to use some curly bracket to envelope all these variables.

Line 315-319. I am actually quite familiar with the global sensitivity analysis and I think I understood what the authors did. However, I am quite sure that the succinct explanation provided in these lines will remain unclear to most of the readers. I would suggest to either explaining it better (i.e., more extensively) or minimizing the explanation with a full discussion in the supplementary material.

Line 340. I know that this is probably the only option the authors had, but I hope they are well aware of the limitations of MODIS NPP product; maybe a sentence forewarning the reader would be necessary.

Line 345. The reference Cotrufo et al 2018 explaining the derivation of the POM and MAOM pools is not published. I guess for the sake of this article is fine, but of course, it would be a great contribution to the community if the values of POM and MAOM for the 154 sites would be provided as a part of the LUCAS database or somewhere as part of the article.

Line 368-369. This is probably more a philosophical than a practical point. However, I wonder if a rigorous numerical optimization for such type of models, where the model structure is very uncertain and difference between observed and simulated SOC could be related more to the initialization problem rather than to model structure or parameters is really needed. Given the fact that 4 parameters only were optimized and several replicates were made, this is probably an added value and unlikely a problem here, but still I wonder if is not giving too much weight to the data. How do the results look alike without optimization? This is briefly stated in Line 469-470 but it would actually be interesting to look at it in more detail.

Line 379. Maybe an explicit statement that optimized parameter values are reported in Table S2 would be useful here.

Line 386-387. How seasonal variability in C-inputs and temperature is accounted for? This is not very clear from the manuscript.

Line 407. The value for NPP and sand content differ from the mean value provided in Table 3.

Figure 2. What is the initial condition for the simulation of 1000 years depicted in Figure 2? Do you start from nearly steady state carbon pools or from carbon pools equal to "zero"?

Line 455. Why colder temperatures favor POM? Is this related to the sensitivity of decomposition?

Line 473-475. Table 2. Maybe I am missing something obvious but the units of decay parameters as "k1" to "k10" should be [gC gC-1 day-1], otherwise when multiplied by the pool (Eq. 1-11 in the supplementary material) you will get [gCˆ2 day-1] rather than [gC day-1].

Line 491. This is definitely expected given that variability in litter input, e.g., litter composition and stoichiometry root: shoot ratios are underestimated and soil moisture is not accounted for.

Line 496-497. For almost all of the analyzed sub-groups in terms of site-conditions of Figure 6, bulk SOC observations are mostly between 50-75 MgC/ha. I think this relatively narrow range complicates the identification of the control exerted by temperature, precipitation, soil texture or biomes and therefore also the model testing. A more reasonable test will require more distinguished values of SOC across different conditions, probably using other biomes and climates.

Line 521. I don't want to sound too pessimistic and overall I really like the approach of the authors but bridging the gap toward Ecosystem and Earth System Models still requires a considerable amount of work to test the reliability of temporal dynamics and plant-soil feedbacks. This should be stated in the manuscript.

Line 552. Also the dynamics of microbial pool in the soil is not explicitly simulated; however, the underestimation of variability is most likely due to underestimation of variability in the inputs and the steady-state assumption in the model, as you wrote in the next few lines.

Line 558-559. I am not sure why soil moisture controls should be so important at high-latitude, these sites are rarely water limited, I would expect lack of soil-moisture controls to be more important in South-Europe.

Line 621-622. This is a great point, and I am looking forward for further work of the authors along this line.

Figure 1. Just as a suggestion, up to the authors, it would be nice to have some of the parameters of Table 2 represented also in this plot to link the main fluxes to some of the key parameters regulating the flux.

References

Kirschbaum, M. U. F., and K. I. Paul (2002), Modelling C and N dynamics in forest soils with a modified version of the CENTURY model, Soil Biology & Biochemistry, 34, 341-354

Wieder, W. R., Allison, S. D., Davidson, et al, (2015). Explicitly representing soil microbial processes in Earth system models. Global Biogeochemical Cycles, 29(10), 1782-1800

---

## Author Comment (AC1) · 8 Jan 2019

Thank you for your detailed review. We have responded to all your comments in the attached supplement file. We highly recommend viewing our responses from the attached PDF that is contained within the zip supplement. This attached PDF file includes formatting that makes it much easier to track which comments refer to which points raised by yourself. Additionally, the attached zip supplement includes associated vectorized PDFs of the figures we refer to - these are much easier to interpret than those copied in with this plain-text script.

The supplement file should be available here - https://www.biogeosciences-discuss.net/bg-2018-430/bg-2018-430-AC1-supplement.zip

[Figure]

To satisfy journal submission guidelines we have also pasted our responses to this plain text form (below) but we still recommend using the attached zip supplement.

Many thanks again for your contribution to improving this work, we greatly appreciate your input. Andy Robertson and co-authors

— Herein we copy our responses in plain-text format from the attached supplement —

8th January 2019

Cover Letter and Responses to Reviewer Comments to accompany the manuscript: "Unifying soil organic matter formation and persistence frameworks: the MEMS model"

Authors: Andy Robertson, Keith Paustian, Stephen Ogle, Matthew Wallenstein, Emanuele Lugato, and Francesca Cotrufo

Thank you for your correspondence concerning our manuscript and for giving us the opportunity to resubmit a revised version. All comments from the reviewers have been carefully considered and appropriate responses are made below. Sincerely,

Andy Robertson

  Responses to comments from Thomas Wutzler on "Unifying soil organic matter formation and persistence frameworks: the MEMS model" by Andy D. Robertson et al. Reviewer comments in bold and our responses in normal text. Selected new text in the revised manuscript is pasted here in italics. Reference to the manuscript is given as new line number (L).

General comments

The study of Robertson et al. presents a first version of the MEMS model, a parsimonious dynamical model of soil organic carbon (SOC) development at ecosystem scale, together with a validation across many sites. I enjoyed reading the manuscript, which is well structured and succeeds in getting the fundamental ideas across in a concise way and provides the details in the appendix.

The proposed model is of similar complexity as classical pool-based models but better incorporates recent mechanistic understanding and is better comparable to measurable pools. Hence, it is of great interest to soil science, ecosystem research, and potentially also global change communities. It adds a complementary alternative in the suite of simple to much more detailed SOC models and the study should be published after revisions.

Many thanks for your comments and time spent reviewing our manuscript. We appreciate the detail and clarity of your suggested revisions – this certainly helps us to improve our manuscript. We are glad you enjoyed reading it and are excited to have an opportunity to publish the MEMS model. It is our hope that it can do just as you say and add to the suite of SOC models already available and stimulate discussion of how to advance this field.

Through the revisions described in detail below we hope to have addressed all your comments.

I liked the approach of directly modeling relevant quantities at the scale model purpose, the management scale. I liked the simulation time dependent sensitivity analysis, although Fig. 2 is hard to read.

I suspect part of the difficulty in reading the figure is because the submission guidelines are to embed the picture as a low-quality jpg. The original vectorized PDF is much clearer. However, we have also now hopefully made the figure easier to read by increasing the size of the text and limiting the colours to only the 4 most influential parameters. All other parameters are coloured in greyscale in order from top to bottom. The 'full colour' figure version is included as in a supplementary zipped file attached to these responses (Figure S5 is a lossless vector PDF for detailed inspection if the reader wishes). For your reference we show the new figure below (due to the odd submission process the figure may be repeated in higher quality at the end of this file).

SEE FIGURE 1 AT BOTTOM OF PAGE

The supplementary is complicated by already anticipating several mineral soil layers and sometimes is inconsistent with the main text. For example, there is explicit microbial assimilation in mineral layers in the supplementary, but the main text states that microbes are implicit there. Please, provide a version that matches the main text and the presented model structure.

We apologise for this confusion – the microbial assimilation, as a process, is indeed 'explicit' in that it is represented by fluxes into a microbial pool. However, in this inaugural version of the model the use of a microbial pool is more one to help differentiate the direct versus microbially-processed flux of organic matter inputs (pools C1-C3) to the soil C pools, sensu Liang et al., 2017. Once these inputs are added to the soil pools belowground, then the microbial biomass and associated metabolic processes are implicit (i.e., we assume there is microbial activity and mineralization of the carbon within these soil pools, but we do not represent these processes with discrete pools or fluxes). We certainly appreciate the comment because on review this is an important point that needed to be made clearer. At different points in the main manuscript, we have added additional points as to why we have a distinct microbial pool at the point of entry of the C input, but not after it is processed and transformed into the SOC pools, which have microbes within them.

A large part of the confusion likely resulted from the 'microbial assimilation from litter' section of the supplementary and we can indeed understand why. The use of the layer superscript certainly made our descriptions less clear. Consequently, we have also removed the superscript notations for soil layers that created unnecessary confusion in the supplementary model description. There should now be no inconsistency between the main text and the supplementary materials.

Liang, C., et al. (2017). "The importance of anabolism in microbial control over soil carbon storage." Nature Microbiology 2: 17105.

L197-217: Many of the biogeochemical processes represented by MEMS v1.0 are assumed to be microbially mediated (and therefore result in exo-enzyme breakdown and $CO_2$ production), but only two lead to C assimilation into a distinct microbial biomass pool – from the water-soluble and acid-soluble litter pools (C1 and C2, respectively). In the mineral soil (i.e., pools C5, C8, C9 and C10), microbial anabolism and catabolism are implicit and considered part of the turnover of each pool. This ensures parsimony and allows model parameters to represent the differences in microbial community for each pool, as opposed to the alternative of explicit microbial pools. The C transferred from the C1 and C2 litter pools into microbial biomass is defined by a dynamic CUE parameter controlled by the N content of the input material and the lignocellulose index (LCI; defined as the ratio between acid-insoluble to the sum of acid-soluble + acid-insoluble) of the litter layer (i.e., lower CUE results when a higher proportion of the litter is acid-insoluble). Including microbially-explicit processes in the litter layer helps to determine the proportion of C inputs that result in MAOM vs POM formation (see Liang et al., 2017) and allows for future model versions to account for distinctions between different points of entry for inputs (Sokol et al., 2018). The lack of C transferred from other pools (e.g., C3) into microbial biomass implies their decay from co-metabolism with the more labile C sources (i.e., Klotzbucher et al., 2011; Moorhead et al., 2013). Once assimilated within microbial biomass, the anabolism of microbial activity results in generation of microbial products (i.e., necromass) that form tightly bound aggregates of biofilms and small litter fragments around sand-sized soil particles (Huang et al., 2006; Buks and Kaupenjohann, 2016), and dissolved organic matter (DOM). These contribute to the heavy POM (C5) and litter DOM (C6) pools, respectively. While these specific processes are well supported by relevant literature, to retain parsimony and the generalizable structure required by an ecosystem scale model MEMS v1.0 represents microbial metabolism processes more generally (i.e., by linking them to a dynamic microbial CUE rather than specific community traits).

The wording of "litter layer" and "mineral soil" are used in a fuzzy way. Also it did not became clear to me, how model-data comparison dealt with organic layers, which are neither part of the litter (in the model lacking particulate organic matter (POM)

pools) nor the mineral soil (in the model simulating sorption to minerals). Maybe this partly causes the large model-data discrepancies for broadleaved sites with large POM stocks.

Thank you for raising this. Our categorization of the aboveground and belowground pools as litter layer and mineral soil, respectively, appear to have led to some confusion. We have changed the terminology throughout the manuscript to make clear that all belowground pools (all POM, pools C5 and C10; MAOM, pool C9 and soil DOC, pool C8) are operationally defined as < 2mm in size and sum to what we refer to as total soil (i.e., not that 'mineral soil' only refers to MAOM and that we are using the terms to differentiate between mineral and organic soil layers).

It is our intention that the sum of the C1/2/3 pools equal all the carbon inputs as above and below ground litter. However, we do not have any 'litter layer' measurements to provide us model-data comparisons. In fact, both above- and below-ground litter was removed during the LUCAS soil sampling and post-processing. We agree with you that the current model does not have the ability to simulate a specific organic horizon, and this is why we removed all organic soils (> 12% OC) from our analysis. Initially, simulating organic soil layers was not our initial priority but now after seeing the current model's results it has become a priority for our next steps in development. As a result, we are working to fractionate several soils with high OC content, so we can help pa-rameterize a new model version that has a finer resolution of soil layers to depth. We are also adding an explicit hydrological submodel that will help to improve the model's capability to vary decomposition processes in different environmental conditions. Both should help reduce some of the large model-data discrepancies from this first version.

It is our belief that the model structure should not need to change to better represent an organic horizon (which will be dominated by litter and POM pools), but rather pa-rameter values may differ to help represent how decomposer communities differ with depth/access to fresh inputs. Additionally, we are aware that if future model versions are to represent an organic horizon we would need to implement a mechanism that
reduces sorption to mineral surfaces accordingly to account for large POM accrual (for example in anaerobic conditions). This will be a key feature when we look to test the model in peaty soils where the 'mineral layer' is moved further from the surface while POM (and associated organic layer) accumulates.

Detailed comments for model structure:

Could you, please, elaborate a bit more why you (as well as the LIDEL model) choose microbes to not consume DOM?

This fundamentally comes down to the way we are 'feeding' the microbial pool (in both models). Our assumption is that the microbes consume fresh inputs from the water- and acid- soluble, coarse organic matter (pools C1/C2) and the aboveground DOM (pool C6) that exists is, in fact, what is left over and available to move to the soil. We decided to use this formulation to enable the C6 pool to be measurable as, for example, using the approach described in Soong et al., 2015. Belowground, microbes are assumed to be consuming soil DOM (pool C8) but those processes are implicit to the mineralization equations of those pools, and not related to the microbial assimilation pathways aboveground.

To help clarify this in the manuscript we have revised our descriptions and justification of why we have a microbial pool in MEMS – the primary purpose (at least in this initial model version) is to clearly differentiate between an "ex-vivo" more physical path to SOM formation and an in-vivo microbial processing one (sensu Cotrufo et al., 2015 and Liang et al., 2017). This formulation will be very helpful when trying to match real-world observations of the stoichiometry of different fractions with their corresponding pools in the model.

Cotrufo, M. F., et al. (2015). "Soil organic matter formation from biochemical and physical pathways of litter mass loss." Nature Geosciences. Liang, C., et al. (2017). "The importance of anabolism in microbial control over soil carbon storage." Nature Microbiology 2: 17105.

See quoted text shown above lines around 201 in the main manuscript and L163-174 in the supplementary: Where ãĂŰ(_j^)CxãĂŮ_in^Cy refers to DOM leaching from pool y to pool x on day j. The parameters used are detailed in Table 2 in the main manuscript, and/or defined in previous equation in this section. Note that pool C6 is not the DOM consumed by microbial biomass but rather the amount leftover after microbial activity. In this initial model version, the litter layer only refers to the aboveground component, but the same structure can equally apply to belowground C inputs such as root death. However, measurably, the DOM in the C6 pool is directly equivalent to the belowground soil DOM (C8). In MEMS v1.0, DOM enters the soil through the C6 pool only. When explicit inputs from belowground litter (e.g., roots) are simulated in future versions Eqs. 28-31 can apply for each soil layer adding the DOM that is in excess of microbial activity directly to pool C8 instead of the 'C6' shown in the equations above.

In the LIDEL model there is a C5 microbial products pool also in the litter layer, why do you assume in MEMS that all microbial turnover is transferred to the mineral soil?

The LIDEL model doesn't represent soil, thus there was the need for a microbial product pool in it. The main reason why the C5 pool in MEMS v1 is a SOC pool is because there is little added value (or sense) and the downside of increased complexity if we were to include a specific microbial products pool in soil – heavy SOC pools are made mostly of microbial products. Furthermore, microbial turnover in SOC pools is implicit and thus the microbial products generated by these processes is captured only by the mineralization of each of these pools.

You choose decomposition to be independent of the size of biomass pool to avoid some problematic feedback. Then I suggest to simplify the model even more by replacing microbial biomass turnover by the sum of inputs to the biomass pool. Then you do not need to simulate this pool, save one state variable and several model parameters. If microbial biomass is required for data comparison, you can still compute it assuming near steady state with inputs (e.g. Wutzler 2013).

We discussed at length the possibilities of removing the microbial biomass pool and came to the conclusion that it is required to help us differentiate SOM formation pathways (as mentioned above) and in future versions the "point of entry" sensu Sokol et al. 2018. We acknowledge that your suggestion would likely work for this simple first version of the MEMS model, but in the next stages of model development the fresh organic matter inputs will come from above- and below-ground sources and we will need to be able to differentiate between different rhizosphere inputs, different root types and the aboveground litter. From this, it is our intention to be able to vary parameter values associated with the microbial pool of each point of entry (e.g., aboveground, topsoil, subsoil) so as to represent variability in microbial traits. We also require an explicit microbial pool for the next stages in model development regarding N-immobilization. Since our submission of the manuscript, the Sokol et al., 2018 paper was published and discusses some of the details around our assumptions regarding the split between plant- and microbe-derived SOM, and the importance of getting this right.

Sokol, N. W., Sanderman, J., & Bradford, M. A. (2018). Pathways of mineral‐associated soil organic matter formation: Integrating the role of plant carbon source, chemistry, and point of entry. Global change biology.

Detailed comments for model-data integration:

It should be clarified better also in the discussion, that the model performance was judged by comparing steady state MAOM pools to observations. I am still looking forward to a comparison where a model successfully simulated dynamics compared to observed changes decadal stock changes across many sites.

We agree – we're excited to test the model's ability to replicate 'short-term' changes in soil organic matter dynamics. We have now made it clear that model-data comparisons are against steady-state systems.

L376: averaging parameters is dangerous, because of nonlinearities. I suggest to use only one non-averaged parameter combination. You may pick the fold randomly.

We have made this change and chosen parameter values from a single fold (values were only very slightly different from the averages – see Table S2).

L417-425 To determine the optimized parameter values, a single fold was chosen at random from those that reported the lowest RMSE for each subset of training sites (i.e., each fold). Optimized values differ depending on which measured fraction is compared to model predictions (whether comparing pool C9 to measured MAOM-C, the sum of pools C5 and C10 to measured total POM-C, or the sum of pools C5, C8, C9 and C10 to measured bulk SOC). The new, optimized parameter values (Table S2) were derived from a randomly chosen fold that minimized RMSE when compared to the MAOM fraction.

L376: You can avoid the choice of one criterion among three data streams of MAOM-C, POM-C and bulk SOC by using a cost function based on the sum of squared residuals of all the data streams.

This is a good point. Thanks for the suggestion which we will apply for the next stage of calibration. For this initial parameter estimation, we performed the full optimization procedure on all data streams. While parameter values did vary, the results and general fit was similar regardless of which criterion we chose. Consequently, we do not feel that this change would make considerable difference to the results we are presenting and hope you will agree it would not be worth redoing the entire analysis for this change. An additional factor to consider is that our ongoing development of the MEMS model is already revising some of the parameters (many are being adjusted to accommodate nitrogen effects on carbon transfers) and therefore the values themselves may have little application beyond this initial version.

We do agree with your suggestion though and have added this to the discussion.

L422-428 The new, optimized parameter values (Table S2) were derived from a randomly chosen fold that minimized RMSE when compared to the MAOM fraction. This was chosen (instead of those optimized for POM or bulk SOC) since the MAOM fraction is typically the largest single soil C pool and using this approach led to the biggest overall decrease in RMSE when compared to all available data (Table S2). In future analyses, a more rigorous approach may be to apply a cost function regarding all available measured pool data (e.g., including litter pool data when it is also measured) but for our initial model evaluation we random choice is deemed sufficient

Fig 5: The classification to land use not particularly helpful, because variables are very similar with a high range across these classes, including the mentioned significant different of MAOM:POM (L 485). Furthermore, plotting the distribution of observations and distribution of predictions separately does not help to judge model performance (L488). I suggest instead inspecting and plotting the distribution of model-data residuals of several variables and relating these differences to classes and other environmental conditions. This would indicate which variables and processes are most urgent to extend MEMS v1, as done with the discussing Fig 7.

We understand your point and have in fact plotted these all residuals against the full range of environmental conditions. Unfortunately, these tend to make the results seem worse than they are because the dense number of points near to the 0-residual line cannot be shown well. However, to address your concern we have added a residual plot to the supplementary to illustrate individual residual points (new Figure S6 – see attached in the associated zipped supplementary file). This figure does make an important point, but it is hard to determine clear recommendations of where to focus next developments purely from these figures.

Below we attach an overall summary of individual residuals against mean annual temperature of the sites and in the supplementary we show a similar figure but split by different environmental divisions (new Figure S6).

SEE FIGURE 2 AT BOTTOM OF PAGE

Other detailed comments main text:

Text in Figs 2 and 6 are hard to read. Can you provide a vector graphics of this figure? There are too many classes to distinguish by color, but I have no suggestion how to improve.

We did provide vector graphics versions with the manuscript but unfortunately as part of the peer review process they do not include them and instead choose to embed them in the file. We have tried to address this as described above. The full colour version of figure 2 (new Figure S5) is in the supplementary zipped file and we have replaced the main text figure with a one that is easier to interpret, as described above. We have also increased the font size and changed the resolution slightly of figure 6 to make it clearer. This is also obviously much clearer when viewed on the vectorized file (also attached separately – see Figure 6 in the zipped supplementary file with these responses).

Fig 4: Suddenly, pH is popping up, but was never introduced as a driving variable. I suggest to shortly state that sorption rate is pH dependent, and refer to the eq. 35 in the Appendix.

We have now made this change.

See L266: This parameter can be very difficult to generalise without requiring exhaustive information on soil physiochemical conditions (e.g., clay type, Fe/Al concentration, etc.), but the work of Mayes et al. (2012) presented an empirical relationship between $K\_lm$ and native soil pH, with pH acting as a proxy for mineralogical conditions. As a result, sorption rates to mineral surfaces are dependent on pH (see Equation 35 in supplementary). This relationship (derived from isotherms calculated for 138 soils of varying taxonomies) provides a good starting point for estimating $K\_lm$ and is also used by the MILLENNIAL model (Abramoff et al., 2017).

L 529: These are interesting effect of N in a C-only based model. While the microbially detailed models of Perveen 2014 and Wutzler 2017 attribute low litter N effects to N mining in older usually N-rich pool and accumulation of less processed material, MEMS attributes this to reduced microbial accessibility and reduced DOM production.

Do you think that chemical and stoichiometric effects are two sides of the same coin, or are these competing hypotheses? I am looking forward to the version that explicitly simulates N fluxes.

Thanks, we are excited to be working on simulating N fluxes, which indeed are complex depending on the microbial N demand (stoichiometry), as well as on the energetic (chemistry) and accessibility (physics) of soil organic C pools. Rather than competing, in our opinion these are all simultaneously at play. We follow the LIDEL model (Campbell et al., 2016), according to which both N limitation and C chemistry (i.e., Lignocellulose index) drive microbial decomposition and DOM production, with the most limiting factor being the actual driver of the process. We find the recent model of N input effects on SOC dynamics proposed by Averill and Waring (2018) to be particularly effective at capturing this complexity and may follow their logic in our new model version which will include N.

Averill, C. and B. Waring (2018). "Nitrogen limitation of decomposition and decay: How can it occur?" Global Change Biology 24(4): 1417-1427. Campbell, E. E., et al. (2016). "Tracking the fate of litter carbon using the LItter DEcomposition and Leaching (LIDEL) model." Soil Biology and Biochemistry 100: 160-174

Other detailed comments appendix:

To me its difficult to always keep a list of meaning of pool 1 to 10 in my head. Could you come up with more expressive pool names?

We are aware of this issue. The initial development of this first MEMS v1 model was intended to be an advancement from the LIDEL model and therefore we kept the same names to ease reference to that model. However, we now know that this approach won't be effective as the model grows. For our MEMS v2.0 we are making this change.

L 70: I do no find more information on ub and uk in Table 2 in the main text. I suggest referring to eqs. 19-22.

Apologies for this omission. We have made this change.

L80 More information of the parameters uB, uk, B_x, ãĂŰlaãĂŮ_x and k_x can be found in Campbell et al. (2016) and in the equations below, but briefly: (_jˆ)uB and (_jˆ)uk are rate modifiers to represent the litter chemistry controls (LCI and available nitrogen) on microbial use efficiency, on day j

L 110-112: The long sentence did not became clear to me. Is L_j_C5_C4_gen really a combined flux of bioturbation, . . ., and DOC leaching? I thought the latter one is covered by eq. 33.

You are right that this was confusing. The C4 to C5 flux was inherited from the LIDEL model to represent microbial turnover and you are right in saying that DOC generation from this process is represented elsewhere. We have adjusted the text accordingly and hopefully it is now clearer.

L120 Where ãĂŰ(_jˆ)C5ãĂŮ_genˆC4 refers to the fraction of carbon that is transferred from C4 to C5 (i.e., microbial products transported belowground when physical and hydrological processes mix between the input layer [aboveground litter only in MEMS v1.0] and soil layer) on day j.

eq 47: k8 does not match the text before that states k5.

We have made this change.

L256 While the maximum decay rates (k_x) for most pools are fixed constants, Campbell et al. (2016) suggested that k_3 is best estimated in relation to the maximum decay rate of the microbially-accessible litter (C2) pool (k_2).

Thanks for this work. I suspect MEMS to be included in further model comparisons as a complementary model.

Thank you for your insightful review. We hope the MEMS model can help to stimulate a discussion that advances SOM modelling in the coming years. It is our intention to

participate in model comparisons with MEMS v2.0.

Please also note the supplement to this comment:
https://www.biogeosciences-discuss.net/bg-2018-430/bg-2018-430-AC1-
supplement.zip

[Figure]

**Fig. 1.** New figure 2 in Robertson et al. Global sensitivity analysis with optimized parameters only colourized

**Fig. 2.** Example of all model residuals plot against mean annual temperature of site simulated

[Figure]

---

## Author Comment (AC2) · 8 Jan 2019

Thank you for your detailed review. We have responded to all your comments in the attached supplement file. We highly recommend viewing our responses from the attached PDF that is contained within the zip supplement. This attached PDF file includes formatting that makes it much easier to track which comments refer to which points raised by yourself. Additionally, the attached zip supplement includes associated vectorized PDFs of the figures we refer to - these are much easier to interpret than those copied in with this plain-text script.

The supplement file should be available here - https://www.biogeosciencesdiscuss.net/bg-2018-430/bg-2018-430-AC2-

supplement.zip

Many thanks again for your contribution to improving this work, we greatly appreciate your input. Andy Robertson and co-authors

Please also note the supplement to this comment: https://www.biogeosciences-discuss.net/bg-2018-430/bg-2018-430-AC2-supplement.zip

---

## Author Response (AR1)

Department of Soil and Crop Sciences,
Colorado State University,
Fort Collins,
Colorado 80523-1170
USA

3rd January 2019

Cover Letter and Responses to Reviewer Comments to accompany the manuscript:
**"Unifying soil organic matter formation and persistence frameworks: the MEMS model"**

**Authors:** Andy Robertson, Keith Paustian, Stephen Ogle, Matthew Wallenstein, Emanuele Lugato, and Francesca Cotrufo

      Thank you for your correspondence concerning our manuscript and for giving us the opportunity to resubmit a revised version. All comments from the reviewers have been carefully considered and appropriate responses are made below.

Sincerely,

Andy Robertson

**Responses to comments from Thomas Wutzler on "Unifying soil organic matter formation and persistence frameworks: the MEMS model" by Andy D. Robertson *et al*.**

Reviewer comments in bold and our responses in normal text. Selected new text in the revised manuscript is pasted here in italics. Reference to the manuscript is given as new line number (L).

General comments

**The study of Robertson *et al*. presents a first version of the MEMS model, a parsimonious dynamical model of soil organic carbon (SOC) development at ecosystem scale, together with a validation across many sites. I enjoyed reading the manuscript, which is well structured and succeeds in getting the fundamental ideas across in a concise way and provides the details in the appendix.**

**The proposed model is of similar complexity as classical pool-based models but better incorporates recent mechanistic understanding and is better comparable to measurable pools. Hence, it is of great interest to soil science, ecosystem research, and potentially also global change communities. It adds a complementary alternative in the suite of simple to much more detailed SOC models and the study should be published after revisions.**

Many thanks for your comments and time spent reviewing our manuscript. We appreciate the detail and clarity of your suggested revisions – this certainly helps us to improve our manuscript. We are glad you enjoyed reading it and are excited to have an opportunity to publish the MEMS model. It is our hope that it can do just as you say and add to the suite of SOC models already available and stimulate discussion of how to advance this field.

Through the revisions described in detail below we hope to have addressed all your comments.

**I liked the approach of directly modeling relevant quantities at the scale model purpose, the management scale. I liked the simulation time dependent sensitivity analysis, although Fig. 2 is hard to read.**

I suspect part of the difficulty in reading the figure is because the submission guidelines are to embed the picture as a low-quality jpg. The original vectorized PDF is much clearer. However, we have also now hopefully made the figure easier to read by increasing the size of the text and limiting the colours to only the 4 most influential parameters. All other parameters are coloured in greyscale in order from top to bottom. The 'full colour' figure version is included as a supplementary figure and attached as a lossless vector PDF for detailed inspection if the reader wishes. For your reference we show the new figure below and have also attached the vectorized full-colour PDF version to this response (now Figure S5).

[Figure]

**The supplementary is complicated by already anticipating several mineral soil layers and sometimes is inconsistent with the main text. For example, there is explicit microbial assimilation in mineral layers in the supplementary, but the main text states that microbes are implicit there. Please, provide a version that matches the main text and the presented model structure.**

We apologise for this confusion – the microbial assimilation, as a process, is indeed 'explicit' in that it is represented by fluxes into a microbial pool. However, in this inaugural version of the model the use of a microbial pool is more one to help differentiate the direct *versus* microbially-processed flux of organic matter inputs (pools C1-C3) to the soil C pools, *sensu* Liang *et al*., 2017. Once these inputs are added to the soil pools belowground, then the microbial biomass and associated metabolic processes are implicit (i.e., we assume there is microbial activity and mineralization of the carbon within these soil pools, but we do not represent these processes with discrete pools or fluxes). We certainly appreciate the comment because on review this is an important point that needed to be made clearer. At different points in the main manuscript, we have added additional points as to why we have a distinct microbial pool at the point of entry of the C input, but not after it is processed and transformed into the SOC pools, which have microbes within them.

A large part of the confusion likely resulted from the 'microbial assimilation from litter' section of the supplementary and we can indeed understand why. The use of the layer superscript certainly made our descriptions less clear. Consequently, we have also removed the superscript notations for soil layers that created unnecessary confusion in the supplementary model description. There should now be no inconsistency between the main text and the supplementary materials.

Liang, C., *et al*. (2017). "The importance of anabolism in microbial control over soil carbon storage." *Nature Microbiology* 2: 17105.

*L197-217:*
*Many of the biogeochemical processes represented by MEMS v1.0 are assumed to be microbially mediated (and therefore result in exo-enzyme breakdown and $CO_2$ production), but only two lead to C assimilation into a distinct microbial biomass pool – from the water-soluble and acid-soluble litter pools (C1 and C2, respectively). In the mineral soil (i.e., pools C5, C8, C9 and C10), microbial anabolism and catabolism are implicit and considered part of the turnover of each pool. This ensures parsimony and allows model parameters to represent the differences in microbial community for each pool, as opposed to the alternative of explicit microbial pools. The C transferred from the C1 and C2 litter pools into microbial biomass is defined by a dynamic CUE parameter controlled by the N content of the input material and the lignocellulose index (LCI; defined as the ratio between acid-insoluble to the sum of acid-soluble + acid-insoluble) of the litter layer (i.e., lower CUE results when a*

*higher proportion of the litter is acid-insoluble). Including microbially-explicit processes in the litter layer helps to determine the proportion of C inputs that result in MAOM vs POM formation (see Liang et al., 2017) and allows for future model versions to account for distinctions between different points of entry for inputs (Sokol et al., 2018). The lack of C transferred from other pools (e.g., C3) into microbial biomass implies their decay from co-metabolism with the more labile C sources (i.e., Klotzbucher et al., 2011; Moorhead et al., 2013). Once assimilated within microbial biomass, the anabolism of microbial activity results in generation of microbial products (i.e., necromass) that form tightly bound aggregates of biofilms and small litter fragments around sand-sized soil particles (Huang et al., 2006; Buks and Kaupenjohann, 2016), and dissolved organic matter (DOM). These contribute to the heavy POM (C5) and litter DOM (C6) pools, respectively. While these specific processes are well supported by relevant literature, to retain parsimony and the generalizable structure required by an ecosystem scale model MEMS v1.0 represents microbial metabolism processes more generally (i.e., by linking them to a dynamic microbial CUE rather than specific community traits).*

**The wording of "litter layer" and "mineral soil" are used in a fuzzy way. Also it did not became clear to me, how model-data comparison dealt with organic layers, which are neither part of the litter (in the model lacking particulate organic matter (POM) pools) nor the mineral soil (in the model simulating sorption to minerals). Maybe this partly causes the large model-data discrepancies for broadleaved sites with large POM stocks.**

Thank you for raising this. Our categorization of the aboveground and belowground pools as litter layer and mineral soil, respectively, appear to have led to some confusion. We have changed the terminology throughout the manuscript to make clear that all belowground pools (all POM, pools C5 and C10; MAOM, pool C9 and soil DOC, pool C8) are operationally defined as < 2mm in size and sum to what we refer to as *total soil* (i.e., ***not*** that 'mineral soil' only refers to MAOM and that we are using the terms to differentiate between mineral and organic soil layers).

It is our intention that the sum of the C1/2/3 pools equal all the carbon inputs as above and below ground litter. However, we do not have any 'litter layer' measurements to provide us model-data comparisons. In fact, both above- and below-ground litter was removed during the LUCAS soil sampling and post-processing. We agree with you that the current model does not have the ability to simulate a specific organic horizon, and this is why we removed all organic soils (> 12% OC) from our analysis. Initially, simulating organic soil layers was not our initial priority but now after seeing the current model's results it has become a priority for our next steps in development. As a result, we are working to fractionate several soils with high OC content, so we can help parameterize a new model version that has a finer resolution of soil layers to depth. We are also adding an explicit hydrological submodel that will help to improve the model's capability to vary decomposition processes in different environmental conditions. Both should help reduce some of the large model-data discrepancies from this first version.

It is our belief that the model structure should not need to change to better represent an organic horizon (which will be dominated by litter and POM pools), but rather parameter values may differ to help represent how decomposer communities differ with depth/access to fresh inputs. Additionally, we are aware that if future model versions are to represent an organic horizon we would need to implement a mechanism that reduces sorption to mineral surfaces accordingly to account for large POM accrual (for example in anaerobic conditions). This will be a key feature when we look to test the model in peaty soils where the 'mineral layer' is moved further from the surface while POM (and associated organic layer) accumulates.

Detailed comments for model structure:

**Could you, please, elaborate a bit more why you (as well as the LIDEL model) choose microbes to not consume DOM?**

This fundamentally comes down to the way we are 'feeding' the microbial pool (in both models). Our assumption is that the microbes consume fresh inputs from the water- and acid- soluble, coarse organic matter (pools C1/C2) and the aboveground DOM (pool C6) that exists is, in fact, what is left over and available to move to the soil. We decided to use this formulation to enable the C6 pool to be measurable as, for example, using the approach described in Soong *et al.*, 2015. Belowground, microbes are assumed to be consuming soil DOM (pool C8) but those processes are implicit to the mineralization equations of those pools, and not related to the microbial assimilation pathways aboveground.

To help clarify this in the manuscript we have revised our descriptions and justification of why we have a microbial pool in MEMS – the primary purpose (at least in this initial model version) is to clearly differentiate between an "ex-vivo" more physical path to SOM formation and an in-vivo microbial processing one (*sensu* Cotrufo *et al.*, 2015 and Liang *et al.*, 2017). This formulation will be very helpful when trying to match real-world observations of the stoichiometry of different fractions with their corresponding pools in the model.

Cotrufo, M. F., *et al.* (2015). "Soil organic matter formation from biochemical and physical pathways of litter mass loss." *Nature Geosciences*.
Liang, C., *et al.* (2017). "The importance of anabolism in microbial control over soil carbon storage." *Nature Microbiology* **2**: 17105.

*See quoted text shown above lines around 201 in the main manuscript and L163-174 in the supplementary:*
*Where $_jCx_{in}^{Cy}$ refers to DOM leaching from pool y to pool x on day j. The parameters used are detailed in Table 2 in the main manuscript, and/or defined in previous equation in this section. Note that pool C6 is not the DOM consumed by microbial biomass but rather the amount leftover after microbial activity. In this initial model version, the litter layer only refers to the aboveground component, but the same structure can equally apply to belowground C inputs such as root death. However, measurably, the DOM in the C6 pool is directly equivalent to the belowground soil DOM (C8). In MEMS v1.0, DOM enters the soil through the C6 pool only. When explicit inputs from belowground litter (e.g., roots) are simulated in future versions Eqs. **28-31** can apply for each soil layer adding the DOM that is in excess of microbial activity directly to pool C8 instead of the 'C6' shown in the equations above.*

**In the LIDEL model there is a C5 microbial products pool also in the litter layer, why do you assume in MEMS that all microbial turnover is transferred to the mineral soil?**

The LIDEL model doesn't represent soil, thus there was the need for a microbial product pool in it. The main reason why the C5 pool in MEMS v1 is a SOC pool is because there is little added value (or sense) and the downside of increased complexity if we were to include a specific microbial products pool in soil – heavy SOC pools are made mostly of microbial products. Furthermore, microbial turnover in SOC pools is implicit and thus the microbial products generated by these processes is captured only by the mineralization of each of these pools.

**You choose decomposition to be independent of the size of biomass pool to avoid some problematic feedback. Then I suggest to simplify the model even more by replacing microbial biomass turnover by the sum of inputs to the biomass pool. Then you do not need to simulate this pool, save one state variable and several model parameters. If microbial biomass is required for data comparison, you can still compute it assuming near steady state with inputs (e.g. Wutzler 2013).**

We discussed at length the possibilities of removing the microbial biomass pool and came to the conclusion that it is required to help us differentiate SOM formation pathways (as mentioned above) and in future versions the "point of entry" *sensu* Sokol *et al.* 2018. We acknowledge that your suggestion would likely work for this simple first version of the MEMS model, but in the next stages of model development the fresh organic matter inputs will come from above- and below-ground sources and we will need to be able to differentiate between different rhizosphere inputs, different root types and the aboveground litter. From this, it is our intention to be able to vary parameter values associated with the microbial pool of each point of entry (e.g., aboveground, topsoil, subsoil) so as to represent variability in microbial traits. We also require an explicit microbial pool for the next stages in model development regarding N-immobilization. Since our submission of the manuscript, the Sokol *et al.*, 2018 paper was published and discusses some of the details around our assumptions regarding the split between plant- and microbe-derived SOM, and the importance of getting this right.

Sokol, N. W., Sanderman, J., & Bradford, M. A. (2018). Pathways of mineral-associated soil organic matter formation: Integrating the role of plant carbon source, chemistry, and point of entry. *Global change biology*.

Detailed comments for model-data integration:

**It should be clarified better also in the discussion, that the model performance was judged by comparing steady state MAOM pools to observations. I am still looking forward to a comparison where a model successfully simulated dynamics compared to observed changes decadal stock changes across many sites.**

We agree – we're excited to test the model's ability to replicate 'short-term' changes in soil organic matter dynamics. We have now made it clear that model-data comparisons are against steady-state systems.

**L376: averaging parameters is dangerous, because of nonlinearities. I suggest to use only one non-averaged parameter combination. You may pick the fold randomly.**

We have made this change and chosen parameter values from a single fold (values were only very slightly different from the averages – see Table S2).

*L417-425*
*To determine the optimized parameter values, a single fold was chosen at random from those that reported the lowest RMSE for each subset of training sites (i.e., each fold). Optimized values differ depending on which measured fraction is compared to model predictions (whether comparing pool C9 to measured MAOM-C, the sum of pools C5 and C10 to measured total POM-C, or the sum of pools C5, C8, C9 and C10 to measured bulk SOC). The new, optimized parameter values (Table S2) were derived from a randomly chosen fold that minimized RMSE when compared to the MAOM fraction.*

**L376: You can avoid the choice of one criterion among three data streams of MAOM-C, POM-C and bulk SOC by using a cost function based on the sum of squared residuals of all the data streams.**

This is a good point. Thanks for the suggestion which we will apply for the next stage of calibration. For this initial parameter estimation, we performed the full optimization procedure on all data streams. While parameter values did vary, the results and general fit was similar regardless of which criterion we chose. Consequently, we do not feel that this change would make considerable difference to the results we are presenting and hope you will agree it would not be worth redoing the entire analysis for this change. An additional factor to consider is that our ongoing development of the MEMS model is already revising some of the parameters (many are being adjusted to accommodate nitrogen effects on carbon transfers) and therefore the values themselves may have little application beyond this initial version.

We do agree with your suggestion though and have added this to the discussion.

*L422-428*
*The new, optimized parameter values (Table S2) were derived from a randomly chosen fold that minimized RMSE when compared to the MAOM fraction. This was chosen (instead of those optimized for POM or bulk SOC) since the MAOM fraction is typically the largest single soil C pool and using this approach led to the biggest overall decrease in RMSE when compared to all available data (Table S2). In future analyses, a more rigorous approach*

*may be to apply a cost function regarding all available measured pool data (e.g., including litter pool data when it is also measured) but for our initial model evaluation we random choice is deemed sufficient*

**Fig 5: The classification to land use not particularly helpful, because variables are very similar with a high range across these classes, including the mentioned significant different of MAOM:POM (L 485). Furthermore, plotting the distribution of observations and distribution of predictions separately does not help to judge model performance (L488).**
**I suggest instead inspecting and plotting the distribution of model-data residuals of several variables and relating these differences to classes and other environmental conditions. This would indicate which variables and processes are most urgent to extend MEMS v1, as done with the discussing Fig 7.**

We understand your point and have in fact plotted these all residuals against the full range of environmental conditions. Unfortunately, these tend to make the results seem worse than they are because the dense number of points near to the 0-residual line cannot be shown well. However, to address your concern we have added a residual plot to the supplementary to illustrate individual residual points (new Figure S6). This figure does make an important point, but it is hard to determine clear recommendations of where to focus next developments purely from these figures.

Below we attach an overall summary of individual residuals against mean annual temperature of the sites and in the supplementary we show a similar figure but split by different environmental divisions (new Figure S6).

[Figure]

Other detailed comments main text:

**Text in Figs 2 and 6 are hard to read. Can you provide a vector graphics of this figure? There are too many classes to distinguish by color, but I have no suggestion how to improve.**

We did provide vector graphics versions with the manuscript but unfortunately as part of the peer review process they do not include them and instead choose to embed them in the file. We have tried to address this as described above. The full colour version of figure 2 is in the supplementary and we have replaced the main text figure with a one that is easier to interpret, as described above. We have also increased the font size and changed the resolution slightly of figure 6 to make it clearer. This is also obviously much clearer when viewed on the vectorized file. Also attached separately.

**Fig 4: Suddenly, pH is popping up, but was never introduced as a driving variable. I suggest to shortly state that sorption rate is pH dependent, and refer to the eq. 35 in the Appendix.**

We have now made this change.

*See L266:*
*This parameter can be very difficult to generalise without requiring exhaustive information on soil physiochemical conditions (e.g., clay type, Fe/Al concentration, etc.), but the work of Mayes et al. (2012) presented an empirical relationship between K_lm and native soil pH, with pH acting as a proxy for mineralogical conditions. As a result, sorption rates to mineral surfaces are dependent on pH (see Equation 35 in supplementary). This relationship (derived from isotherms calculated for 138 soils of varying taxonomies) provides a good starting point for estimating K_lm and is also used by the MILLENNIAL model (Abramoff et al., 2017).*

**L 529: These are interesting effect of N in a C-only based model. While the microbially detailed models of Perveen 2014 and Wutzler 2017 attribute low litter N effects to N mining in older usually N-rich pool and accumulation of less processed material, MEMS attributes this to reduced microbial accessibility and reduced DOM production. Do you think that chemical and stoichiometric effects are two sides of the same coin, or are these competing hypotheses? I am looking forward to the version that explicitly simulates N fluxes.**

Thanks, we are excited to be working on simulating N fluxes, which indeed are complex depending on the microbial N demand (stoichiometry), as well as on the energetic (chemistry) and accessibility (physics) of soil organic C pools. Rather than competing, in our opinion these are all simultaneously at play. We follow the LIDEL model (Campbell *et al*., 2016), according to which both N limitation and C chemistry (i.e., Lignocellulose index) drive microbial decomposition and DOM production, with the most limiting factor being the actual driver of the process. We find the recent model of N input effects on SOC dynamics proposed by Averill and Waring (2018) to be particularly effective at capturing this complexity and may follow their logic in our new model version which will include N.

Averill, C. and B. Waring (2018). "Nitrogen limitation of decomposition and decay: How can it occur?" *Global Change Biology* **24**(4): 1417-1427.
Campbell, E. E., *et al*. (2016). "Tracking the fate of litter carbon using the LItter DEcomposition and Leaching (LIDEL) model." *Soil Biology and Biochemistry* **100**: 160-174

Other detailed comments appendix:

**To me its difficult to always keep a list of meaning of pool 1 to 10 in my head. Could you come up with more expressive pool names?**

We are aware of this issue. The initial development of this first MEMS v1 model was intended to be an advancement from the LIDEL model and therefore we kept the same names to ease reference to that model. However, we now know that this approach won't be effective as the model grows. For our MEMS v2.0 we are making this change.

**L 70: I do no find more information on ub and uk in Table 2 in the main text. I suggest referring to eqs. 19-22.**

Apologies for this omission. We have made this change.

*L80*
*More information of the parameters uB, uk, $B_x$, $la_x$ and $k_x$ can be found in Campbell et al. (2016) and in the equations below, but briefly:*
*$_juB$ and $_juk$ are rate modifiers to represent the litter chemistry controls (LCI and available nitrogen) on microbial use efficiency, on day j*

**L 110-112: The long sentence did not became clear to me. Is L_j_C5_C4_gen really a combined flux of bioturbation, . . ., and DOC leaching? I thought the latter one is covered by eq. 33.**

You are right that this was confusing. The C4 to C5 flux was inherited from the LIDEL model to represent microbial turnover and you are right in saying that DOC generation from this process is represented elsewhere. We have adjusted the text accordingly and hopefully it is now clearer.

*L120*
*Where $_jC5_{gen}^{C4}$ refers to the fraction of carbon that is transferred from C4 to C5 (i.e., microbial products transported belowground when physical and hydrological processes mix between the input layer [aboveground litter only in MEMS v1.0] and soil layer) on day j.*

**eq 47: k8 does not match the text before that states k5.**

We have made this change.

*L256*
*While the maximum decay rates ($k_x$) for most pools are fixed constants, Campbell et al. (2016) suggested that $k_3$ is best estimated in relation to the maximum decay rate of the microbially-accessible litter (C2) pool ($k_2$).*

**Thanks for this work. I suspect MEMS to be included in further model comparisons as a complementary model.**

Thank you for your insightful review. We hope the MEMS model can help to stimulate a discussion that advances SOM modelling in the coming years. It is our intention to participate in model comparisons with MEMS v2.0.

**Responses to comments from Anonymous Referee #2 on "Unifying soil organic matter formation and persistence frameworks: the MEMS model" by Andy D. Robertson *et al*.**

Reviewer comments in bold and our responses in normal text. Selected new text in the revised manuscript is pasted here in italics. Reference to the manuscript is given as line number (L).

Overall review

**The authors present a new soil biogeochemistry model, MEMS v1.0, that explicitly represents biochemical complexity of litter pools, microbial biomass, mineral associated organic matter and particulate organic matter. The model has the capability of including variable CUE in litter decomposition and mechanisms leading to SOM stabilization and saturation of mineral associated carbon fraction. Four key model parameters are calibrated to reproduce soil fractionation observations of mineral associated and particulate organic matter fractions and the model is evaluated in reproducing topsoil SOC in more than 8000 sites across different land-uses in Europe with satisfactorily results.**

**Constructing models that are based on measurable carbon pools rather than on the old framework assigning turnover rates to a given number of unmeasurable carbon pools is a very important endeavor and the authors are definitely moving beyond conventional SOC modeling. It is especially important to have models that link litter decomposition processes and SOM formation processes, which is rarely the case, as stated by the authors (L 89-91). I am very much in favor of such a type of approach and supportive of the author's effort. The manuscript is very well written and clearly presented and the introduction frames very well the problem.**

**I would be happy to have a few clarifications on some technical aspects and about one important assumption related to the role of the microbial pool. These are written in a number of minor comments that hopefully can be addressed.**

Many thanks for your constructive comments and praise. We have responded to each of your comments in detail below and hope to have satisfactorily addressed any concerns or queries you may have had. Regarding your points about the microbial pool please see our detailed response on those comments below. It is our hope that this publication and the resulting MEMS model can help to both stimulate a fruitful discussion and advance the practice of SOC modelling.

**I would also invite the authors to tone down the role of MEMS v1.0 as "ecosystem model", since the current version is still far from being there. As a matter of fact, in several instances (e.g., Line 606) the authors state that the model is incomplete (e.g., lack of hydrological and nutrient cycle) and that these deficiencies will be addressed in future model developments. The model represents SOM dynamics at the "ecosystem scale". However, for various reasons but especially because the temporal dynamics are not evaluated in this article, I would invite to use cautious statements in the link with ecosystem models. Only the steady-state conditions are tested. A correct representation of temporal dynamics is key for coupling with other models. At this stage, this is a quite significant limitation for application in ecosystem models. Furthermore, feedbacks between soil and vegetation cannot be considered.**

Thank you for your point. We are fully aware of the limitations of this first version of our model and readily acknowledge that it is not an ecosystem model yet. It was never our intention to 'oversell' the model's capability but to rather highlight the possibilities for integration with other ecosystem model components (e.g., plant growth, hydrology, etc.) given the more realistic model structure. You are certainly correct that being able to simulate non-steady-state dynamics will be the true test of our model and to date it is more of a working proof of concept model than one to directly compare with conventional SOM models.

Throughout the revised manuscript we have tried to play down links or comparisons with true ecosystem models. However, we do maintain that the model is designed to operate the ecosystem *scale*. We have also added a few points to highlight the limitations of our steady-state comparative approach.

*L318-320*
*These driving variables are external inputs of the initial model version but may be obtained from coupled climate and plant growth submodels in future versions, when incorporated into a full ecosystem model.*

*L575-582*
*MEMS v1.0 was designed to consolidate recent advances in our understanding of SOM formation and persistence into a parsimonious mathematical model that uses a generalizable structure which, after further development, can be implemented in Ecosystem and Earth System model applications*

*L665-667*
*In its current capacity, MEMS v1.0 is far from being able to simulate full ecosystems and is limited in scope regarding the land use scenarios it can simulate accurately.*

**Other simplifications are that NPP is prescribed from MODIS, the model does not account for temporal dynamics of soil moisture or for nutrient cycles, the root:shoot ratio is prescribed for various biomes. However, these are overall clearly described. I would also appreciate some additional discussion about the issue in comparing pools, which are spun up at the equilibrium with observed pools (Line 366-367). The authors are aware of the issue and they briefly discussed it. However, most of the description of the results and the calibration effort convey somehow the intention to match C-pools as closely as possible. Given the expected difference between actual SOC and "steady- state" SOC, I would have allowed more freedom to the model and focus on comparing patterns as in Fig. 5 and 6 rather than absolute quantities.**

The focus of comparing patterns rather than absolutes was indeed our initial end goal, however after we ran the model and saw relatively good agreement with absolutes as well we felt it important to report these results. We agree that there are many reasons why our simulated SOC stocks would not match those measured but our choice to only look at grasslands and forests was a way to examine those sites that may be in, or close to, equilibrium. Your point is a good one though and we have tried to adjust some of our language in the discussion to focus more on comparisons with general patterns than on exact numbers. Several qualifying statements have been included when we do compare with absolutes.

*L452-454*
*In addition to comparing measured values with those predicted at steady-state (which may not be an accurate assumption for many sites), a more general comparison was performed to examine groups of sites under similar site conditions.*

*L565-569*
*While the model's performance comparing absolute C stocks appears good, this is done with the assumption that these topsoil C stocks at forest and grassland sites in our analysis are at steady-state. This is unlikely to be true and therefore it is encouraging when general trends are as expected (as is the case for many of the land uses and for many of the different environmental divisions; Figure 6).*

*L606-608*
*There are also limitations of our approach given that very few of the sites will likely be under true steady-state conditions, leading to further discrepancies between model predictions and measured values.*

**Despite these limitations, the manuscript is undoubtedly a novel contribution to the field and surely a step in the right direction.**

Many thanks for your comments and time spent reviewing our manuscript. We certainly appreciate the opportunity to add the MEMS model to those currently driving progress in the field of SOM modelling.
* * *
Minor comments

**Line 75. It is cited later on, however, Wieder et al 2015 would fit well also here.**

We have now added this.

*L80-82*
*Consequently, there have been several calls to represent this new understanding and re-examine how microbial activity is simulated in SOM models (Schmidt et al., 2011; Moorhead et al., 2014; Campbell and Paustian, 2015; Wieder et al., 2015).*

**Line 96. Maybe one sentence with additional explanations for K vs r strategies (e.g., copiotrophic and oligotrophic microbial functional groups) is necessary, not all the "modelers" may be aware of these concepts.**

Thank you for the suggestion – we have now added this extra detail.

*L103*
*A recent paradigm has emerged that emphasizes the role of microbial life strategies (e.g., K vs r, referring to copiotrophic and oligotrophic microbial functional groups) and carbon use efficiency (CUE) in the formation of SOM from plant inputs (Dorodnikov et al., 2009; Cotrufo et al., 2013; Lehmann and Kleber, 2015; Kallenbach et al., 2016).*

**Line 113-114. The issue related to the lack of inputs or information to derive model parameters and validate model responses, of course, is a very important one and may compromise practicality as written by the authors. However, modeling efforts in the direction of more mechanistic representations of the soil system can shed light on the importance of processes and interactions that were not accounted or quantified before, they may provide guesses for the magnitude of certain pools/fluxes and may motivate the collection of those data that are necessary to test mechanistic predictions. In other words, they can have a value in process explanation rather than a predictive value.**

A good point, well raised. We have added this to the introduction help bolster the points we made. Thank you.

*See L60-65:*
*Structuring a SOM model around these known and quantifiable biogeochemical pools and processes has the potential to drastically reduce uncertainty by enhancing opportunities for parameterization and validation of models with empirical data. Furthermore, mechanistic models can have value in process explanation as well their value in predictive capabilities; such models can pinpoint the processes that have the greatest influence on a system even when they are not traditionally determined empirically.*

**Line 174-178. In a certain way, also the CENTURY model, especially in more updated versions (e.g., Kirschbaum and Paul, 2002) accounts for nitrogen and lignin content of the litter, which are affecting the turnover rates of the various litter pools. Additionally, their subdivision in metabolic and structural litter pools is not far from the subdivision in the pools C1, C2, C3. This may be**

**acknowledged in the manuscript or if major differences, which I cannot recognize, do exist, they need to be remarked.**

We feel that the MEMS interpretation of these divisions is different to those in CENTURY, but we do acknowledge the similarities. However, these alterations may not qualify as 'major differences' but rather different formulations of the same general ideas. For example, at this early stage the litter chemistry and N content of the inputs are fixed and therefore similar to the lignin:N effects in CENTURY, however when we include a discrete N submodel, N-availability will be dynamic and influence those processes differently through time.

With this first description of MEMS we do not mean to suggest that it is better or worse to any of the more conventional models (including CENTURY) but rather that it presents another way of addressing the same questions about SOM dynamics. In some respects, MEMS is very similar to other models, and in other respects it is quite different. A full model-vs-model comparison was obviously beyond the scope of this manuscript. Therefore, to avoid direct comparisons between the conventional SOM models and MEMS, we deliberately did not discuss specifics about how they differ. To hopefully address this comment, we have added a single sentence to help clarify our position.

*See L186-188*
*This structure is similar to the LIDEL model (Campbell et al., 2016) and follows the hypotheses that both N availability and lignin content influence decomposition by affecting microbial activity (Aber et al., 1990; Manzoni et al., 2008; Sinsabaugh et al., 2013; Moorhead et al., 2013). Similar approaches have also been used in many of the updated traditional SOM models (e.g., lignin:N ratios in CENTURY; Kirschbaum and Paul, 2002).*

**Line 189-190. The assumption of considering a microbial pool (C4) for the litter component is probably the decision in terms of model construction, which leaves me more bewildered. This pool, presumably, is mostly located aboveground, even though is not stated explicitly, and does not have an explicit role in the turnover of soil organic matter. Now, if anything, I would have make the reverse choice. Because of accessibility constraints and relatively paucity of microbial biomass in the soil, the decomposition of SOM is likely controlled explicitly by microbial biomass, while the decomposition of litter, which is mostly located aboveground (especially for land covers different from grassland) and air exposed is unlikely limited by microbial biomass. Maybe, my understanding of the system is wrong, but it would be useful to have a clarification of the rationale of such an assumption and eventually of the potential consequences.**

Your understanding of the systems is perfectly correct. However, our decision to explicitly represent microbes in the litter layer of MEMS v1.0 was based on their importance informing the relevant SOM formation pathways (i.e., direct *vs* microbially-processed), not their impacts on decomposition. Consequently, this is also why we deliberately did not limit the discussion of a microbial pool to aboveground litter only – our structure implies that there must be a microbial pool at each point of carbon input (e.g., the litter layer, rhizosphere, etc.) so that the model can account for the carbon inputs that are microbially processed, and the amount of DOM that results.

We have added some extra information in the main manuscript (excerpts below) but also wanted to include a little more detail here to help clarify our rationale of why we have a microbial pool. At potential different "points of entry", carbon inputs contribute to MAOM or POM formation in differential amounts depending on the microbial community (as per Sokol *et al.* 2018). This is represented by the MEMS model structure by having an explicit microbial pool when organic matter enters the system but not after it; belowground, microbial biomass and associated metabolic processes are implicit (i.e., we assume there is microbial activity and mineralization of the carbon within these soil pools but we do not represent these processes with discrete pools or fluxes).

Sokol, N. W., Sanderman, J., & Bradford, M. A. (2018). Pathways of mineral-associated soil organic matter formation: Integrating the role of plant carbon source, chemistry, and point of entry. *Global change biology*.

*L201-205:*
*Many of the biogeochemical processes represented by MEMS v1.0 are assumed to be microbially mediated (and therefore result in exo-enzyme breakdown and $CO_2$ production), but only two lead to C assimilation into a distinct microbial biomass pool – from the water-soluble and acid-soluble litter pools (C1 and C2, respectively). In the mineral soil (i.e., pools C5, C8, C9 and C10), microbial anabolism and catabolism are implicit and considered part of the turnover of each pool. This ensures parsimony and allows model parameters to represent the differences in microbial community for each pool, as opposed to the alternative of explicit microbial pools. The C transferred from the C1 and C2 litter pools into microbial biomass is defined by a dynamic CUE parameter controlled by the N content of the input material and the lignocellulose index (LCI; defined as the ratio between acid-insoluble to the sum of acid-soluble + acid-insoluble) of the litter layer (i.e., lower CUE results when a higher proportion of the litter is acid-insoluble). Including microbially-explicit processes in the litter layer helps to determine the proportion of C inputs that result in MAOM vs POM formation (see Liang et al., 2017) and allows for future model versions to account for distinctions between different points of entry for inputs (Sokol et al., 2018). The lack of C transferred from other pools (e.g., C3) into microbial biomass implies their decay from co-metabolism with the more labile C sources (i.e., Klotzbucher et al., 2011; Moorhead et al., 2013). Once assimilated within microbial biomass, the anabolism of microbial activity results in generation of microbial products (i.e., necromass) that form tightly bound aggregates of biofilms and small litter fragments around sand-sized soil particles (Huang et al., 2006; Buks and Kaupenjohann, 2016), and dissolved organic matter (DOM). These contribute to the heavy POM (C5) and litter DOM (C6) pools, respectively. While these processes are well supported by relevant literature, to retain parsimony MEMS v1.0 represents microbial metabolism processes implicitly as per their description in LIDEL.*

**Line 200. Please explain better what do you mean "represents microbial metabolism processes implicitly"**

Apologies – the use of 'implicit' in this context was not correct. Hopefully the new sentence is clearer.

*L210-217*
*Once assimilated within microbial biomass, the anabolism of microbial activity results in generation of microbial products (i.e., necromass) that form tightly bound aggregates of biofilms and small litter fragments around sand-sized soil particles (Huang et al., 2006; Buks and Kaupenjohann, 2016), and dissolved organic matter (DOM). These contribute to the heavy POM (C5) and litter DOM (C6) pools, respectively. While these specific processes are well supported by relevant literature, to retain parsimony and the generalizable structure required by an ecosystem scale model MEMS v1.0 represents microbial metabolism processes more generally (i.e., by linking them to a dynamic microbial CUE rather than specific community traits).*

**Line 268-269. It could also be, simply, that microbial growth is stimulated and there are more microbes that can also degrade faster the chemically recalcitrant substrates. If I understood correctly, this is not an effect that can be captured by the model without an explicitly microbial pool acting on POM (C5, C10) and MAOM (C8) decomposition.**

As mentioned above, you are right for traditional SOM models. However, because our soil pools are physically-defined with a level of accessibility specific to that pool, our ultimate approach is to modify the parameters of processes for C-mineralization from each pool as the conditions (e.g., nutrient availability, input chemistry, point of entry) change. This would allow the different microbial community traits to be represented for each of the different pools. However, we acknowledge that this is more of a point for the next stages of model development and does not apply to MEMS v1.0.

**Line 270-273. Generally speaking, microbial respiration will be related to microbial activity and CUE. Being not considered microbial activity in the soil, it is not very clear without looking in detail at the Supp. Material how respiration is computed and which fraction of the decomposition is assumed to be. While you refer to $CO_2$ efflux, "respiration" is never mentioned in the Supplementary Material, which is quite surprising.**

We have updated the terminology the refer to C-mineralization as the decomposition process which then results in CO2. We have added an extra sentence to the main text that states that microbial activity and the resulting respiration is computed through decomposition estimates after other processes are calculated, and we refer the reader to the supplementary for more detail. Some information in the supplementary has also been made clearer.

*L281-285*
*Thus, the decay rate constants represent total mass loss potential, embodying DOM-C generation as well as $CO_2$ emissions, as per a recent decomposition conceptualization (Soong et al., 2015). The total amount of heterotrophic respiration is the sum of $CO_2$ produced from the biotic decay of all model pools after other fluxes (e.g., DOM generation) are calculated (more detail can be seen in the Supplementary).*

**Line 281. I would also add that pH controls are quite important. The authors are already well aware of this but neglecting soil moisture controls is a quite significant simplification.**

We are aware and this is key to further development of the model. We have included the mention of pH here now.

*L301-304*
*Simulating the influence of other important controls on decomposition, such as water, oxygen, pH and nutrients, are beyond the scope of this inaugural version of the MEMS model but are central to future development efforts.*

**Line 293. At this stage is not clear how NPP values are derived. Maybe, it is worth to state that this must be an external input to the model. This is actually what mostly separate a "soil organic matter model" from an "ecosystem model".**

We have now added this extra information.

*L312-320*
*Initializing MEMS v1.0 requires external inputs of basic site characteristics (climatic and edaphic conditions as well as land management information) and ideally measurements of daily C input. However, C inputs are rarely available at daily time scales. Consequently, for this inaugural version of the MEMS model we employ a simple function to interpolate daily C inputs from annual Net Primary Productivity (NPP), partitioning aboveground/belowground and to the simulated soil layer using land-use specific root:shoot ratios and a simple root distribution function (Poeplau, 2016). These driving variables are external inputs of the initial model version but may be obtained from coupled climate and plant growth submodels when incorporated into a full ecosystem model. Details of these approaches are given in the supplementary materials and all required driving variables are shown in Table 3.*

**Table 3. The text-box with "site-specific values required" applies to all the site condition variables (e.g., from NPP to soil temperature). This is not clear from the current Table where site-specific values seem to refer to "rock fraction of soil layer" only. I would suggest to use some curly bracket to envelope all these variables.**

This has now been done to the best of our ability given the formatting requirements of the journal. We will ensure this is done and clear for the final typesetting.

**Line 315-319. I am actually quite familiar with the global sensitivity analysis and I think I understood what the authors did. However, I am quite sure that the succinct explanation provided in these lines will remain unclear to most of the readers. I would suggest to either explaining it better (i.e., more extensively) or minimizing the explanation with a full discussion in the supplementary material.**

We have now added further detail to our description in the main text. Hopefully this helps to make our methods clearer to all readers.

*L325-358*
*The default parameter values (i.e., those governing C turnover and fluxes between pools) used by MEMS v1.0 are informed by data from relevant literature (Table 2). However, different studies may suggest different values based on discrete site conditions, meaning a priori estimates may not necessarily be generalizable across all sites that the model could simulate. A variance-based global sensitivity analysis was performed to determine each parameter's relative contribution to the change in each state variable (i.e., determining which parameters have the largest influence on the size of each model pool). The sensitivity analysis was repeated for different simulation lengths (1 – 1000 years) as different fluxes operate at different temporal scales, thereby meaning that the relative importance of each parameter changes through time. Initial pool sizes were set to 0 and the model was initialized to simulate a steady-state scenario based on average site conditions (derived from ~8000 forest and grassland sites in the Land-Use/Land Cover Area Frame Survey (LUCAS) dataset ([Toth et al., 2013] – see Table 3). Specifically, this meant starting a model run with no C in the system and gradually building up the litter and soil pools until they reached equilibrium based on driving variables (soil type, C inputs, climate) that remain fixed over time. To evaluate how much each model parameter (e.g., decay rates, DOM generation rates, etc.; see Table 2) effects the amount of C in each pool (i.e., C1-C11; Figure 1) parameter values were changed to be higher or lower from their baseline and pool sizes are tracked over simulation time. Note that all temperature modifier parameters ($T_{ref}$, $T_{opt}$, $T_{Q10}$, $T_{lag}$ and $T_{shp}$; Table 2) were excluded in this sensitivity analysis as the resulting $T_{mod}$ has the same effect on all decay rates. Maximum and minimum values of all other parameters (n = 24) were defined as 50 % above and below the literature-derived (baseline) value (Table 2). Using Latin Hypercube techniques to sample within the full parameter space, a global sensitivity varying all parameters was used to determine total variance for changes to each model pool (i.e., how much each pool changes in size when all parameters vary up to 50 %). Then, in turn, each individual parameter was fixed at its baseline value while all others varied. This defines each parameter's contribution to a pool's variance, averaged over variations in all other parameters (Sobol, 2001; Saltelli et al., 2008) (i.e., how much each pool changes in size when all parameters, except one, vary up to 50%). When normalized over the global sensitivity variance, a contribution index provides the proportion of variance explained by each parameter. The analysis was run 10,000 times to define the total parameter space and the whole procedure was repeated annually for simulation lengths between 1 to 1000 years. Put simply, 10,000 different combinations of parameter values between the minimums and maximums were used to repeatedly run the model for 1000 years given average site conditions. The results showing changes in pool size correspond to the changes in parameter values (e.g., when maximum decay rate of MAOM is increased, pool C9 may decrease in size but others may increase). The impact that a single parameter has on pool size, compared to that of all parameters, is described by the contribution index, where the total effect of all the parameters is equal to the maximum change in pool size. Note that the results of a global sensitivity analysis of this kind are non-directional and do not indicate whether a parameter increases or decreases a pool size, but rather that it simply changes from the baseline.*

**Line 340. I know that this is probably the only option the authors had, but I hope they are well aware of the limitations of MODIS NPP product; maybe a sentence forewarning the reader would be necessary.**

We are indeed aware of the limitations of using the MODIS NPP estimates. We have also checked a 10-year average of NPP data for each site and noted the variability (and considered redoing the analysis). However, the variability for one site's 10-year average is considerably lower than the variability across Europe and therefore we concluded there was little value in redoing everything, given our limited expectations and reliance on the simulated absolute values.

*L380-384*
*Complimented with geo-referenced estimates of annual NPP from MODIS satellite data (ORNL DAAC, 2009), and daily temperature data from the Climate Prediction Center's Global Temperature (CPC-GT) database (NOAA, 2018), this provided all driving variables required to run MEMS v1.0. The use of modelled/interpolated NPP and climate data is not recommended over measurement data directly collected from the site(s) being simulated, but for the analysis herein these measured data were unavailable.*

**Line 345. The reference Cotrufo et al 2018 explaining the derivation of the POM and MAOM pools is not published. I guess for the sake of this article is fine, but of course, it would be a great contribution to the community if the values of POM and MAOM for the 154 sites would be provided as a part of the LUCAS database or somewhere as part of the article.**

We agree and will make these available as part of this paper submission. The data will available at: http://esdac.jrc.ec.europa.eu/

**Line 368-369. This is probably more a philosophical than a practical point. However, I wonder if a rigorous numerical optimization for such type of models, where the model structure is very uncertain and difference between observed and simulated SOC could be related more to the initialization problem rather than to model structure or parameters is really needed. Given the fact that 4 parameters only were optimized and several replicates were made, this is probably an added value and unlikely a problem here, but still I wonder if is not giving too much weight to the data. How do the results look alike without optimization? This is briefly stated in Line 469-470 but it would actually
be interesting to look at it in more detail.**

The pre- and post-optimized results did not differ significantly for some environmental divisions (e.g., hot, wet, sandy, grasslands) but did for others. We tend to agree with you that our optimization was a little more than what was needed given the early stage of model development, however we wanted to demonstrate how the parameter estimation approach could apply using real measured data. We performed several analyses to assess model performance before and after optimization, but we feel the manuscript already includes a lot of detail and this extra information would be of little value for the majority of readers.

**Line 379. Maybe an explicit statement that optimized parameter values are reported in Table S2 would be useful here.**

We did refer to this table here already but have added an extra reference to hopefully make it clearer.

*L422-426*
*The new, optimized parameter values (Table S2) were derived from a randomly chosen fold that minimized RMSE when compared to the MAOM fraction. This was chosen (instead of those optimized for POM or bulk SOC) since the MAOM fraction is typically the largest single soil C pool and using this approach led to the biggest overall decrease in RMSE when compared to all available data (Table S2).*

**Line 386-387. How seasonal variability in C-inputs and temperature is accounted for? This is not very clear from the manuscript.**

The annual temporal dynamics of C-inputs are derived from a simple distribution function for this first version. We assume a normal distribution around mid-summer so that 75% of the C inputs are added between April and August (Northern Hemisphere). This is a very simplistic way of doing things and is the same for all land use types and locations in our analysis. However, we felt this was more realistic than the same amount every single day. Of course, because we are simulating a steady-state system the resulting difference in effect is minimal but in future versions these C inputs will be coupled to a plant growth model which will be much more accurate.

Regarding seasonal temporal dynamics of temperature, we simply use the daily values for each site and therefore we hope the values used are accurate and account for seasonal variability. This is already stated in the main text.

We have added a little information about this to the main text and point the reader to the supplementary for more detail.

*L430-432*
*Driving variables of edaphic conditions and land-use type were extracted for each site from LUCAS and combined with daily estimates of C inputs and temperature (derived from simple interpolations assuming a normal distribution of MODIS annual NPP data [see Supplementary for details] and CPC-GT daily maximum and minimum air temperature data, respectively). Where these data were unavailable, the site was removed from further evaluation.*

**Line 407. The value for NPP and sand content differ from the mean value provided in Table 3.**

Yes, well noticed. However, the difference comes from the fact that in our methods (line 407) we refer to the median values whereas table 3 states the means. We felt the medians were a better way of describing the overall dataset we were using but the means in table 3 were simply a way of showing the average value – the actual values used in our analysis obviously varied with each site.

**Figure 2. What is the initial condition for the simulation of 1000 years depicted in Figure 2? Do you start from nearly steady state carbon pools or from carbon pools equal to "zero"?**

We start with the carbon pools equal to zero. We did repeat the process with several different starting condition scenarios, but the overall effects were the same. We chose this one simply because it was the easiest to interpret (although we acknowledge that it has so many colours it is still hard to interpret fully). We have added this information to both the new figure legend and in the methods section.

*L332-336*
*Initial pool sizes were set to 0 and the model was initialized to simulate a steady-state scenario based on average site conditions (derived from ~8000 forest and grassland sites in the Land-Use/Land Cover Area Frame Survey (LUCAS) dataset ([Toth et al., 2013] – see Table 3). Specifically, this meant starting a model run with no C in the system and gradually building up the litter and soil pools until they reached equilibrium based on driving variables (soil type, C inputs, climate) that remain fixed over time.*

**Line 455. Why colder temperatures favor POM? Is this related to the sensitivity of decomposition?**

Partially. The main reason why this relationship occurs in MEMS v1.0 is because the MAOM and POM pools reach different equilibrium amounts under different temperatures – under steady-state, MAOM will reach equilibrium at roughly the same amount in all temperatures (i.e., near to the saturation limit), however the POM pools do not saturate and so when temperature is low, decomposition is low, and they will accumulate more before reaching equilibrium. Ultimately you are correct in assuming that temperature is assumed to have a bigger effect on the decomposition of POM than on the decomposition of MAOM (*sensu* Benbi *et al.*, 2014). Our early attempts to differentiate between these sensitivities showed exciting results but were not based on rigorously tested measurements. A key focus of the next stages in model development is in the different sensitivities for the different pools so we hope to include these explicitly in MEMS v2.0.

Benbi D K, Boparai A K, Brar K. 2014. Decomposition of particulate organic matter is more sensitive to temperature than the mineral associated organic matter. *Soil Biol Biochem.***70**: 183–192.

**Line 473-475. Table 2. Maybe I am missing something obvious but the units of decay parameters as "k1" to "k10" should be [gC gC-1 day-1], otherwise when multiplied by the pool (Eq. 1-11 in the supplementary material) you will get [gC^2 day-1] rather than [gC day-1].**

Thanks for pointing out this error. We meant to simply write day$^{-1}$ and this results in the same effect. We have now changed throughout.

**Line 491. This is definitely expected given that variability in litter input, e.g., litter composition and stoichiometry root: shoot ratios are underestimated and soil moisture is not accounted for.**

Agreed. We have now added this extra information.

*L602-610*
*While average agreement between measured and modelled soil C stocks was very good for MEMS v1.0, the model failed to capture the wide range in total POM-C stocks that were observed at the fractionated LUCAS sites (Figure 5). This may be because this first version of the model does not include several of the key controls on POM dynamics, such as water/oxygen limitations (Keiluweit et al., 2016), aggregation (Gentile et al., 2011), activity of soil fauna (Frouz, 2018) and nutrient availability (Bu et al., 2015; Averill and Waring, 2018). There are also limitations of our approach given that very few of the sites will likely be under true steady-state conditions, leading to further discrepancies between model predictions and measured values. Furthermore, the variability in driving variables of litter chemistry, N content and root:shoot ratios are underestimated when using our approach of grouping many different land uses into broad classes.*

**Line 496-497. For almost all of the analyzed sub-groups in terms of site-conditions of Figure 6, bulk SOC observations are mostly between 50-75 MgC/ha. I think this relatively narrow range complicates the identification of the control exerted by temperature, precipitation, soil texture or biomes and therefore also the model testing. A more reasonable test will require more distinguished values of SOC across different conditions, probably using other biomes and climates.**

We agree and accordingly we are currently in the process of fractionating soils from the NEON network of sites that includes a wide range of ecotypes and climates – see https://www.neonscience.org/field-sites/field-sites-map. Once available, it is our hope that these data will help to improve the ability of the MEMS model to simulate a much more diverse set of soils. The relatively narrow range in this analysis of the LUCAS sites results primarily from the very large number of sites. Our initial analysis here was to try and see if general trends looked good and now we are moving on to more site-specific comparisons where we have much higher quality input data.

**Line 521. I don't want to sound too pessimistic and overall I really like the approach of the authors but bridging the gap toward Ecosystem and Earth System Models still requires a considerable amount of work to test the reliability of temporal dynamics and plant-soil feedbacks. This should be stated in the manuscript.**

We agree. We acknowledge the limitations of this early model version and have down-played the point slightly. However, we do feel that the change in approach and model structure can pave the way for an easier link to existing plant growth models and ecosystem models.

*L564-566*
*MEMS v1.0 was designed to consolidate recent advances in our understanding of SOM formation and persistence into a parsimonious mathematical model that uses a generalizable structure which, after further development, can be implemented in Ecosystem and Earth System model applications.*

**Line 552. Also the dynamics of microbial pool in the soil is not explicitly simulated; however, the underestimation of variability is most likely due to underestimation of variability in the inputs and the steady-state assumption in the model, as you wrote in the next few lines.**

We have added to this as shown above. Also, we hope the extra clarification of how microbial activity is simulated implicitly in the soil pools will help to clarify this point.

**Line 558-559. I am not sure why soil moisture controls should be so important at high-latitude, these sites are rarely water limited, I would expect lack of soil-moisture controls to be more important in South-Europe.**

You are right that these sites are not water limited but rather they are water saturated. In these situations, it is possible that anaerobic conditions persist and limit C-mineralisation. You are right that water limitations on the other end of the spectrum (too dry) will be prevalent in Southern Europe, and this is another source of high residuals when we do not include soil moisture controls.

**Line 621-622. This is a great point, and I am looking forward for further work of the authors along this line.**

Thank you. We are also keen to work more towards these goals.

**Figure 1. Just as a suggestion, up to the authors, it would be nice to have some of the parameters of Table 2 represented also in this plot to link the main fluxes to some of the key parameters regulating the flux.**

While we tend to agree that it could be nice to have a single figure with all the information on it, we chose to keep figure 1 as simple as possible so it can serve as a simple way of conveying the overall structure, rather than all the details. We appreciate the suggestion though and will look into including more details on future figures.

**MARKED-UP MANUSCRIPT VERSION BELOW THIS POINT**

[revised manuscript text omitted]

**Full model description of MEMS v1.0**

**Mathematical representation of MEMS v1.0**

Below are the differential equations for dynamics through time as calculated by MEMS v1.0. For simplicity, many of the individual fluxes are summarized by single names (e.g., $C1_{in}^i$ to represent total inputs to the C1 pool from litter material $i$, instead of including the separate calculation). Please refer to the equations provided in this Supplementary

Materials. Parameter descriptions can be found in Table 2 of the main manuscript. Please note that the below list equations are fully representative of the carbon dynamics of MEMS v1.0 but are layer- and time-specific. However, for simplicity are presented in a generalized form.

$$\frac{dC1}{dt} = C1_{in}^i - (uk * C1 * k_1) \tag{1}$$

$$\frac{dC2}{dt} = C2_{in}^i - (uk * C2 * k_2) - (C2 * LIT_{frg}) \tag{2}$$

$$\frac{dC3}{dt} = C3_{in}^i - (C3 * k_3) - (C3 * LIT_{frg}) \tag{3}$$

$$\frac{dC4}{dt} = C4_{ass}^{C1} + C4_{ass}^{C2} - (C4 * k_4) \tag{4}$$

$$\frac{dC5}{dt} = C5_{gen}^{C4} + C5_{frg}^{C2} + C5_{frg}^{C3} - (C5 * k_5) \tag{5}$$

$$\frac{dC6}{dt} = C6_{in}^i + C6_{in}^{C1} + C6_{in}^{C2} + C6_{in}^{C3} + C6_{in}^{C4} - C8_{in}^{C6} \tag{6}$$

$$\frac{dC7}{dt} = C1_{co2} + C2_{co2} + C3_{co2} + C4_{co2} + C5_{co2} + C8_{co2} + C9_{co2} + C10_{co2} \tag{7}$$

$$\frac{dC8}{dt} = C8_{in}^{C5} + C8_{in}^{C6} + C8_{in}^{C10} - sorption - (C8 * DOC_{lch}) - (C8 * k_8) \tag{8}$$

$$\frac{dC9}{dt} = sorption - (C9 * k_9) \tag{9}$$

$$\frac{dC10}{dt} = C10_{frg}^{C2} + C10_{frg}^{C3} - (C10 * k_{10}) \tag{10}$$

$$\frac{dC11}{dt} = (C8 * DOC_{lch}) \tag{11}$$

**Carbon inputs from external sources**

In MEMS v1.0 the above- and below-ground plant residue inputs are combined and input to the system on a daily timestep. These total inputs are partitioned between C1, C2, C3 and C6 as a function of the external source ($i$) input properties (Eqs. 12-15): the cold water extractable fraction of the hot-water extractable litter input ($f_{DOC}^i$), the hot water extractable fraction of the litter input ($f_{SOL}^i$) and acid-insoluble fraction of the litter input ($f_{LIG}^i$).

$$_j^L C1_{in}^i = \left(_j^L CT^i * f_{SOL}^i\right) - \left(_j^L CT^i * f_{SOL}^i * f_{DOC}^i\right) \tag{12}$$

$$_j^L C2_{in}^i = _j^L CT^i - \left(_j^L CT^i * \left(f_{SOL}^i + f_{LIG}^i\right)\right) \tag{13}$$

$$_j^L C3_{in}^i = \left(_j^L CT^i * f_{LIG}^i\right) \tag{14}$$

$$\frac{L}{j}C6_{in}^i = \frac{L}{j}CT^i * f_{SOL}^i * f_{DOC}^i \qquad (15)$$

Where $\frac{L}{j}X_{in}^i$ is refers to the daily carbon input to pool $X$ from external source $i$  on day $j$, and $\frac{L}{j}CT^i$ is the total daily carbon input from external source $i$  on day $j$. For MEMS v1.0 the layer is fixed to the aboveground litter layer only, allowing for use of the same functions as those presenting in the LIDEL model (Campbell *et al*., 2016). However, future versions may incorporate the same structure for different points of entry for C inputs (e.g., root death and the rhizosphere).

Once allocated to their initial pools, the carbon is susceptible to assimilation in microbial biomass if it is water-soluble (C1) or acid-soluble (C2) but only co-metabolized if it is acid-insoluble (C3). The contents of these pools represent compounds of increasing chemical complexity (e.g., C1, mostly soluble carbohydrates, phenols and amino acids; C2, mostly cellulose, xylans and other hemicelluloses; C3, mostly lignin aboveground and suberin/cutin belowground) and are associated with decreasing microbial use efficiency.

**Microbial assimilation from litter pools**

Many of the biogeochemical processes represented by MEMS are assumed to be microbially mediated, and therefore are associated with C-mineralization and the resulting carbon dioxide ($CO_2$) emissions from microbial respiration. The primary carbon losses  result from the metabolic processes of bacteria and fungi within the soil and are aligned with the mathematical representations as described by Campbell *et al*. (2016) and, in part, summarise the findings of Sinsabaugh *et al*. (2013), Moorhead *et al*. (2013) and Soong *et al*. (2015). In addition, carbon assimilation  by microbial biomass (C4) in the litter layer results from the balance between anabolic and catabolic processes and thus, as biomass is formed, dissolved organic matter (DOM) and $CO_2$ are also produced . Microbial assimilation is a function of nitrogen content and lignocellulosic index (Eq. 16) of the structural litter pools (C2 and C3; organic matter > 2 mm)  and controlled by maximum decomposition rates for C1 ($k_1$) and C2 ($k_2$) that assume first-order decay.

$$\frac{L}{j}LCI_{lit} = \frac{\frac{L}{j}C3}{\left(\frac{L}{j}C2 + \frac{L}{j}C3\right)} \qquad (16)$$

$$\frac{L}{j}C4_{ass}^{C1} = uB * B_1 * (1 - la_4) * uk * k_1 * \frac{L}{j}C1 \qquad (17)$$

$$\frac{L}{j}C4_{ass}^{C2} = uB * B_2 * (1 - la_1) * uk * k_2 * \frac{L}{j}C2 \qquad (18)$$

Where $\frac{L}{j}C4_{ass}^{C1}$ and $\frac{L}{j}C4_{ass}^{C2}$ refer to the fraction of the given litter pool (i.e., C1 or C2) that is microbially assimilated to pool C4  on day $j$ from pool C1 or C2, respectively. Note that these functions are make microbial assimilation explicit in this  aboveground litter layer. In the soil itself, microbial assimilation of organic matter is still occurring but assumed to be implicit and incorporated in the carbon mineralization rates for each of the soil pools (e.g., C5, C8, C9 and C10). In future versions of the model, the same general structure can apply, with an explicit microbial component at the different  points of entry (i.e., rhizospheric inputs vs aboveground litter) but  parameter values may differ between layers, when more are added.

Detail about the concepts behind this approach can be found in Sokol *et al*., 2018.

More information of the parameters $uB$, $uk$, $B_x$, $la_x$ and $k_x$ can be found in Campbell *et al.* (2016) and in the equations below , but briefly:

• $_j^L uB$ and $_j^L uk$ are rate modifiers to represent the litter chemistry controls (LCI and available nitrogen) on microbial use efficiency,  on day $j$.

$$_j^L uB = min\left(\left(\frac{1}{1+e^{-N_{max}(N_{lit}-N_{mid})}}\right),\left(1-e^{-0.7\left(\left|_j^L LCI_{lit}-0.7\right|*10\right)}\right)\right) \quad \textbf{(19)}$$

$$_j^L uk = min\left(\left(\frac{1}{1+e^{-N_{max}(N_{lit}-N_{mid})}}\right),\left(e^{-3*_j^L LCI_{lit}}\right)\right) \quad \textbf{(20)}$$

Where $N_{max}$ and $N_{mid}$ are maximum and mid points of litter nitrogen content having an impact on microbial use efficiencies, using a logistic curve (see Figure S7). $N_{lit}$ and $_j^L LCI_{lit}$ are the input material nitrogen content and LCI

being simulated on day $j$.

IMPORTANT NOTE – In MEMS v1.0 there is no nitrogen cycling and therefore the $N_{lit}$ value is not dynamic, as it likely should be. Consequently, MEMS v1.0 uses the nitrogen content of the input material, and therefore $N_{lit}$ is a constant through time and across layers. This constant nitrogen value is consistent with the approach used by the

LIDEL model (Campbell *et al*., 2016) however it is expected that a dynamic nitrogen (i.e. be $_j^L N_{lit}$ – as equivalent to

$_j^L LCI_{lit}$) content would more likely reflect real-world conditions, especially in extended periods without litter input.

• $B_1$ and $B_2$ are maximum growth efficiencies associated with the water-soluble and acid-soluble litter pools (C1 and C2), respectively (See Table 2 in the main manuscript).

• $la_1$ and $la_4$ are estimates of carbon in DOM generation from leaching the decayed litter pools  on day $j$.

$$_j^L la_1 = min\left(\left(E_{Hmax}-\frac{(E_{Hmax}-E_{Hmin})}{LCI_{max}}*_j^L LCI_{lit}\right),\left(E_{Hmax}-\frac{(E_{Hmax}-E_{Hmin})}{N_{max}}*N_{lit}\right)\right) \quad \textbf{(21)}$$

$$_j^L la_4 = min\left(\left(E_{Smax}-\frac{(E_{Smax}-E_{Smin})}{LCI_{max}}*_j^L LCI_{lit}\right),\left(E_{Smax}-\frac{(E_{Smax}-E_{Smin})}{N_{max}}*N_{lit}\right)\right) \quad \textbf{(22)}$$

Where $E_{Hmax}$ and $E_{Hmin}$ are the maximum and minimum amount of DOM leached from decay of acid-soluble litter (C2), and $E_{Smax}$ and $E_{Smin}$ are the maximum and minimum amount of DOM leached from decay of water-soluble litter (C1). $LCI_{max}$ refers to the maximum lignocellulosic index that can have an impact on these rates. As noted above, $N_{lit}$ and $_j^L LCI_{lit}$ are the nitrogen content of input material and LCI  being simulated on day $j$.

•   $k_1$ and $k_2$ are the maximum decay rates of water-soluble (C1) and acid-soluble (C2) litter pools, respectively (See Table 2 in the main manuscript).

**Microbial mortality and necromass production**

After carbon is metabolized by microbes and incorporated in pool C4, the death and products of microbial activity result in the compounds that form the coarse, heavy particulate SOM (C5) that is often found coating sand particles in the > 53 µm soil fraction (Ludwig *et al.*, 2015). In the aboveground litter layer simulated by MEMS v1.0, this process of microbial biomass decay results in loss to DOC (C6) and $CO_2$ (C7), in addition to the C5 pool belowground.

$$^{L}_{j}C5^{C4}_{gen} = B_3 * (1 - la_2) * k_4 * ^{L}_{j}C4 \tag{23}$$

Where $^{L}_{j}C5^{C4}_{gen}$ refers to the fraction of carbon that is transferred from C4 to C5 (i.e., microbial products transported belowground when physical and hydrological processes mix between the input layer [aboveground litter only in

MEMS v1.0] and soil layer)

on day *j*.

The flux from the aboveground microbial biomass pool (C4) is assumed to move belowground, to the first soil layer (see Figure 1 in the main manuscript). More information of the parameters $B_3$, $la_2$

and $k_4$ can be found in Table 2 in the main manuscript, but briefly, $B_3$ refers to a maximum rate of microbial product (C5) generation per unit of microbial biomass (C4) decayed, $la_2$ refers to the maximum amount of DOM produced per unit of microbial biomass (C4) decayed and $k_4$ refers to the maximum rate of microbial biomass (C4) decay.

**Fragmentation and perturbation**

To quantify the transfer of carbon from large (> 2 mm) particulates to small particulates belowground, simple parameter values have been allocated to represent first-order rates of transfer from both structural litter pools (C2 and

C3). As model development continues, these rates will be improved to provide more mechanistic relationships with site conditions (see Braakehekke *et al.*, 2011). See Table 2 for information about the parameter used in MEMS v1.0

($LIT_{frg}$). The amount of litter C fragmented and transferred vertically from structural litter pools to the belowground

POM pools (C5 and C10) is also governed by the $POM_{split}$ parameter that defines how much of the total is allocated to C5.

$$^{L}_{j}C5^{C2}_{frg} = POM_{split} * LIT_{frg} * ^{L}_{j}C2 \tag{24}$$

$$^{L}_{j}C5^{C3}_{frg} = POM_{split} * LIT_{frg} * ^{L}_{j}C3 \tag{25}$$

$$^{L}_{j}C10^{C2}_{frg} = (1 - POM_{split}) * LIT_{frg} * ^{L}_{j}C2 \tag{26}$$

$$^{L}_{j}C10^{C3}_{frg} = (1 - POM_{split}) * LIT_{frg} * ^{L}_{j}C3 \tag{27}$$

Where $^{L}_{j}CX^{CY}_{frg}$ refers to the amount of carbon that is transferred from pool *CY* to pool *CX*  on day *j*.

**Dissolved organic matter production**

Dissolved organic matter plays a major role in the MEMS model as it is the only way in which carbon can sorb to mineral surfaces in the soil, meaning that if there is limited DOM there will also be limited stabilization in MAOM (C9). Consequently, DOM production from all model pools is simulated explicitly according to the formulae provided by the LIDEL model (Campbell *et al.*, 2016) and based on empirical data in Soong *et al.* (2015). Each timestep, the aboveground litter layer DOM (C6) receives a fraction of inputs from external sources directly (Eq. 15; $_{j}^{L}C6_{in}^{i}$), from all litter layer pools ($_{j}^{L}C6_{in}^{C1}$, $_{j}^{L}C6_{in}^{C2}$, $_{j}^{L}C6_{in}^{C3}$) and from microbial biomass ($_{j}^{L}C6_{in}^{C4}$).

$$_{j}^{L}C6_{in}^{C1} = la_4 * uk * k_1 * {_{j}^{L}C1} \tag{28}$$

$$_{j}^{L}C6_{in}^{C2} = la_1 * uk * k_2 * {_{j}^{L}C2} \tag{29}$$

$$_{j}^{L}C6_{in}^{C3} = la_3 * k_3 * {_{j}^{L}C3} \tag{30}$$

$$_{j}^{L}C6_{in}^{C4} = la_2 * k_4 * {_{j}^{L}C4} \tag{31}$$

Where $_{j}^{L}Cx_{in}^{Cy}$ refers to DOM leaching from pool *y* to pool *x*  on day *j*. The parameters used are detailed in Table 2 in the main manuscript, and/or defined in previous equation in this section. Note that pool C6 is not the DOM consumed by microbial biomass but rather the amount leftover after microbial activity. In this initial model version, the litter layer only refers to the aboveground component, but the same structure can equally apply to belowground C inputs such as root death.  However, measurably, the DOM in the C6 pool  is directly equivalent to the belowground soil DOM (C8). In MEMS v1.0, DOM enters the soil through the C6 pool only.  When explicit inputs from belowground litter (e.g., roots) are simulated in future versions Eqs. 28-31 can apply for each soil layer adding the DOM that is in excess of microbial activity directly to pool C8 instead of the 'C6' shown in the equations above. Similarly, root exudates can be simulated as direct addition to the C8 pool of any specific soil layer. Hence, just as the litter layer DOM (C6) receives inputs from the aboveground litter layer pools, the soil DOM (C8) would receive inputs from the belowground pools (e.g., decomposing root matter and root exudation). In addition, the soil DOM pool receives inputs from the POM and MAOM pools ($_{j}^{L}C8_{in}^{C5}$, $_{j}^{L}sorption$, $_{j}^{L}C8_{in}^{C10}$) as well as from leached litter DOM (C6). Here, the *sorption* flux represents the net carbon exchange between soil DOM (C8) and MAOM (C9).

$$_{j}^{L}C8_{in}^{C5} = la_3 * k_5 * {_{j}^{L}C5} \tag{32}$$

$$_{j}^{L}C8_{in}^{C6} = DOC_{frg} * {_{j}^{L}C6} \tag{33}$$

$$_{j}^{L}C8_{in}^{C10} = la_3 * k_{10} * {_{j}^{L}C10} \tag{34}$$

The parameter values are defined in Table 2 in the main manuscript. As with the $LIT_{frg}$ parameter, the $DOC_{frg}$ value in MEMS v1.0 is set as a tuning parameter and simply assumes first-order rates to allocate a given proportion of the carbon in litter layer DOM pool (C6) to the soil DOM pool (C8) each timestep. As noted earlier, these functions are layer-specific and therefore in a multi-layer version of MEMS, there would be vertical leaching of DOM between C8

pool of different layers, instead of from the aboveground C6 pool alone (i.e., to replace Eq. 33).

**Sorption and desorption**

The formation of organo-mineral complexes in MEMS v1.0 is represented by a net sorption-desorption process that uses the amount of soil DOM (C8) to estimate adsorption rates based on a Langmuir isotherm (Kothawala *et al.*,

2008). The key elements of this isotherm are the 'binding affinity' ($K_{lm}$) – see Eq. 35 – and maximum sorption capacity ($Q_{max}$) – see Eq. 36 – which are controlled by site-specific conditions (soil pH and soil texture, respectively).

It is worth noting that each of these site-specific conditions are provided as driving variables to the model, and are constants that represent the site at time-zero (i.e., soil pH is not simulated to change through time). The net sorption rate ($sorption$) aims to account for several different sorption mechanisms (e.g., cation bridging, surface complexation, etc.) to retain parsimony. A more accurate net flux may simulate the different mechanisms individually to allow for more detailed representation of different mineralogies as per Six *et al.* (2002) (e.g., dominated by 2:1

clays *vs* 1:1 clays). Future development of MEMS may adopt these changes.

$$^{L}K_{lm} = 10^{\left(-0.186\ ^{L}soilpH - 0.216\right)} \tag{35}$$

Where $^{L}soilpH$ refs to the 'native' soil pH of  simulated soil . The soil pH, as used in Eq 35, acts as a proxy for mineralogical differences between soils, with higher native soil pH being equated with weaker chemical bonding. This tenet is adopted from the regression provided in Mayes *et al.* (2012) and results in $K_{lm}$ being estimated as in the MILLENNIAL model (Abramoff *et al.*, 2017). However, the MEMS v1.0 estimate of $Q_{max}$ does not follow the MILLENNIAL model and instead calculates a general relationship between maximum soil carbon capacity and soil texture using the entire dataset of Six *et al.* (2002). This takes a simple linear regression approach using the soil layer's percent silt and clay content (i.e., $100 - sand$)

$$^{L}Q_{max} = {}^{L}\rho * (0.26126 * (100 - {}^{L}sand) + 11.07820) * (1 - {}^{L}rock) \tag{36}$$

Where $^{L}\rho$ refers to the bulk density of the soil  at the site being simulated. Note that the bulk density is a conversion specific to the depth of the soil layer that converts a concentration from the regression of Six *et al.* (2002)

to carbon density (e.g., gC m$^{-2}$ layer depth$^{-1}$) and therefore the equations shown here assume a 1 meter deep layer for simplification. Both the sand content ($^{L}sand$) and rock fraction ($^{L}rock$) are expressed in percent (i.e., 0-100)

. The resulting equation to represent net sorption is controlled by a Langmuir saturation function, using the amount of soil DOC (C8) available for sorpt_______ion as well as the saturation deficit of MAOM (C9).

Note, all coefficients in the equation below are layer- and timestep-specific.

$$^{L}_{j}sorption = {}^{L}_{j}C8 * \frac{\left(\left(\frac{\left(^{L}K_{lm}*^{L}Q_{max}*^{L}_{j}C8\right)}{1+\left(^{L}_{j}K_{lm}*^{L}_{j}C8\right)}\right) - ^{L}_{j}C9\right)}{^{L}Q_{max}} \tag{37}$$

Where $\frac{L}{j}sorption$ is a net exchange of carbon between the soil DOM (C8) and MAOM (C9) pools  given their size on day $j$. Since $K_{lm}$ and $Q_{max}$ are site-specific parameters, and the pool sizes (C8 and C9) are dynamic through time, there are interactions between these factors which mean sorption rates are not necessarily comparable between sites. This sorption process is assumed to be abiotic in that it results in no $CO_2$ emitted. As a net rate, sorption and desorption are not simulated individually which may make it difficult to represent potential priming effects on organo-mineral associations (e.g., Keiluweit *et al*., 2015). Future MEMS model version will explore these feedbacks further.

**Decomposition and pool decay rates**

Apart from the litter layer DOM (C6), each of the state variables in MEMS v1.0 decay directly with unique decay rates informed by literature values (see Table 2). This decay results in $CO_2$ emissions which continually accumulate in the sink C7. The amount of $CO_2$ associated with each microbial process is equivalent to the amount of carbon leftover after losses to DOM are calculated so the decay rate constants for pool $x$ ($k_x$) also embody explicit DOM

generation and not just $CO_2$ emissions, as is more common in traditional SOM models (e.g., CENTURY or RothC).

As with earlier equations, these below  can be layer- and time-specific but for simplicity are presented in a generalized form.

$$C1_{co2} = \left( \left(1 - (uB * B_1)\right) * (1 - la_4)\right) * uk * k_1 * C1 \tag{38}$$

$$C2_{co2} = \left( \left(1 - (uB * B_2)\right) * (1 - la_1)\right) * uk * k_2 * C2 \tag{39}$$

$$C3_{co2} = (1 - la_3) * k_3 * C3 \tag{40}$$

$$C4_{co2} = \left((1 - B_3) * (1 - la_2)\right) * k_4 * C4 \tag{41}$$

$$C5_{co2} = (1 - la_3) * k_5 * C5 \tag{42}$$

$$C8_{co2} = k_8 * C8 \tag{43}$$

$$C9_{co2} = k_9 * C9 \tag{44}$$

$$C10_{co2} = (1 - la_3) * k_3 * C10 \tag{45}$$

Where all parameters are defined in Table 2 in the main manuscript and earlier in this section. While the maximum decay rates ($k_x$) for most pools are fixed constants, Campbell *et al*. (2016) suggested that $k_3$ is best estimated in relation to the maximum decay rate of the microbially-accessible litter (C2) pool ($k_2$).

$$\frac{L}{j}k_3 = k_2 * \left( \frac{0.2}{1 + \frac{200}{e^{8.15 * \frac{L}{j}LCI_{lit}}}} \right) \tag{46}$$

$$k_8 = \frac{\left( \left((0.000099) * \left(\frac{1}{100}\right)\right) + \left((0.000855) * \left(\frac{1}{42}\right)\right) + \left((0.001796) * \left(\frac{1}{13}\right)\right) \right)}{sum\left( \left(\frac{1}{100}\right), \left(\frac{1}{42}\right), \left(\frac{1}{13}\right) \right)} \tag{47}$$

Note that when $k_2$ is a fixed value, $k_3$ only fluctuates with changes in the LCI of the litter layer.
Also note that because the maximum decay rate of acid-insoluble litter ($k_3$) is determined relative to the
LCI of all litter pools  on a given day (*j*) the parameter itself can also be layer- and time-specific.
At present, $CO_2$ emitted from soil DOM (determined by the maximum decay rate, $k_8$) is associated with the values
presented in Kalbitz *et al.* (2005).

$$k_8 = \frac{\left(\left((0.000099)*\left(\frac{1}{100}\right)\right)+\left((0.000855)*\left(\frac{1}{42}\right)\right)+\left((0.001796)*\left(\frac{1}{13}\right)\right)\right)}{sum\left(\left(\frac{1}{100}\right),\left(\frac{1}{42}\right),\left(\frac{1}{13}\right)\right)} \tag{47}$$

**Decay rate modifiers**

Soil temperature is simulated to have a polynomial relationship with decomposition, modifying each pool's decay
rate according to the mean soil temperature of that layer on that day. The rationale behind this is to attempt to capture
microbial processes and equate with realistic changes in enzymatic activity to be consistent with Michaelis-Menten
kinetics. This follows the same function that is used by the STANDCARB 2.0 model (Harmon and Domingo, 2001)
and produces a multiplier based on provided coefficients of optimum decomposition temperature ($T_{opt}$), the rate at
which the decomposition rate increases with a 10 °C increase ($T_{Q10}$), the reference temperature at which that $Q_{10}$ value
was derived ($T_{ref}$), the shape of the excessive temperature limitation ($T_{shp}$) and the difference between optimum
temperature and the decline above that threshold ($T_{lag}$).

$$_{j}^{L}T_{mod} = e^{\left(-\left(\frac{_{j}^{L}soilT}{T_{opt}+T_{lag}}\right)\right)^{T_{shp}}} * T_{Q10}^{\frac{_{j}^{L}soilT-T_{ref}}{T_{ref}}} \tag{48}$$

Where $_{j}^{L}T_{mod}$ is the temperature multiplier applied to decomposition of pools  on day *j*, given the soil
temperature  on that day ($_{j}^{L}soilT$). An initial MEMS v1.0 evaluation (prior to use with the LUCAS sites
reported in the main manuscript), indicated the model consistently overestimated decomposition due to the
temperature modifier effect. Consequently, the coefficients reported in Harmon and Domingo (2001) were revised
down from those reported in Table 2 of the main manuscript ($T_{opt}$ reduced to 35 °C, $T_{shp}$ reduced to 3, $T_{lag}$ increased
to 7 °C and $T_{Q10}$ increased to 3). In MEMS v1.0 this single function is used for all pools and over the single soil layer,
however, it is also sufficiently generalizable to represent varying temperature sensitivities of the different pools (i.e.,
through the $T_{Q10}$ coefficient) and of different layers. In which case, the temperature modifier would be specific to
pool *x*  on day *j* – e.g. $_{j}^{L}T_{mod}^{x}$. Furthermore, in future versions of the MEMS model, we expect more explicit
and complex relationships to temperature and moisture.

**DOM transfer through soil layers**

MEMS v1.0 does not have an explicit hydrological model, however this is likely needed for MEMS outputs to be reliably compared with empirical data at most sites (soil moisture often has a considerable influence on SOM formation and decomposition rates). Consequently, this is one of the first developments intended for MEMS. As a placeholder, leaching is assumed to be a unidirectional process with DOM lost to deeper soil layers (in the single-layer version) at a given maximum rate. This follows a first order rate of loss and simply assumes half the highest literature value found when performing a search of relevant studies.

**Driving variables and initializing MEMS v1.0**

**Site inputs and interpolating daily values from annual measurements**

Driving variables of MEMS v1.0 can be either provided manually if they are known, or interpolated/estimated using basic site information. The format of this input information is typically in comma separated values (CSV) or any other ASCII text format and in R (R Core Team, 2018) is stored as a dataframe. As a single-layer, carbon model that only simulates litter and soil components of a site, MEMS v1.0 includes only a few *essential* driving variables. These fall into three major categories (climatic, edaphic and land use). For convenience, a summary of these essential inputs is provided in Table 3 of the main manuscript. The model operates on the assumption that a user must have measurements of soil pH, soil bulk density, annual NPP, sand content and rock fraction in order to simulate the site. Additionally, if daily temperature data are not known, the maximum, minimum and mean annual temperature can be used to interpolate daily values.

At the time of writing, daily soil temperature is the only climatic variable simulated in MEMS v1.0. The model can either be initialized using real, site-specific temperature data (if available), or daily values can be roughly estimated using a simple sine function related to the mean annual temperature (MAT) of the site (Eq. 49). This sine function provides 365 days of temperature values that are normally distributed around the MAT (therefore ensuring that the average from these daily values will also equal the MAT provided), with the peak of this sine on Julian day 182 (July 1st). This assumes the site is in the northern hemisphere but simulating a site in the southern hemisphere simply requires changing the sign of the 1.5 coefficient in Equation 49 below.

$$\frac{L}{j}soilT = \frac{T_{range}}{2} * sin\big((2 * PIseq) - 1.5\big) + MAT \tag{49}$$

Where $\frac{L}{j}soilT$ is the soil temperature in degrees Celsius  on day *j*, $T_{range}$ is the difference between the maximum daily soil temperature and minimum daily soil temperature measured over a year in degrees Celsius, PIseq is a sequence of 365 values evenly distributed from 0 to pi ($\approx 3.14159$), and MAT is the mean annual temperature in degrees Celsius of the site in question. While this approximation provides more realistic inputs than a constant temperature for each day, where possible, real, measured values should be imported separately as a list of average daily soil temperature values.

It should be noted that this sine function (with an intra-annual variation of $T_{range}$ degrees Celsius) may not work well for sites near the equator where reduced seasonal dynamics mean that a smoothed sine curve does not represent reality.

The $T_{range}$ coefficient in Equation 49 is ideally calculated from estimates/measurements of a site's maximum and minimum soil temperatures of an average year, included alongside the MAT as inputs. However, these are optional and instead, a constant $T_{range}$ value (i.e., the same range at all sites simulated) can be set as a global parameter as shown in Table 2 in the main manuscript. This should be chosen carefully by the model user to best represent their site(s). It should also be noted that when simulating deeper soil layers they are also less likely to see large fluctuations in soil temperature and this should be considered when the user initializes multi-layer versions of the MEMS model.

**Land use and management conditions**

As with the sine function estimate soil temperature, the daily carbon inputs ($_jCT^i$) can also be estimated crudely according to a simplistic relationship with annual net primary productivity (NPP) – Equation 50).

$$_jCT^i = dnorm(seqDAY, peakDAY, sdNPP) * annNPP \tag{50}$$

Where $_jCT^i$ are the daily total carbon inputs from material $i$ on day $j$, $seqDAY$ is a list of 365 integers that represent each day of the year, $peakDAY$ is a parameter value to specify the julian day of year when inputs peak (around which a normal distribution is generated) and $sdNPP$ is the 'width' of the distribution around the peak value. The $annNPP$

value is the site-specific annual NPP value in gC m$^{-2}$ yr$^{-1}$. The $sdNPP$ parameter (specified as a global parameter) can be modified to represent different intra-annual distributions of the total carbon inputs. Specifically, this can change how 'quickly' the inputs are added to the soil (is the whole carbon input added within a few days or is it spread out over months?). For different land uses, $sdNPP$ may change according to the trends in plant growth at a given site.

However, when simulating an equilibrium scenario where steady-state inputs are assumed, this has little or no effect over long simulations (i.e., 500+ years).

In most systems the total annual NPP is not directly equivalent to the total carbon inputs to the topsoil layer.

Consequently, MEMS v1.0 reduces the annual amount based on how much of the total can be realistically expected to be input to the specific layer given that site's land use. For example, Bolinder et al. (2007) suggest that, in arable sites where all residues are returned to soil, the proportion of annual NPP that is input to all soil varies between 55%

and 78%. Whereas when all residues are removed, the proportion input can be as little as 21%. Furthermore, not all of this will be input to the topsoil layer simulated by MEMS v1.0. Consequently, before the daily inputs are interpolated from an annual value using Equation 50, the total is reduced based on best estimates for the land use and management routines of the site simulated.

$$_jaCT^i = {}_jCT^i * \left(\frac{1}{RtoS^i+1}\right) * (1 - {}_jaHARV^i) \tag{51}$$

$$_jbCT^i = {}_jCT^i * \left(\frac{RtoS^i}{RtoS^i+1}\right) * (1 - {}_j^LbHARV^i) \tag{52}$$

Where $_jaCT^i$ and $_jbCT^i$ are the aboveground and belowground carbon inputs of material $i$ on day $j$. The aboveground and belowground split is achieved by use of a land-use specific root to shoot ratio of material $i$ ($RtoS^i$) which are then reduced by fixed fractions (i.e., 0-1) to represent any losses through harvesting. Another parameter to describe natural losses due to weather (e.g., high winds) is also possible and resides as a placeholder in the general crop parameters file of MEMS v1.0. After the realistic aboveground fraction of NPP is derived, it can then replace the $_jCT^i$ term in

Equation 50 and be used to interpolate daily inputs. However, the belowground fractions of NPP also includes inputs that are likely allocated to deeper soil layers than the topsoil simulated by MEMS v1.0. Consequently, the $_jbCT^i$ as calculated in Equation 52 is reduced by use of a Michaelis-Menten style function (see Kätterer *et al.*, 2011) to proportion roots to the simulated soil layer.

$$_j^L bCT^i = \,_jbCT^i * \left( \frac{^L depth * (Rdep_{50} + Rdep_{max})}{Rdep_{max} * (Rdep_{50} + {}^L depth)} \right) \tag{53}$$

Where $_j^L bCT^i$ is the belowground carbon input of material $i$  on day $j$, $^L depth$ is the depth of the soil in centimetres, $Rdep_{50}$ is the soil depth from the surface at which 50 % of the root biomass is proportioned in centimeters, and $Rdep_{max}$ is the maximum rooting depth in centimeters. These last two parameters are site specific but can be generalized according to different land-uses, reducing the number of inputs required by the model user.

For information regarding these generalized parameters, see Canadell *et al.* (1996) and Jackson *et al.* (1996). For an example implementation of Equation 53 for the purpose of simulating SOM dynamics, see Poeplau (2016).

As with the interpolation of daily soil temperature from MAT, estimating daily values of carbon input are less precise than using real measured data. When possible, empirical data should be preferred and can be input along with daily climate data.

(see attached files for high-resolution versions)

**Figure S1** – Site information of all 8192 forest and grassland sites of the LUCAS dataset (Toth *et al*., 2013) used for validation of the MEMS v1.0 soil organic matter model. Different shapes represent different land use classes and all are overlaid over each other (grass = circles, *n* = 3487; broadleaved forests = triangle, *n* = 1590; mixed forest = crosses, *n* = 1402; coniferous forest = squares, *n* = 1713).

[Figure]

**Figure S2** - Geographical distribution of 154 grassland and forest sites chosen for fractionation (a representative subsample of the total LUCAS database, see Toth *et al*., 2013). Reported mean annual temperature, mean annual precipitation and sand content are indicated for each site along with Net Primary Productivity (NPP) in 2009 derived from MODIS. Symbols indicate the land

 use division within grassland and forest. Cin is the C input, MAP is the mean annual precipitation
and MAT is the mean annual temperature.

**Figure S3** - Summary statistics of the site information and soil C stocks for four land use classes (Grassland, n=78; Broadleaved forest, n=25; Coniferous forest, n=27; Mixed forest, n=24) across Europe. Boxplots indicate the median, first and third quartiles with the box and maximum and minimum at the extent of the whiskers. Outliers beyond the 95% are shown by individual points. MAT = Mean Annual Temperature; MAP = Mean Annual Precipitation; NPP = Net Primary Productivity; SOC = Soil Organic Carbon; POM = Particulate Organic Matter; MAOM = Mineral-Associated Organic Matter.

[Figure]

**Figure S4** - One-way ANOVA results with pairwise comparisons for each measured fractionation data (bulk soil C stock, mineral-associated organic matter (MAOM) C stock, particulate organic matter (POM) C stock, and the MAOM:POM ratio) between the four land use classes (Grassland, n=78; Broadleaved forest, n=25; Coniferous forest, n=27; Mixed forest, n=24) of topsoils (0-20 cm) from 154 sites across Europe.

Significant differences indicated by p-values for each pair (p < 0.001, red; p < 0.01, orange; p < 0.05, yellow; p < 0.1, green; p > 0.1, blue). NPP = Net Primary Productivity.

[Figure]

**Figure S5** – Fully-colourised version of main text Figure 2. Global sensitivity analysis results showing the relative contribution of each parameter to a change in carbon stock of each pool in MEMS v1.0 (leached carbon to deeper soil layers [pool C11] is omitted for clarity). Details of each parameter and the abbreviations used can be found in Table 2. The sensitivity analysis was repeated annually for simulation times between 1 and 100 years, every 10 years after that to 400-year simulations and every 100 years after that up to a 1000-year simulation. Results are presented on a log scale in years. Parameters involved in different SOM formation processes are grouped by colour: yellows – parameters that define DOM

 leaching from the organic horizon to the soil layer; reds – parameters that affect microbial carbon use efficiency, purples – parameters that affect
organic matter vertical transport to deeper layers, greens – maximum decay rates.

[Figure]

**Figure S6** – Variability in model-data residuals compared with mean annual temperature for 8192 forest and grassland sites of the LUCAS dataset
(Toth *et al.*, 2013) simulated with the MEMS v1.0 soil organic matter model. Residuals indicate the modelled minus measured total topsoil (0-20
cm) organic carbon stock in MgC ha[-1] for each of four land-use classes (Grassland, red; Broadleaved forest, blue; Coniferous forest, purple; Mixed
forest, green). Sites are divided into high and low groups of mean annual precipitation, MAP (top *vs* bottom panels), soil texture (left *vs* right
panels) and annual carbon inputs (provided by net primary productivity, NPP) (alternating panels left to right).

[Figure]

**Figure S75** - Modifiers for microbial carbon use efficiency and rates of water-soluble and acid-soluble litter fractions decay by lignocellulosic index (A and B) and initial litter percent nitrogen (C). Reproduced with permission from Campbell *et al.*, 2016.

[Figure]

**Table S1** - Fractionation scheme to measure each OM pool of MEMS v1.0. Physical particle size is given sequentially from top to bottom (i.e. C9 pools are between 0.45 μm and 53 μm in size). Soil particles ($< 2mm$) are primary particles obtained after soil aggregates dispersion. All SOM fractions can be separated sequentially on one soil sample by first isolating the DOM through centrifugation, separating the solid subnatant into a light POM and a heavy fraction by density (at 1.8 g/cm$^3$) and the latter into a heavy POM and a MAOM by wet sieving (at 53μm). NDF – Neutral detergent fibre; ADF – Acid detergent fibre; HWE – Hot-water extractable.

| | | ABOVEGROUND | BELOWGROUND 1st soil layer | BELOWGROUND $n$th soil layer |
|---|---|---|---|---|
| > 2 mm | HWE | C1$^a$ | C1$^{b1}$ | C1$^{bn}$ |
| | ADF | C2$^a$ | C2$^{b1}$ | C2$^{bn}$ |
| | NDF | C3$^a$ | C3$^{b1}$ | C3$^{bn}$ |
| > 53 μm (> 1.8g cm$^{-3}$) | | | C5$^{b1}$ | C5$^{bn}$ |
| (< 1.8g cm$^{-3}$) | | | C10$^{b1}$ | C10$^{bn}$ |
| > 0.45 μm | | | C9$^{b1}$ | C9$^{bn}$ |
| < 0.45 μm | | C6 | C8$^{b1}$ | C8$^{bn}$ |
| Not size defined | | C4$^a$ | C4$^{b1}$ | C4$^{bn}$ |

**Table S2 -** Optimized parameter values for the mid-point of the nitrogen modifier (*Nmid*), maximum decay rate for coarse, heavy particulate organic matter (*k5*), maximum decay rate for mineral-associated organic matter (*k9*) and maximum decay rate for light particulate organic matter (*k10*). Depending on what fraction was match (measured-modelled comparisons), different parameter values were derived. Root mean square error (RMSE) was minimised for each unique parameter set and assessed for each fraction (Mineral-Associated Organic Matter, MAOM; total Particulate Organic Matter, POM; bulk soil Soil Organic Carbon, SOC). Note that total POM refers to the composite of light and heavy POM measurements and the sum of the C5 and C10 pools). Analysis was performed on 154 forest and grassland sites from the LUCAS database – see Figure S2 and Figure S3 for more information.

| Parameter | Default (Initial optimized range) | Optimized for POM | Optimized for MAOM | Optimized for total SOC |
|---|---|---|---|---|
| ***Nmid*** | 1.750 (0.875 – 2.625) | 1.617 | 0.9223 | 2.4548 |
| ***k5*** | $5.00^{-4}$ ($6.0^{-5}$ – $1.0^{-3}$) | $5.766$$^{-4}$ | $2.376^{-4}$ | $2.51$$^{-4}$ |
| ***k9*** | $2.19^{-5}$ ($1.0^{-5}$ – $4.0^{-5}$) | $2.33$$^{-5}$ | $2.98$$^{-5}$ | $3.97^{-5}$ |
| ***k10*** | $2.96^{-4}$ ($1.0^{-4}$ – $1.0^{-3}$) | $4.31$$^{-4}$ | $2.94$$^{-4}$ | $3.01$$^{-4}$ |
| **RMSE between measured and modelled C stocks for 154 sites (Mg C ha⁻¹)** | | | | |
| Total SOC | 35.5 | 35.9 | 35.2 | 33.5 |
| POM-C | 23.4 | 23.5 | 23.1 | 25.5 |
| MAOM-C | 17.9 | 17.8 | 17.5 | 20.2 |

**Supplementary References**

Abramoff, R., Xu, X., Hartman, M., O'Brien, S., Feng, W., Davidson, E., Finzi, A., Moorhead, D., Schimel, J., Torn, M. & Mayes, M. A. (2018). The Millennial model: in search of measurable pools and transformations for modeling soil carbon in the new century. *Biogeochemistry*, 137(1-2), 51-71.

Bolinder, M. A., Janzen, H. H., Gregorich, E. G., Angers, D. A., & VandenBygaart, A. J. (2007). An approach for estimating net primary productivity and annual carbon inputs to soil for common agricultural crops in Canada. *Agriculture, Ecosystems & Environment*, *118*(1-4), 29-42.

Braakhekke, M. C., Beer, C., Hoosbeek, M. R., Reichstein, M., Kruijt, B., Schrumpf, M., & Kabat, P. (2011). SOMPROF: A vertically explicit soil organic matter model. *Ecological modelling,* 222(10), 1712-1730.

Campbell, E. E., Parton, W. J., Soong, J. L., Paustian, K., Hobbs, N. T., & Cotrufo, M. F. (2016). Using litter chemistry controls on microbial processes to partition litter carbon fluxes with the litter decomposition and leaching (LIDEL) model. *Soil Biology and Biochemistry*, 100, 160-174.

Canadell, J., Jackson, R. B., Ehleringer, J. B., Mooney, H. A., Sala, O. E., & Schulze, E. D. (1996). Maximum rooting depth of vegetation types at the global scale. *Oecologia*, 108(4), 583-595.

Harmon, M., and J. Domingo (2001), A User's Guide to STANDCARB Version 2.0: A Model to Simulate the Carbon Stores in Forest Stands, Dep. of For. Sci., Oreg. State Univ., Corvallis.

Jackson, R. B., Canadell, J., Ehleringer, J. R., Mooney, H. A., Sala, O. E., & Schulze, E. D. (1996). A global analysis of root distributions for terrestrial biomes. *Oecologia*, 108(3), 389-411.

Kalbitz, K., Schwesig, D., Rethemeyer, J., & Matzner, E. (2005). Stabilization of dissolved organic matter by sorption to the mineral soil. *Soil Biology and Biochemistry*, 37(7), 1319-1331.

Kätterer, T., Bolinder, M. A., Andrén, O., Kirchmann, H., Menichetti, L. (2011) Roots contribute more to refractory soil organic matter than aboveground crop residues, as revealed by a long-term field experiment. *Agriculture Ecosystems and Environment*, 141(1-2), 184–192.

Keiluweit, M., Bougoure, J. J., Nico, P. S., Pett-Ridge, J., Weber, P. K., & Kleber, M. (2015). Mineral protection of soil carbon counteracted by root exudates. *Nature Climate Change*, *5*(6), 588.

Kothawala, D. N., Moore, T. R., & Hendershot, W. H. (2008). Adsorption of dissolved organic carbon to mineral soils: A comparison of four isotherm approaches. *Geoderma*, 148(1), 43-50.

Ludwig, M., Achtenhagen, J., Miltner, A., Eckhardt, K. U., Leinweber, P., Emmerling, C., & Thiele-Bruhn, S. (2015). Microbial contribution to SOM quantity and quality in density fractions of temperate arable soils. *Soil Biology and Biochemistry*, *81*, 311-322.

Mayes, M. A., Heal, K. R., Brandt, C. C., Phillips, J. R., & Jardine, P. M. (2012). Relation between soil order and sorption of dissolved organic carbon in temperate subsoils. *Soil Science Society of America Journal*, 76(3), 1027-1037.

Moorhead, D. L., Lashermes, G., Sinsabaugh, R. L., & Weintraub, M. N. (2013). Calculating co-metabolic costs of lignin decay and their impacts on carbon use efficiency. *Soil Biology and Biochemistry*, 66, 17-19.

Poeplau, C. (2016). Estimating root: shoot ratio and soil carbon inputs in temperate grasslands with the RothC model. *Plant and soil*, 407(1-2), 293-305.

R Core Team (2018). R: A language and environment for statistical computing. R Foundation for Statistical Computing, Vienna, Austria. URL https://www.R-project.org/.

Sinsabaugh, R. L., Manzoni, S., Moorhead, D. L., & Richter, A. (2013). Carbon use efficiency of microbial communities: stoichiometry, methodology and modelling. *Ecology letters*, 16(7), 930-939.

Six, J., Conant, R. T., Paul, E. A., & Paustian, K. (2002). Stabilization mechanisms of soil organic matter: implications for C-saturation of soils. *Plant and soil*, 241(2), 155-176.

Sokol, N. W., Sanderman, J., & Bradford, M. A. (2018). Pathways of mineral-associated soil organic matter formation: Integrating the role of plant carbon source, chemistry, and point of entry. *Global change biology*. https://doi.org/10.1111/gcb.14482

Soong, J. L., Parton, W. J., Calderon, F., Campbell, E. E., & Cotrufo, M. F. (2015). A new conceptual model on the fate and controls of fresh and pyrolized plant litter decomposition. *Biogeochemistry*, 124(1-3), 27-44.

Toth G., Jones A., Montanarella L. (2013) LUCAS Topsoil Survey — methodology, data and results. In: JRC Technical Reports. European Union,

Luxemburg.

---

## Author Response (AR2)

Department of Soil and Crop Sciences,
Colorado State University,
Fort Collins,
Colorado 80523-1170
USA

18th February 2019

Cover Letter and Responses to Reviewer Comments to accompany the manuscript:
**"Unifying soil organic matter formation and persistence frameworks: the MEMS model"**

**Authors:** Andy Robertson, Keith Paustian, Stephen Ogle, Matthew Wallenstein, Emanuele Lugato, and Francesca Cotrufo

Thank you for your correspondence concerning our manuscript and for giving us the opportunity to resubmit a revised version. All comments from the reviewers have been carefully considered and appropriate responses are made below.

Sincerely,

Andy Robertson

**Responses to comments from Thomas Wutzler on "Unifying soil organic matter formation and persistence frameworks: the MEMS model" by Andy D. Robertson *et al*.**

Reviewer comments in bold and our responses in normal text. Selected new text in the revised manuscript is pasted here in italics. Reference to the manuscript is given as new line number (L).

General comments

**All my points have been answered.**
**The paper should be published.**
**My comments refer to line numbers in the author response.**

**Now that I can read Fig. 2, I have a few additional comments.**

**Fig. 2: I assume that panel mineral soil < 2mm indicates sum of the carbon pools. If this is correct, I suggest to explicitly state this equivalence.**

Yes, your assumption is correct. We have now added this level of detail to the figure legend.

L1051-1061:
*Figure 2 - Global sensitivity analysis results showing the relative contribution of each parameter to a change in carbon stock of each pool in MEMS v1.0 (leached carbon to deeper soil layers [pool C11] is omitted for clarity) after simulation to steady-state. The two top left panels represent the sum of soil pools (C5, C8, C9 and C10) and organic layer pools (C1, C2, C3, C4 and C6), respectively. Details of each parameter and the abbreviations used can be found in Table 2. The sensitivity analysis was repeated annually for simulation times between 1 and 100 years, every 10 years after that to 400-year simulations and every 100 years after that up to a 1000-year simulation. Results are presented on a log scale in years. The four parameters that were optimized in our analysis (Table S2) are coloured to highlight their importance in the different pools (mid-point of logistic curve where nitrogen content of input influences microbial carbon use efficiency, Nmid, red; maximum decay rate of heavy particulate organic matter, k5, orange; maximum decay rate of mineral-associated organic matter, k9, blue; maximum decay rate of light particulate organic matter, k10, green). A fully colourised version of these results can be in Figure S5.*

**Fig 2: I did not expect that the rate of the light POM (k10 green) would have such a high importance at centennial times, although the pool is stated to be much smaller than the MAOM pool. How do you explain this? Text at L484 states that its relative contribution diminishes, but I cannot see this from Fig 2.**

The light POM pool (C10) can dominate total soil C depending on the system (e.g., evergreen forest in cold sandy climates) – so this pool isn't always smaller than the MAOM. However, the conditions chosen for the sensitivity analysis were median values. In this case, the range of MAOM:POM pool sizes can be seen in that panel of figure 3 and the median is around a 2:1 ratio of MAOM:POM. The high relative sensitivity of total SOM to k10 is likely caused by that single parameter having almost all the influence on the light POM (C10) pool, whereas the MAOM (C9) pool is influenced by a number of different parameters. Overall, the MAOM parameters and light POM parameters do each account for ~45% of total SOM sensitivity, each. At centennial timescales, the relatively sensitivity for k10 impacts on total SOM does drop to around 45% from ~80% (the parameters that influence MAOM saturation take up more of the sensitivity below the green).

**L 489: I do not readily understand how Fig. 2 can be interpreted as a depiction of how each pool accumulates over time. Please, either omit or elaborate a bit more.**

Yes, this was poorly worded. We have changed the text as per below. Thanks for the suggestion.

L585-587:

*Figure 2 can be interpreted as a depiction of how the C pools of MEMS v1.0 are impacted by different parameters as each pool accumulates over time.*

**L230 Minor issue: It took me some time to understand that the comma after "(pool C8)" introduced a new main clause. I suggest rewording sentence to start with the topic of the section instead of the topic of the former section.**

We feel that the current phrasing is more appropriate as it links in directly from the previous section. We understand and appreciate the suggestion though.

**L600ff Logical leap: The text argues that Fig 2 shows that short-term parameters influence the immediate dynamics of the MAOM pool. Fig 2 is based on buildup of stocks from zero, where initial dynamics is of course governed by initial input from pools with fast dynamics. Contrary, the statement is very general and you would need to show that this also holds true for a disturbance to developed steady states. I suggest to either omit this point or to demonstrate the statement by a small simulation scenario in a supplementary.**

We agree that this assumption is currently untested. Consequently we have altered the text as per below.

L588-592:
*Many of the parameters that influence the processes of POM formation and persistence (e.g., LITfrg, Nmid, LCImax, etc.) have relatively high importance (i.e., sensitivity) to changes in total SOM within relatively short time frames (i.e., < 10 years; Figure 2). This may potentially capture the important real-world trend that POM is typically more vulnerable to decomposition with disturbance compared to MAOM (Cambardella and Elliott, 1992). However, disturbance impacts were not evaluated in the inaugural study.*